# Integrating artificial intelligence and optogenetics for Parkinson's disease diagnosis and therapeutics in male mice

Bobae Hyeon [1,2,3], Jaehyun Shin [4,14], Jae-Hun Lee [5,14], Woori Kim[2,3], Jea Kwon [6], Heeyoung Lee[1], Dae-gun Kim[7,8], Choong Yeon Kim [9,10], Sian Choi [1], Jae-Woong Jeong [9,11,12], Kwang-Soo Kim [2,3], C. Justin Lee [5] ✉, Daesoo Kim [7] ✉ & Won Do Heo [1,7,13] ✉

Parkinson's disease (PD), a progressive neurodegenerative disorder, presents complex motor symptoms and lacks effective disease-modifying treatments. Here we show that integrating artificial intelligence (AI) with optogenetic intervention, termed optoRET, modulating c-RET (REarranged during Transfection) signalling, enables task-independent behavioural assessments and therapeutic benefits in freely moving male AAV-hA53T mice. Utilising a 3D pose estimation technique, we developed tree-based AI models that detect PD severity cohorts earlier and with higher accuracy than conventional methods. Employing an explainable AI technique, we identified a comprehensive array of PD behavioural markers, encompassing gait and spectro-temporal features. Moreover, our AI-driven analysis highlights that optoRET effectively alleviates PD progression by improving limb coordination and locomotion and reducing chest tremor. Our study demonstrates the synergy of integrating AI and optogenetic techniques to provide an efficient diagnostic method with extensive behavioural evaluations and sets the stage for an innovative treatment strategy for PD.

Parkinson's disease (PD) is a globally prevalent neurodegenerative disorder characterised by significant motor impairments[1]. PD patients experience progressive and complex motor symptoms, including bradykinesia, postural instability, rigidity, tremor and gait abnormalities, largely attributed to the degeneration of dopaminergic (DA) neurons in the substantia nigra pars compacta (SNc)[1–5]. PD pathology is further marked by the deposition of Lewy bodies, which are predominantly composed of aggregated alpha-synuclein (aSyn), contributing to toxicity and disease propagation[2,6]. Although the aetiology of PD is still incompletely understood, advances in PD research have identified potential therapeutic candidates, including glial cell line-derived neurotrophic factor (GDNF) family ligands[1,4,5,7]. However, these insights have yet to culminate in the development of disease-modifying therapies, highlighting a critical unmet need in PD research[1,4,5,7,8].

In PD research, mouse models are indispensable for investigating disease mechanisms and evaluating potential treatments, yet

[1]Department of Life Sciences, Korea Advanced Institute of Science and Technology (KAIST), Daejeon, Republic of Korea. [2]Molecular Neurobiology Laboratory, McLean Hospital and Department of Psychiatry, Harvard Medical School, Belmont, MA, USA. [3]Program in Neuroscience, Harvard Medical School, Belmont, MA, USA. [4]Innovation, AKQA, Victoria, VIC, Australia. [5]Center for Cognition and Sociality, Institute for Basic Science (IBS), Daejeon, Republic of Korea. [6]Max Planck Institute for Security and Privacy (MPI-SP), Bochum, Germany. [7]Department of Brain and Cognitive Sciences, KAIST, Daejeon, Republic of Korea. [8]ACTNOVA, Daejeon, Republic of Korea. [9]School of Electrical Engineering, KAIST, Daejeon, Republic of Korea. [10]KAIST Information & Electronics Research Institute, Daejeon, Republic of Korea. [11]KAIST Institute for Health Science and Technology, Daejeon, Republic of Korea. [12]KAIST Institute for NanoCentury, Daejeon, Republic of Korea. [13]KAIST Institute for the BioCentury, KAIST, Daejeon, Republic of Korea. [14]These authors contributed equally: Jaehyun Shin, Jae-Hun Lee. ✉e-mail: cjl@ibs.re.kr; daesoo@kaist.ac.kr; wondo@kaist.ac.kr

traditional behavioural assessments often fall short, constrained by oversimplified metrics that fail to encapsulate the complex PD symptoms[6]. This limitation hinders a comprehensive understanding of the disease's progression and responses to treatments. Addressing this, recent advancements in artificial intelligence (AI) – particularly in pose-estimation techniques – have opened a new era in behavioural analysis, enabling precise measurement of a wide array of kinematic features, facilitating the detection of subtle behavioural nuances that were previously overlooked[9–12]. Building on this foundation, we initially introduced the AI Vision Analysis for Three-dimensional Action in Real-time (AVATAR) system in a preprint[13], which has been validated in several peer-reviewed studies[14–16]. Application of AVATAR and similar AI-driven frameworks promise a transformative shift in PD research, offering more nuanced and extensive insights into the characteristic behavioural patterns of disease phenotypes and opening new avenues for the development of targeted therapies.

As the functional receptor of GDNF family ligands, c-RET is central to their neuroprotective effects in DA neurons[7,17–19]. Several clinical trials have evaluated these ligands due to their promising neuroprotective properties; however, the outcomes have been largely inconclusive, with beneficial effects observed only in subsets of patients[1,4,5,7,8]. These inconsistencies highlight the need for alternative tools to investigate the crosstalk between PD-associated factors and c-RET signalling in PD models[8,20]. Recently, we introduced a genetically encoded optogenetic tool termed optoRET as a proof-of-concept platform for precisely modulating c-RET signalling in the SNc of living mice[21]. Unlike traditional ligand-based approaches, optoRET enables temporally controlled, ligand-independent activation, offering a versatile experimental framework for exploring c-RET-targeted therapeutic strategies. This temporal control may be particularly valuable, as prolonged exposure to GDNF ligands has been associated with diminished neuroprotective efficacy[22,23]. Although findings have not been entirely consistent, early studies reported that c-RET downregulation may contribute to the limited efficacy of GDNF in aSyn overexpression models[24,25]. The interaction between aSyn and c-RET appears to be context-dependent and influenced by regulatory factors such as Nurr1 and Nedd4[8]. Notably, a more recent study demonstrated that activating GDNF/RET signalling can mitigate aSyn pathology and preserve DA neurons in an aSyn-preformed fibril model[26]. Given these complexities, overexpression of optoRET could offer an additional advantage by enabling c-RET modulation independently of endogenous receptor levels. Whilst aSyn overexpression models have recognised limitations[8], evaluating the therapeutic potential of optoRET in such models remains valuable to determine whether targeted and temporally controlled c-RET modulation via an exogenous protein could effectively address the current challenges faced by GDNF-based therapies.

By integrating optoRET with an advanced AI-based methodology, we aimed to achieve two objectives in this study. First, we aimed to present an AI-based system that leverages pose estimation for a nuanced assessment of PD in freely moving mice, whilst capturing a broad spectrum of PD symptoms and providing detailed behavioural characteristics. Second, we sought to evaluate the efficacy of optoRET in mitigating PD symptoms. This innovative integration of optogenetic and AI-based technologies aims to advance the development of effective diagnostic and therapeutic strategies for PD, addressing critical unmet needs in neurodegenerative disease research.

## Results

### hA53T PD mouse model with two distinct PD severity cohorts
To establish an aSyn PD mouse model, we utilised viral-mediated overexpression of human A53T-aSyn (hA53T) – the first identified mutation in familial PD, known to facilitate fibril formation[6,27–30]. Although A53T represents a wild-type (WT) variant in rodents, the introduction of hA53T is pathogenic in mice due to the sequence

differences in other residues[31,32] (Supplementary Fig. 1a). Despite the limitation that aSyn levels may exceed pathological levels observed in human PD[33], numerous studies have validated hA53T overexpression as a robust PD mouse model, recapitulating key pathological hallmarks and the progressive development of behavioural symptoms[6,28–30,34,35]. Although bradykinesia is a canonical symptom of PD, aSyn overexpressing animal models often exhibit hyperactivity, affected by the upregulation of the nigral DA receptor D1 and the downregulation of the striatal DA transporter[6,29,30,34].

We injected an adeno-associated virus (AAV), DJ/8 serotype, to transduce a red fluorescent protein (RFP) or hA53T into the SNc of B6J male mice. Initial tests with unilateral hA53T injections failed to yield the expected dose-dependent PD severity, as assessed by the elevated beam walking test (BWT; Supplementary Fig. 1b-e), leading us to refine the protocol to bilateral injections (Fig. 1a). The control (CT) group consisted of four subgroups, including mice with no virus injection (NI), empty vector (EV), 1X RFP (R1), 5X RFP (R5) virus injections; and two PD groups with 1X or 5X doses of hA53T (A1 or A5) into the bilateral SNc regions (Fig. 1b; Supplementary Table 1). To exclude the possibilities that different genetic backgrounds could affect the PD diagnostic system, we included DAT-CRE[36] line, which was used for the expression of DIO-optoRET[21] in later experiments (Supplementary Table 1).

To assess the motor dysfunction, a set of behavioural tests were performed, including the BWT, rotarod test (RRT), open-field test (OFT) and tail suspension test (TST) with a multi-vision recording system (AVATAR studio), followed by the histological analysis of the striatal TH fibre density and nigral DA cell counts (Fig. 1a). The comparative analysis of the CT subgroups suggested that mechanically induced lesion (MIL, i.e., EV group) did not significantly worsen the motor function nor decrease the nigral DA cell counts (Fig. 1a–h; Table 1; Supplementary Fig. 2). Contrarily, the protein overload lesion (POL, i.e., R1 or R5 groups) showed evidence of toxicity affecting motor function in RRT (R5, $p = 0.0254$) and DA cell integrities. Notably, the R5 group showed approximately 29% SNc DA neuronal loss ($p = 0.0325$), despite no significant decrease in the striatal TH fibre density, which is in line with the previous studies[37,38].

In the PD groups, motor dysfunction progressively worsened over time and the changes were significant at the endpoint (10 wk), compared to the EV group (Fig. 1c–f). The RRT score (RRS) in A5 decreased more significantly than in A1 (RRS [%]: EV = 94.40 ± 3.67, $n = 10$; A1 = 48.84 ± 12.20, $n = 5$; A5 = 28.72 ± 5.98, $n = 6$) and the BWT score (BWS) between the two PD groups was significantly different ($p = 0.0020$; BWS [%]: EV = 78.12 ± 6.66, $n = 10$; A1 = 287.6 ± 23.92, $n = 11$; A5 = 381.6 ± 31.53, $n = 13$). Further behavioural assessments revealed that our PD mice exhibited various PD phenotypes, including altered locomotion and motor coordination, tremor and dystonia (Supplementary Fig. 3; Supplementary Movies 1–5)[6,29,30]. In the histological analysis, both PD groups exhibited significant decreases in striatal TH fibre density (STR: A1 = 50.07 ± 4.82%, $n = 9$; A5 = 31.41 ± 1.68%, $n = 5$) and in nigral DA cell counts (SNC: A1 = 38.03 ± 2.08%, $n = 9$; A5 = 22.00 ± 2.57%, $n = 5$; Fig. 1g, h; Supplementary Fig. 2b). The differences between the two PD severity cohorts were also significant ($p = 0.0148$ and 0.0201 for striatal TH and nigral DA counts, respectively), with the latter corresponding to a 26% greater cell loss in the A5 group compared to the A1 group. Collectively, the results indicate that bilateral alpha-Synuclein-induced lesion (AIL) in the SNc of mice induces significant PD symptoms and pathology in a dose-dependent manner, although POL per se exerts some toxicity (Supplementary Note 1).

### AI-based PD diagnosis for precision and sensitivity in mice
In our previous result, we used traditional task-dependent behavioural tests to assess motor dysfunction in our PD model mice. However, these tests are limited to specific metrics, missing the complexity of PD symptomatology (Supplementary Fig. 3). This prompted us to employ

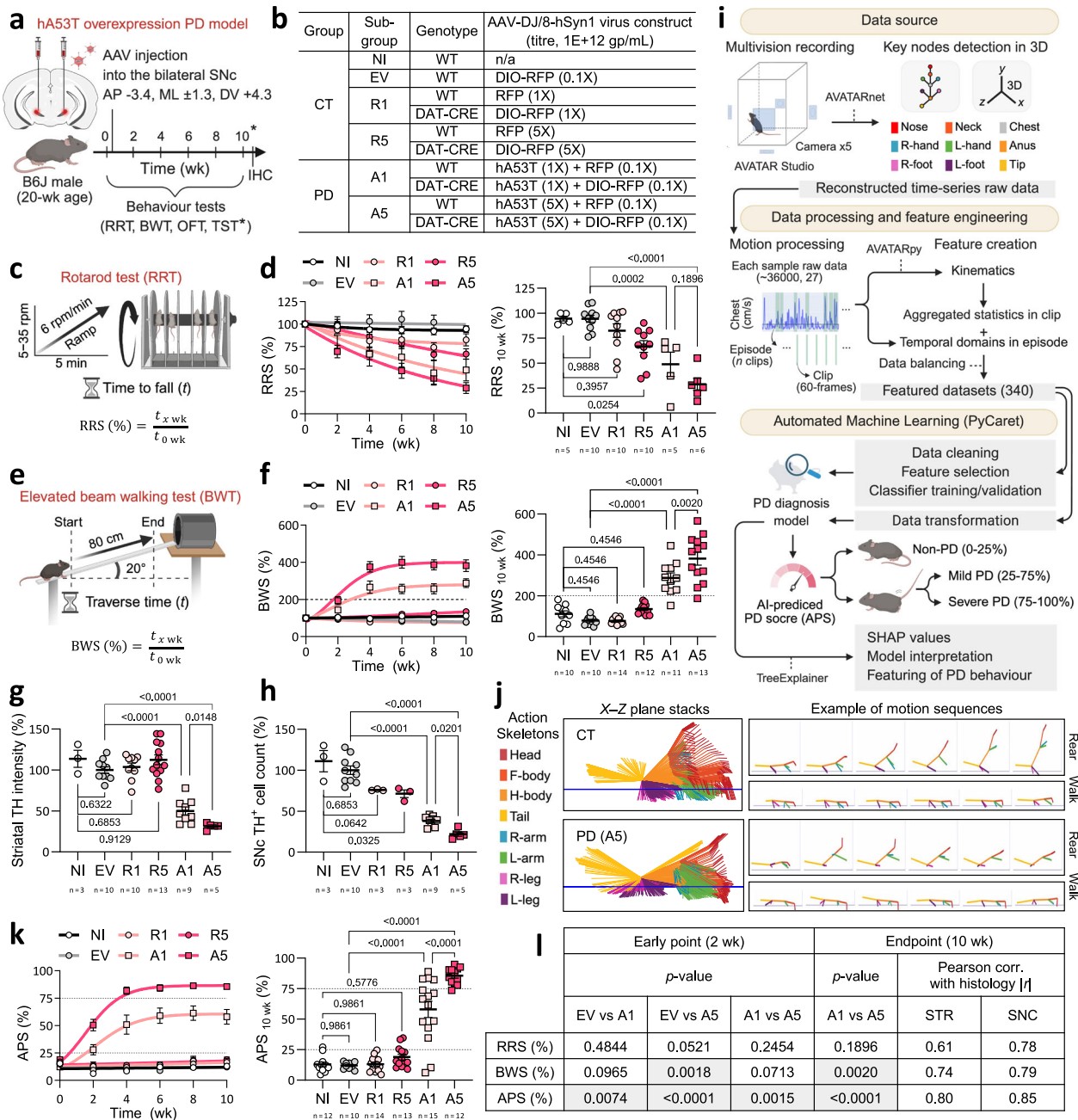

**Fig. 1 | AI-powered diagnosis in distinct severity cohorts of the hA53T PD mouse model. a** Schematic of the experimental setup for establishing the bilateral hA53T PD mouse model, highlighting stereotaxic injection sites into the SNc of B6J male mice. **b** Table summarising experimental groups: control (CT: NI, EV, R1, R5) and PD (A1, A5), detailing the AAV-DJ/8-hSyn1 constructs and doses. See Supplementary Table 1 for details. **c, e** Schematics of the rotarod test (RRT) and elevated beam walking test (BWT), indicating how the respective scores (RRS and BWS) are derived. **d, f** RRS and BWS comparisons between CT and PD groups shown as fitted line and dot plots, longitudinally (0–10 wk) and at the 10 wk endpoint. **g, h** Comparative histological analysis between CT and PD groups, showing striatal TH intensity (STR) and SNc cell counts (SNC), normalised to the EV group (%) at 10 wk. **i** Machine learning (ML) pipeline for developing an AI model to diagnose and assess PD severity. The AVATARnet architecture is adapted from YOLOv4 and modified to include 53 CNN layers. 3D coordinates are computed using triangulation and bundle-adjustment. AI techniques are indicated by dashed lines,

with automated ML processes utilising PyCaret highlighted at the bottom. **j** Representative 3D reconstructed action skeletons of motion sequences in the $X$–$Z$ plane from the CT and PD clusters, with the anus as the reference point. Skeleton images of motion sequences were sampled with a temporal offset: every 5 frames for rearing and 3 frames for walking. **k** Comparisons of XGB model predicted PD scores (AI-predicted PD scores, APS) between CT and PD groups (unseen mice), shown in fitted line and dot plots longitudinally (0–10 wk) and at the 10 wk endpoint. Dashed lines on the graphs indicate health status boundaries (%): non-PD (NP; 0–25), mild PD (25–75), and severe PD (> 75). **l** Table summarising comparative results from behavioural assessments (RRS, BWS and APS). Significant *p*-values are highlighted by shading. Data are shown as mean ± SEM; sample sizes (*n*) are indicated in plots. All statistical analyses were performed using one-way ANOVA followed by Holm–Sidak post hoc corrections for multiple comparisons. Schematics were created in BioRender. Heo, W. (2025) https://BioRender.com/cykyauz. Source data are provided in the Source Data file.

**Table 1 | Quantitative comparison of analytic metrics across experimental groups affecting the DA system**

| | Group mean ± SEM (n) in % at endpoint (10 wk) | | | | | |
| --- | --- | --- | --- | --- | --- | --- |
| **Group** | **Control (CT)** | | | | **PD** | |
| **Sub-Group** | **No injection (NI)** | **Empty vector (EV)** | **RFP low dose (R1)** | **RFP high dose (R5)** | **hA53T low dose (A1)** | **hA53T high dose (A5)** |
| **Lesion type** | **N/A** | **Mechanically induced (MIL)** | **Protein overload (POL)** | | **Alpha-Synuclein-induced (AIL)** | |
| RRS | 94.54 ± 2.29 (5) | 94.40 ± 3.67 (10) | 82.54 ± 6.70 (10) | 66.67 ± 5.87 (10) | 48.84 ± 12.20 (5) | 28.72 ± 5.98 (6) |
| BWS | 111.6 ± 14.50 (10) | 78.12 ± 6.66 (10) | 76.71 ± 4.03 (14) | 134.6 ± 7.17 (12) | 287.6 ± 23.92 (11) | 381.6 ± 31.53 (13) |
| APS | 12.94 ± 1.92 (12) | 12.11 ± 0.85 (10) | 13.04 ± 1.60 (14) | 18.96 ± 2.34 (13) | 58.01% ± 6.69 (15) | 85.61% ± 1.94 (12) |
| STR | 113.6 ± 10.39 (3) | 100 ± 3.89 (10) | 103.8 ± 4.64 (10) | 112.5 ± 6.01 (13) | 50.07 ± 4.82 (9) | 31.41 ± 1.6 (5) |
| SNC | 111.1 ± 12.98 (3) | 100 ± 5.17 (10) | 75.90 ± 0.58 (3) | 71.33 ± 4.73 (3) | 38.03 ± 2.08 (9) | 22.00 ± 2.57 (5) |

an AI-based pose estimation technique for a more comprehensive analysis that reveals detailed behavioural phenotypes crucial for accurate and early PD diagnosis. With the AVATAR system, we reconstructed the 3D motion sequences of freely moving mice from the OFT video recordings in the AVATAR studio (Fig. 1i). This system detects key nodes of a mouse's body in each movie frame and returns the 3D coordinates of the nodes (Fig. 1i).

Initially, we reduced the dimensions of the pose datasets to 3D using t-distributed stochastic neighbour embedding (t-SNE) and analysed the temporal variability in the CT and PD (A5) groups. The CT group's t-SNE plots showed no time-dependent patterns, whereas the A5 group data clustered temporally (Supplementary Fig. 4a). Using k-means, we identified seven clusters, including two distinct PD clusters that grew with PD progression (Fig. 1j; Supplementary Fig. 4c, d). However, interpreting behavioural data in an unsupervised manner has proven challenging, prompting us to create feature-engineered datasets to train classifiers in a supervised fashion.

Through motion processing of the pose data, each of the continuous moving sequence was assigned as a unit of an Episode, which was subdivided into short clips (2 s) (Fig. 1i). We extracted kinematic features, aggregated them into statistical descriptors for each clip and computed temporal domain parameters for episodes, resulting in a total of 340 features (Supplementary Fig. 5). We prepared the feature-engineered datasets from the CT and A5 groups for training and validation of various binary classification models with PyCaret[39], a low-code automated machine learning (ML) framework, in Python (Supplementary Fig. 6a, b; Supplementary Table. 2). After evaluating 15 models and applying feature selection procedures, we identified 30 optimal features (Supplementary Figs. 3c, d, 5b). The Extreme Gradient Boosting (XGB) model achieved the best performance in terms of accuracy, recall and F1 scores (the harmonic mean of precision and recall), correctly classifying 90% of the cross-view validation dataset (Supplementary Figs. 6e, f, 7; Supplementary Note 2).

The XGB model classified each clip of moving episodes as non-PD (NP) or PD and the proportion of PD clips over total clips for each mouse was expressed as an AI-predicted PD score (APS, %; Supplementary Fig. 6g). We further evaluated our model performance by testing an unseen-mouse dataset (Fig. 1k). At the endpoint (10 wk), APS was significantly higher in the A1 (58.01% ± 6.69, n = 15) and A5 (85.61% ± 1.94, n = 12) groups, compared to the EV group (12.11% ± 0.85, n = 10). The difference between the two PD severity cohorts was highly significant (p < 0.0001), indicating that APS is a more sensitive method than RRS or BWS (p = 0.1896 and 0.0020, respectively; Fig. 1l).

Based on the data distribution, we defined APS ranges for mouse health status as follows: 0–25 as non-PD (NP), 25–75 as mild PD and 75–100 as severe PD (Fig. 1k). Notably, at 2 wk post-surgery, both PD groups were assessed as mild PD status, with significantly higher APS than the EV group (EV = 9.42 ± 1.15, n = 10; A1 = 27.94 ± 4.70, n = 14, p = 0.0074; A5 = 50.41 ± 5.23, n = 13, p < 0.0001; Supplementary Fig. 8).

This indicates that APS enables early detection of PD, whereas RRS and BWS fail to distinguish mild PD at this early time point (Fig. 1l).

To evaluate the specificity of APS for PD, we assessed female SOD1-G93A[40] mice, a model of amyotrophic lateral sclerosis (ALS), a motor neuron disease (MND) (Supplementary Fig. 9). Despite marked motor dysfunction in the ALS group (RRS = 6.11 ± 1.85%, n = 2), their APS remained low (4.24 ± 1.38%, n = 2). These results support the conclusion that the high APS observed in PD groups reflects PD-specific pathology, rather than general motor deficits.

To further corroborate these findings, we performed correlation analyses between the behavioural assessments (RRS, BWS and APS) and histological data. Datasets (CT and PD groups) were filtered for each pair of metrics to compute the Pearson correlation coefficients (r; Supplementary Figs. 10, 11). The analysis revealed that APS has the highest |r| with most pairs of metrics analysed, even higher than the STR/SNC pair, which was largely decreased by R5 group (Supplementary Fig. 10, plot sets #6–8). Additionally, the buffering effect of DA system was evident in the APS and BWS correlation graphs (Supplementary Fig. 11, yellow ovals), reflecting the pre-symptomatic period before 40–60% of DA neuronal death[41], a feature absent in RRS. Collectively, the results indicate that APS detects PD with high precision, is sensitive in discriminating between different PD severity cohorts, enables early PD detection and strongly correlates with other behavioural assessments and pathology of the DA system (Fig. 1l; Supplementary Note 3).

## Top behavioural features impacting AI PD prediction in mice

To gain insight into the characteristic features of hA53T PD mice, we employed an explainable AI technique, the TreeExplainer[42,43], to decipher the XGB model's decision-making process (Supplementary Fig. 6g). TreeExplainer revealed key features impacts on predictions, encapsulated in SHapley Additive exPlanations (SHAP, v0.41.0)[42,43] values that quantify each feature's influence on the PD likelihood. The 20 most impactful features were identified and their effects were visualised in a SHAP summary plot, offering a concise yet comprehensive view of the model's analytical depth (Fig. 2a). Supplementary analyses further validated the stability and generalisability of these top 20 features (Supplementary Figs. 12, 13; Supplementary Notes 4, 5).

Next, we assessed longitudinal variability of the top 20 features using kernel density estimate (KDE) plots and Kullback–Leibler (KL) divergence. Data distributions for most features in the CT group remained relatively stable over time, whereas those in the PD (A5) group showed significant variability, highlighting distinct behavioural changes associated with PD progression (Supplementary Figs. 14–17; Supplementary Tables 3,4). The ALS group also exhibited notable variability in 75% of the features; however, their divergence patterns differed markedly from those observed in the PD group, with 40% of the features changing in the opposite direction (Supplementary Figs. 18–22; Supplementary Tables 5–7). Although this single-feature-focused analysis provides valuable insights, it does not capture

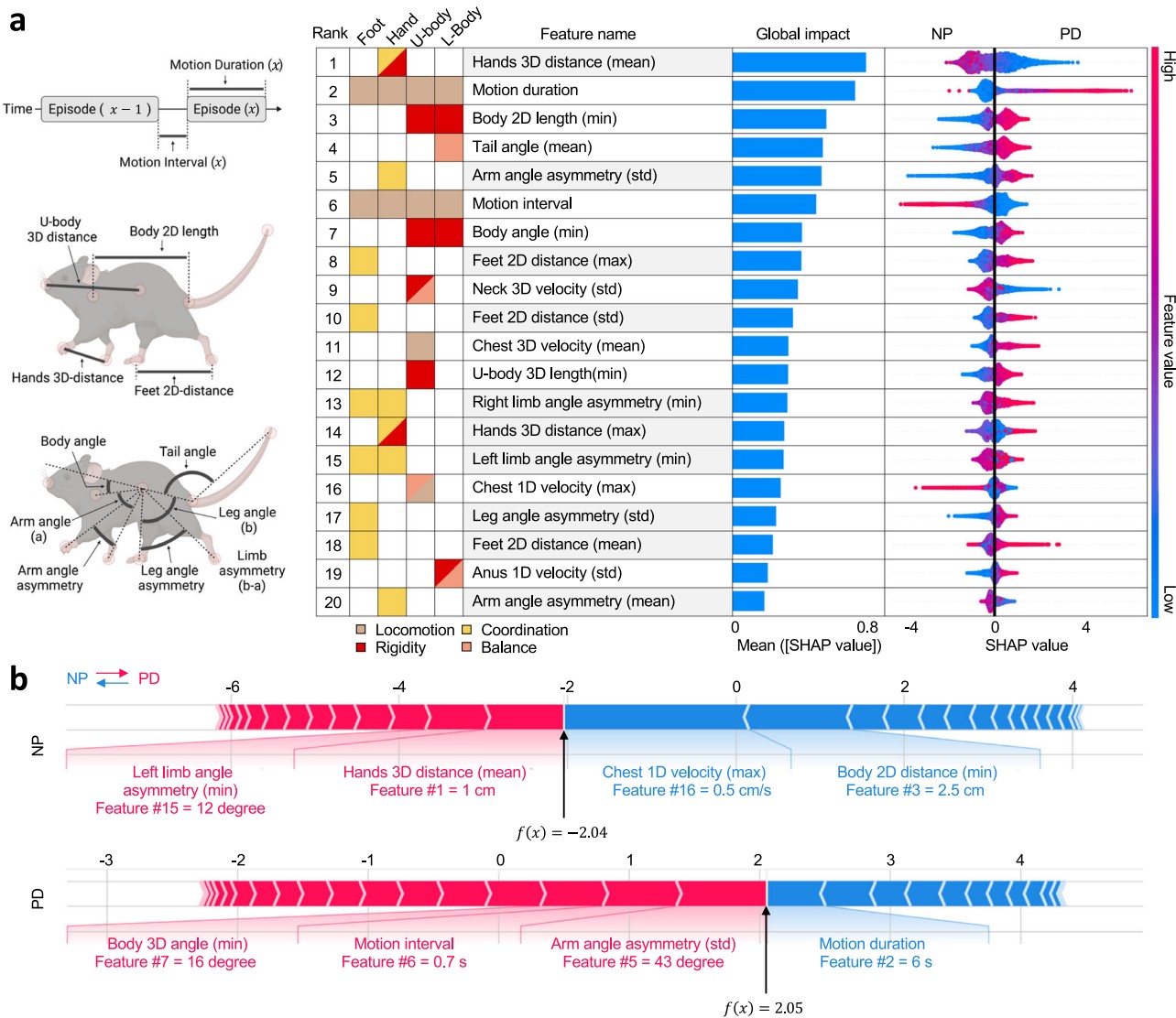

**Fig. 2 | Exploring PD phenotypes through top 20 behavioural features with insights from XGB model interpretation. a** Graphic summary of the top 20 features with schematics of key feature components. Features ranked by their global impact on the XGB model, depicted by blue horizontal bar graphs (from the validation dataset). The colour-coded boxes detail the feature-associated body parts and the PD symptomatic categories. The beeswarm plots (SHAP summary plot) offer local explanations for each feature, with individual dots representing motion clips. The position of each dot is determined by SHAP values (log odds of PD likelihood), reflecting its impact direction and magnitude on model prediction (negative and positive for non-PD [NP] and PD, respectively). The dot colour represents the magnitude of feature value. Clusters of dots denote the prevalence of a feature's effect on the model output. Limb-associated features are further highlighted with shading. **b** Representative SHAP force plots for motion clips correctly classified as NP and PD, illustrating the integration of feature impacts on model predictions. Each plot aligns feature contributions (SHAP values) along the x-axis, culminating in the total effect denoted as $f(x)$, with the main features annotated with their rank and feature values. Schematics were created in BioRender. Heo, W. (2025) https://BioRender.com/cykyauz.

potential interactions or combined effects of multiple features, thus warranting cautious interpretation.

The XGB model's predictive strength stems from the aggregated influence of all features, as reflected in their SHAP values (Fig. 2b). High-ranking features are pivotal predictors that markedly influence model outcomes. Nonetheless, lower-ranked features may still significantly impact specific instances, as evidenced by the extended tails in the beeswarm plots, representing high-magnitude of SHAP values (Fig. 2a). Thus, the model relies not on single-feature SHAP values but on the cumulative effect of all features per instance (Fig. 2b). Another advantage of our model is its ability to account for feature interactions. SHAP dependence plots for each feature illustrate the influence of other features as the degree of vertical dispersion of the data points (Supplementary Fig. 23). We employed colour coding to represent the values of the most interacting feature to the plotted feature,

elucidating the relationship between them, which otherwise would remain undetermined.

**Disrupted limb movement coordination in PD mice**

The XGB model pinpointed the top 20 features as crucial for PD diagnosis, particularly emphasising limb movements by ranking half of these features related to hands and feet (highlighted with shading, Fig. 2a). Interlimb asymmetry was a significant factor, 50% of limb features showing clear divisions in PD, contrasting with the more uniform distribution in NP (Supplementary Figs. 23, 24). Additionally, high standard deviations for lateral limb movements were noted in PD mice. The mean hands 3D distance (#1) in NP mice ranged from 2–3 cm, whereas in PD mice it was 0.5–1.7 cm, indicating restricted hand mobility (Supplementary Figs. 23, 24). Similarly, the maximum feet 2D distance (#8) was wider in PD mice (3.52–5 cm) than in NP mice

(2–3.62 cm), indicating a wider stance in PD. Our findings highlight that limb movement patterns are critical differentiators between PD and NP mice. Specifically, PD mice exhibit disrupted limb motor coordination, characterised by variable movement patterns, small hand movement amplitudes and wider stances.

## Altered locomotion and postural imbalances in PD mice

Motion duration (#2) and motion interval (#6) further differentiated PD from NP mice, longer durations (>30 s) and motion intervals (>150 s) serving as strong indicators of PD and NP, respectively, as evidenced by right-sided and left-sided red tails in beeswarm plots (Fig. 2a; Supplementary Figs. 23, 24). Longitudinal analysis supports these findings, revealing progressively increased motion duration in the A5 group and increased motion intervals in the CT group over time (Supplementary Figs. 14–17). The latter likely reflects habituation to repeated exposure to the same environment – an effect notably absent in the A5 group. Our findings, highlighting increased motion durations and the absence of habituation effects on motion intervals as distinguishing PD characteristics, corroborate earlier observations that aSyn-overexpressing animal models frequently exhibit hyperactivity[6,29,30,34].

SHAP analysis of maximum chest 1D velocity (#16) delineated two data ranges for both NP and PD mice, indicating distinct movement patterns (Supplementary Fig. 23). The major feature value range indicative of PD was much lower than that of NP (0.1–0.3 cm/s and 0.3–0.5 cm/s, respectively; Supplementary Figs. 23, 24). Notably, KDE plots of this feature in the A5 group showed a progressive narrowing of the higher velocity range as PD progressed, corresponding to a significant decrease in the percentage of data above the key value – from 18% to 1% over 0 to 10 wk (Supplementary Fig. 15). As high-velocity vertical movements are typically associated with rearing behaviour in mice, this finding implies that PD mice may exhibit slower or more gradual vertical movements, suggesting postural imbalances such as difficulties maintaining an upright position. Despite its lower rank, the pronounced left-sided red tail in the beeswarm plot underscores rapid vertical chest movements during rearing as a strong indicator of NP mice (Fig. 2a). The SHAP force plot for a representative NP clip illustrates how a high value of this feature influences the model outcome (Fig. 2b).

## Increased rigidity in core body regions of PD mice

In the analysis of the core body regions (including neck, chest, anus), the minimum body 2D length (#3), as a critical feature, indicated that NP values were below and PD values above the key value of 4.63 cm – suggesting increased rigidity in PD body posture (Fig. 2 and Supplementary Figs. 23, 24). Conversely, higher values for the minimum body angle (#7) were associated with PD, indicating a more stooped posture than in NP mice (Fig. 2 and Supplementary Figs. 23, 24). SHAP analysis of the standard deviation of the neck 3D velocity (#9) revealed two distinct data clusters for PD mice – a major range of 2–4 cm/s and a minor range of 8–12 cm/s – in contrast with the single range of 4–11 cm/s observed in NP mice, indicating more consistent neck movement dynamics in the latter (Supplementary Figs. 23, 24). In addition, longitudinal analysis suggested a nonlinear progression in the neck movement patterns of PD mice, with variability initially increasing and then decreasing after an inflection point at 6 wk (Supplementary Figs. 15, 16). Collectively, these findings suggest that PD mice exhibit not only rigid but also stooped body postures and experience transitions in the pattern of neck movement dynamics, reflecting disease progression.

## Optogenetics mitigates PD progression in mice

Given the complexities surrounding c-RET and its potential as a therapeutic target for PD[30,44], we investigated the therapeutic impact of optoRET[21] in our aSyn PD mouse model, aiming to determine whether selective and temporally controlled modulation of c-RET signalling could present a viable alternative to conventional GDNF therapies. We co-administered AAV-DJ/8-hSyn1-DIO-optoRET and hA53T (A1 or A5) to the SNc regions in the mouse brain, followed by behavioural assessments (Fig. 3a). We automated optoRET activation using the Wireless Network for Behavioural Neuroscience (WNBN)[45], an internet-based control system, with blue LED cage lids, enabling scheduled light exposure without manual intervention. Light schedules – daily (S#1), biweekly (S#2), or alternate days (S#3) – were programmatically set via the user-friendly website.

The APS indicated that optoRET activation was statistically ineffective in the A5 mice but significantly inhibited PD progression in the A1 mice under the S#2 and S#3 schedules (A1O2, A1O3; Fig. 3b, c; Supplementary Fig. 25; Supplementary Table 8; Supplementary Note 6). Notably, none of the A1 mice in either the A1O2 or A1O3 groups were classified as having severe PD (Fig. 3b). Both groups showed no significant differences compared to the EV group but differed significantly from the untreated A1 group at 10 wk (Supplementary Fig. 25d). Specifically, the A1O3 group also showed consistent improvement in BWS, again demonstrating no significant difference compared to the EV group but differing significantly from the untreated A1 group at 10 wk (Supplementary Fig. 25f). Histological analysis revealed that optoRET photoactivation under the alternate days (S#3) schedule significantly mitigated the reduction in the striatal TH fibre density and nigral DA neuronal count in A1 mice (Fig. 3d, e). Furthermore, correlation analyses between the histological data and the behavioural data from EV, A1 and A1O3 groups validated our findings (Supplementary Fig. 25g) and supported previous observations (Supplementary Fig. 11).

To further validate the efficacy of our optoRET-based therapeutic approach, we evaluated two additional treatments: the standard PD symptomatic medication L-DOPA (50 mg/kg[46], i.p., administered daily [A1L1] or alternate days [A1L3]) and the preclinically validated RET agonist (BT44[47–49], 120 μM, delivered via constant-rate direct brain infusion [A1BT]). We assessed treatment efficacy with RRS, BWS and APS (Supplementary Fig. 26a–i). Under our experimental conditions, neither treatment provided significant beneficial effects in terms of these behavioural assessments (detailed interpretations of the BT44 and L-DOPA treatment groups are provided in Supplementary Notes 7, 8, respectively). Interestingly, both L-DOPA groups (A1L1, A1L3) developed pronounced trunk/neck dystonia, consistent with previous reports, reflected by significantly reduced mean axial bending angle (Supplementary Fig. 26j–m)[50–52]. Collectively, these results indicate that optoRET photoactivation effectively mitigates PD progression and neurodegeneration in the hA53T PD mouse model, contingent upon specific light application schedules.

## Optogenetics improves limb coordination in PD mice

To understand why the A1O3 group exhibited the most effective improvements in behavioural assessments, we leveraged interpretations from the XGB model. Firstly, decision tree analysis revealed that only 16% of the randomly selected motion clips in the A1O3 group were classified as PD (Supplementary Fig. 27). In addition, KDE plots demonstrated that the feature distributions for the A1O3 group more closely resembled those of the CT group (Supplementary Fig. 28). Furthermore, the KL divergences from the CT counterpart were generally lower in the A1O3 group than in the A1 group, suggesting a reduction in PD-related alterations (Supplementary Fig. 29 and Supplementary Table 9). Moreover, the mean feature values of the A1O3 group were more closely aligned with those of the CT group compared to the other PD groups (Supplementary Figs. 30, 31; Supplementary Tables 10–15).

Following statistical analyses, we assessed the PD-affected features across PD groups and performed the treatment response evaluation (TRE) in the A1O3 group (Fig. 3f and Supplementary Fig. 31). In

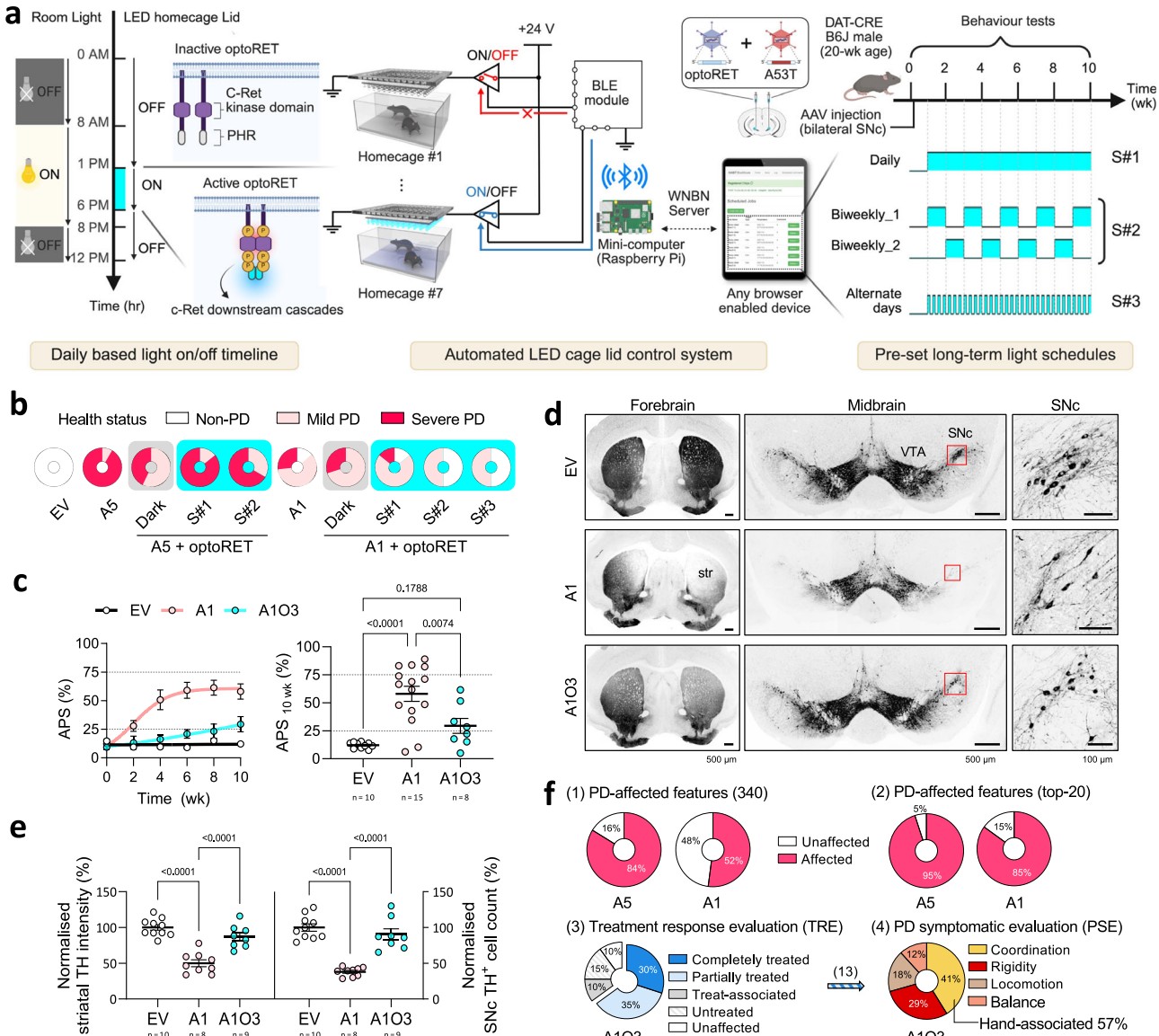

**Fig. 3 | Evaluation of optoRET in alleviating PD symptoms and neurodegeneration. a** Schematic of the experimental setup for optoRET treatment in PD mice. On the left side, timeline presents daily based room light and LED cage lid on/off times. The centre illustrates the automated LED cage lid control system, triggering the daily-based LED toggle signals at pre-set schedules. On the right side, the pre-set long-term light schedules are depicted: daily (S#1), biweekly (S#2), or alternate days (S#3). **b** Pie charts of health status of mice in non-PD (NP), mild, or severe PD statuses. Groups with optoRET unstimulated (dark) or stimulated with light schedules (S#1–3) are highlighted with grey or cyan background, respectively. **c** Comparisons of XGB model predicted PD scores (APS) between EV, A1, and A1O3 groups, shown in fitted line and dot plots longitudinally (0–10 wk) and at the 10 wk endpoint. **d** Representative TH-stained IHC images of forebrain and midbrain sections. Red boxes indicate the SNc regions shown in the magnified images on the right. **e** Comparative histological analysis between EV, A1, and A1O3 groups,

showing striatal TH intensity (STR) and SNc cell counts (SNC), normalised to the EV group (%) at 10 wk (STR [left plots, %]: 100 ± 3.89, 50.07 ± 4.82, 87.18 ± 5.62; SNC [right plots, %]: 100 ± 5.17, 38.03 ± 2.08, 90.30 ± 7.84, respectively). **f** Pie charts, detailing the assessment of PD-affected features, treatment response evaluation (TRE), and PD symptomatic evaluation (PSE). Arrow in between the lower charts indicate the treated segment for further analysis, with feature counts noted in brackets. Data are shown as mean ± SEM; sample sizes (*n*) are indicated in plots. All statistical analyses were performed using one-way ANOVA, with Tukey's post hoc correction in panel c and Holm–Sidak correction in panel d for multiple comparisons. Schematics were created in BioRender. Heo, W. (2025) https://BioRender.com/cykyauz. Source data are provided in the Source Data file. str striatum, SNc substantia nigra pars compacta, VTA ventral tegmental area, TH tyrosine hydroxylase. WNBN Wireless Network for Behavioural Neuroscience, BLE Bluetooth low energy.

the A1 group, 85% of the top 20 features were affected, whereas in the A1O3 group, 65% were treated, leaving only 15% untreated. In the PD symptomatic evaluation (PSE) of the treated features, improvement in coordination – particularly in hand movements – was most pronounced. When comparing TRE in the A1O2 group, we observed complete overlap in the treated features; however, two additional beneficial effects were observed exclusively in the A1O3 group: standard deviation of the feet 2D distance (#10) and minimum upper body 3D length (#12) (Supplementary Fig. 31a). Collectively, these results

indicate that optoRET photoactivation exerts beneficial effects on motor function in mild hA53T PD mice, notably by enhancing limb coordination.

## Optogenetics improves the spectral signatures of PD mice

Our initial feature engineering, focused on the statistical attributes of motion sequences, might have underrepresented spectro-temporal aspects of PD symptoms. To address this, we employed the Time Series Feature Extraction Library (TSFEL)[53] in Python for a comprehensive

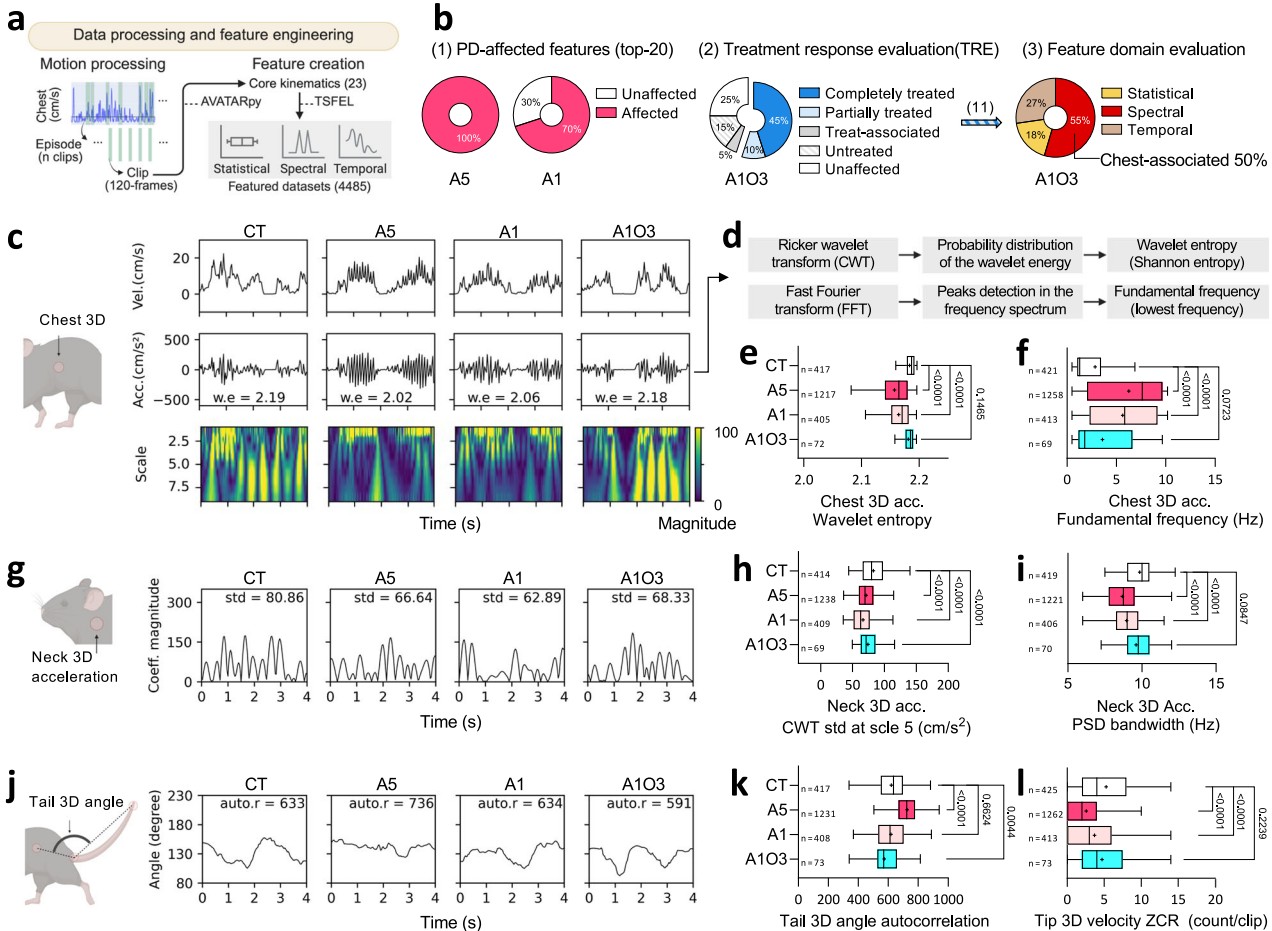

**Fig. 4 | Spectro-temporal insights into optoRET intervention, exploring tremor alleviation and movement complexities in PD. a** ML pipeline, highlighting the feature engineering process in the TSFEL model to identify key statistical and spectro-temporal features of PD behaviour. **b** Pie charts, detailing the evaluations of PD-affected features, treatment responses, and treated feature domains. Arrow in between the charts (2 and 3) indicate the treated segment for further analysis, with feature counts noted in brackets. **c** Representative line graphs of chest 3D velocity (vel.) and acceleration (acc.) over time, alongside scalograms from Ricker wavelet transformation (CWT) of the latter. Computed wavelet entropy (w.e) of the acc. signal is marked on each graph. **d** Diagrams of the computational processes for the wavelet entropy and fundamental frequency of chest 3D acc. data. **e, f** Plots of wavelet entropy and fundamental frequency of chest 3D acc. signals at 10 wk. **g** Representative line graphs of the CWT signal at scale 5 for neck 3D acc. over time.

Computed std of the signal is marked on each graph. **h, i** Plots of the CWT signal at scale 5 for neck 3D acc. and power spectrum density (PSD) bandwidth of neck 3D acc. signal at 10 wk. **j** Representative line graphs of tail 3D angles over time. Computed autocorrelation (auto.r) of the signal is marked on each graph. **k, l** Plots of the tail 3D angle autocorrelation and zero-crossing rate (ZCR) of tip 3D velocity. Box plots show the median (black line), mean ('+' symbol), and interquartile range (box) with Tukey-style whiskers. Sample sizes (n) are indicated in plots. All statistical analyses were performed using one-way ANOVA followed by Holm−Sidak post hoc corrections for multiple comparisons. See Supplementary Table 18 for details. Schematics were created in BioRender. Heo, W. (2025) https://BioRender.com/cykyauz. Source data are provided in the Source Data file. ML machine learning, TSFEL Time Series Feature Extraction Library, Coeff. coefficient.

spectro-temporal domain analysis. We refined our motion processing by clip fragmentation with a long clip (4 s) and created 4485 features per clip (Fig. 4a, Supplementary Fig. 32). To prioritise impactful features distinguishing PD from NP, we applied a similar ML pipeline to that used in the XGB model, developing the TSFEL model. This model achieved high evaluation metrics, including 90% accuracy and its prediction on an unseen-mouse dataset yielded results that were in line with those of XGB model (Supplementary Fig. 33a–c).

Interpretation of the TSFEL model revealed that the top 20 features across the statistical, spectral and temporal domains were crucial for identifying PD (8, 8 and 4, respectively; Supplementary Figs. 33d, 34–38; Supplementary Tables 16, 17). Statistical evaluation of the top 20 features showed that all were affected in the A5 group, whilst 70% were affected in the A1 group (Fig. 4b; Supplementary Fig. 39; Supplementary Table 18). In the A1O3 group, TRE showed that 55% of the features were treated, 15% remained untreated, and 5% were treat-associated. The spectral domain, notably chest-associated features, was predominant among the treated features, suggesting that

optoRET has a significant effect on improving the spectral aspects of chest movements.

SHAP analysis identified a decrease in the wavelet entropy[53,54] of the chest 3D acceleration signal (#14) as a PD indicator (Supplementary Fig. 33d). This reduction in wavelet entropy signifies that chest movements in PD mice are more repetitive or predictable (Fig. 4c, d). Importantly, the PD groups showed a lower mean feature value with a platykurtic distribution in the KDE plots (Fig. 4e; Supplementary Fig. 35 [#14]). This indicates not only an increase in predictability but also a wider variation in chest movement amplitudes among PD mice. The scalograms of Ricker continuous wavelet transformed (CWT) chest 3D acceleration signals further revealed a pronounced shift in chest movement towards higher frequencies in the PD groups (Fig. 4c). Intriguingly, the fundamental frequency of chest movements in the PD groups increased to 5.6–6.3 Hz, nearly double that of the CT group, which was 2.8 Hz (Fig. 4d, f). This shift towards higher frequencies approaches the typical tremor frequency range observed in PD patients (4–6 Hz)[1], suggesting a potential link between chest

movement patterns and tremor genesis. However, the increase in fundamental frequency in the treated group was not statistically significant, being 3.6 Hz, suggesting a potential moderating effect of the treatment on tremor development. Collectively, our findings highlight optoRET's potential to alleviate the spectral features of PD, particularly chest movement and tremor, underscoring its promise as a targeted therapeutic approach.

## Mouse neck and tail movement complexities with optogenetics

In TRE for the A1O3 group, neck-related features accounted for 25% of the untreated features among those affected in both PD groups (Supplementary Table 18). Notably, the standard deviation of the CWT neck 3D acceleration signal at scale 5 (#3) was identified as the most impactful among the untreated features (Fig. 4g, h; Supplementary Fig. 33). SHAP analysis revealed a negative correlation between this feature's value and PD predictions, whilst its longitudinal distribution indicated diminishing values as PD severity progressed (Supplementary Figs. 33, 35 [#3]). These findings indicate that neck movement at this scale is constrained in PD mice, whereas NP mice exhibit more dynamic movement at this signal pattern (Fig. 4g). In line with this, the PD groups showed a significant decrease in the power spectrum density (PSD) bandwidth[53,55,56] of the neck 3D acceleration signal, indicating a reduced dynamic range (Fig. 4i). This trend was not observed in the A1O3 group, suggesting that optoRET treatment improves some aspects of neck movement dynamism, albeit limitations in certain aspects.

Among the treat-associated features (274 out of a total of 4485 features), tail-related features were prominently represented, accounting for 27% of these features (Supplementary Table 18). The tail angle autocorrelation[53] (#2) was identified as the most impactful, with an increase in its value predicting PD; notably, the A5 group displayed a significantly higher mean (Fig. 4j, k; Supplementary Fig. 33). This suggests less varied movements, indicative of reduced motor control diversity as PD advances. Additionally, a decrease in the zero-crossing rate (ZCR) of tip 3D velocity was observed in PD mice, indicating fewer directional changes and a simplified tail movement pattern (Fig. 4l). Intriguingly, despite the challenges, the treated group showed reduced mean autocorrelation (#2), with a ZCR similar to that of the CT group (Fig. 4k, l). This suggests that optoRET treatment not only alleviates some PD-related tail movement impairments but also introduces a unique, treat-associated behavioural pattern.

## Optogenetics improves turning and rearing in PD mice

In our prior analysis, we detailed various behavioural features associated with PD phenotypes and the therapeutic impact of optoRET. To enhance our understanding and further validate these findings, we incorporated more intuitive and generalised features within categorised behaviours. Initially, we annotated turning, rearing and walking events across 161 long clips (120 frames) from the CT, A5, A1 and A1O3 groups (14, 11, 10 and 4 mice, respectively). This approach yielded a curated dataset from which we engineered 3, 11 and 28 features for the respective events. The core metrics for the turning and gait features and representative clips are highlighted in Fig. 5a, b.

Notably, we found that turning and rearing behaviours were significantly affected in the PD groups (Supplementary Figs. 40, 41 and Supplementary Tables 19, 20). The A5 group showed increased turning duration, whilst both the A5 and A1 groups exhibited reduced turning velocity (Fig. 5c, d). However, this reduction was not significant in the A1O3 group compared to the CT group (Fig. 5d). In terms of rearing behaviour, the PD groups showed greater variability in chest vertical movement velocity throughout the rearing events (Fig. 5e). Specifically, the A5 group exhibited a marked reduction in peak velocity during the rearing transition phases (Fig. 5f), corroborating our observations above (Fig. 2a, #16).

## Optogenetics enhances gait in PD mice

Given the diversity of gait metrics and the relatively extensive array of gait features, we prioritised these features by applying AI techniques as above. The Extra Trees Classifier (ET) with 20 selected features emerged as the best-performing model, achieving an 80% accuracy rate in the classification of the non-PD (NP) and PD gaits from the validation dataset (Supplementary Fig. 42a, b). This model evaluates individual steps, computing the ratio of PD-characteristic steps to total steps as a PD gait score (%) for each mouse. The predicted PD gait scores for the CT, A5, A1 and A1O3 groups from an unseen-mouse dataset (n = 3, 3, 10 and 4 mice, respectively) aligned with the PD scoring results (Fig. 5g).

Analysis of the top 20 gait features showed that most mean values were significantly higher in the PD groups except for leg angle anticorrelations (Fig. 5h, Supplementary Figs. 42–44; Supplementary Tables 21, 22). In TRE for the A1O3 group, 60% of the features were marked as treated, whilst 15% remained untreated (Fig. 5i). Notably, untreated features were primarily associated with the spatial domain, especially the postural components, including feet distance (#3) and stance width (#8) (Fig. 5h). Nonetheless, distribution analysis indicated that the treatment still had a preventative effect on abnormally high feet distances, observed in both PD groups (Supplementary Fig. 43).

Upon reviewing raw video recordings of walking events corresponding to high PD gait scores, we observed increased ranges and variability in leg angles, along with instances of feet dragging and, at times, trailing without toe-off, indicative of severely impaired gait (freezing gait; Supplementary Fig. 45 and Supplementary Movies 3, 6). In these instances, we noted a significant separation between the feet and anus positions, prompting us to create heatmaps for visualising foot placements relative to the anus in the CT, A5, A1 and A1O3 groups (50, 51, 27 and 27 clips, respectively; Fig. 5j). Quantification of foot placements in the heatmaps' left quartiles (negative x-axis, denoting feet behind the anus) highlighted a marked increase in feet lagging in the PD groups, a deviation not seen in the treated group (Fig. 5k, l). Collectively, our findings indicate that optoRET significantly improved gait in PD, particularly by preventing foot trailing, which is the PD-specific gait phenotype observed in our study.

## Discussion

In this study, we established AI-driven PD diagnostics and an optogenetic approach as a therapeutic strategy. The XGB model provided labour-efficient yet accurate PD diagnostics, achieving 90% accuracy in distinguishing PD from non-PD mice. AI-predicted PD score (APS) demonstrated higher sensitivity in discriminating PD severity cohorts (p < 0.0001, at 10 wk) and enabled earlier detection (at 2 wk) with both A1 (p = 0.0270) and A5 (p < 0.0001) groups, outperforming traditional behavioural assessments such as rotarod and beam walking test scores (RRS and BWS, respectively). These findings suggest that our method may serve as a valuable tool for drug discovery targeting early stages or varying PD severity levels, complementing conventional assessment methods[41]. Furthermore, APS showed strong correlations with established neuropathological hallmarks, including striatal TH intensity and nigral DA cell counts (|r| = 0.80 and 0.85, respectively), highlighting its value as a behavioural proxy for PD pathology. This innovative approach also allowed comprehensive insight into the PD behaviour phenotypes as PD progresses, facilitating the therapeutic evaluation of treatments.

Through model interpretation, our AI-based analysis elucidated crucial behavioural features of PD, with disrupted limb coordination, especially interlimb asymmetry, standing out as a primary determinant. Our comparative analysis with a motor neuron disease (MND) supported that our findings were specific to PD pathology and not simply a general indicator of motor dysfunction. This insight is crucial for treatment response evaluation [TRE]), which, as demonstrated, our optogenetic interventions mitigate PD progression by particularly

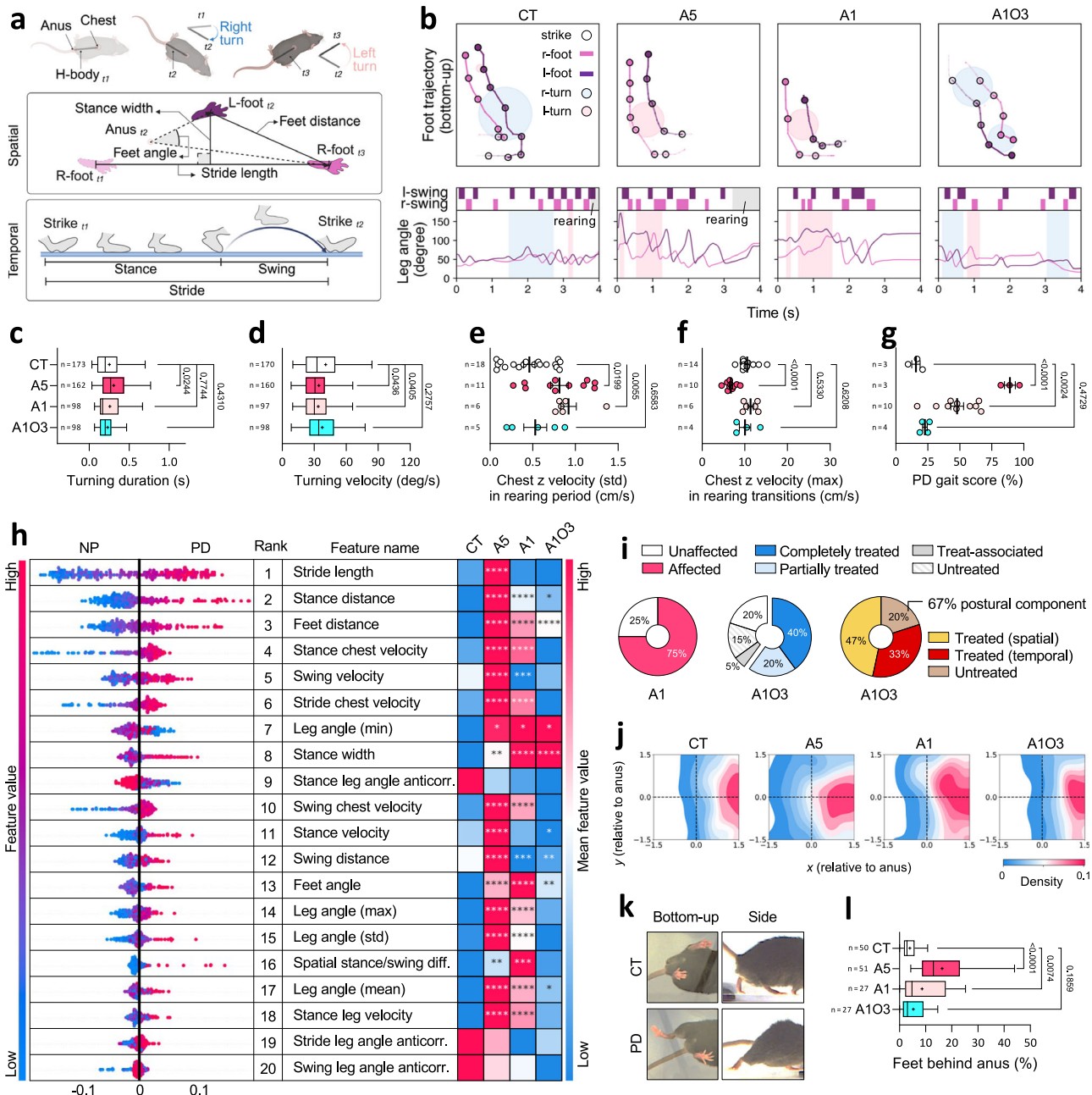

**Fig. 5 | optoRET enhancement in locomotion behaviour and prevention of foot trailing, the PD gait signature. a** Schematic of the key feature components, particularly for turning and gait events. **b** Representative foot trajectories and leg angle line graphs over time, alongside the marked behaviour events including turning, rearing and swing phases in gait. **c,d** Plots of turning duration and velocity. **e,f** Plots of chest vertical velocity standard deviations (std) and maximum (max) in rearing period and transition phases, respectively. **g** Plots of Gait model predicted PD gait scores for the CT, A5, A1 and A1O3 groups. **h** Graphic summary of the top 20 features of Gait model. The beeswarm plots (SHAP summary plot) offer local explanations of each feature and the heatmaps details mean feature values, with statistical significances indicated by asterisks. **i** Pie charts, detailing the evaluations of PD-affected features, treatment responses, and untreated or treated feature domains. **j** Heatmaps of feet 2D position densities, relative to anus for CT, A5, A1

and A1O3 groups (50, 51, 27 and 27 clips, respectively). Dashed lines indicate the $x$- and $y$-axis intersection at 0. **k** Representative snapshot images of CT and PD mice, highlighting the foot trailing behind the anus for the latter. **l** Plots of the percentage of feet positions behind the anus, depicted in the negative $x$-axis of heatmaps. Box plots show the median (black line), mean ('+' symbol), and interquartile range (box) with Tukey-style whiskers. For dot plots, data are presented as mean ± SEM. Samples sizes are indicated on each plot. See Supplementary Tables 21, 22 for definitions of listed features and statistical details of heatmaps in panel h, respectively. All statistical analyses were performed using one-way ANOVA, with Benjamini–Hochberg post hoc correction in panel c,d and Holm–Sidak correction in all other panels for multiple comparisons. Schematics were created in BioRender. Heo, W. (2025) https://BioRender.com/cykyauz. Source data are provided in the Source Data file.

enhancing limb coordination and locomotion and reducing tremor. Despite the efficacy of optoRET in alleviating PD symptoms, it had a limited impact on neck rigidity, an area marked as untreated. Intriguingly, the treatment introduced unique tail movement patterns, an area marked as treat-associated, revealing the emergence of treat-

induced behaviours. The value of TRE was further underscored in the comparative analysis with other treatments, including L-DOPA and a RET agonist (Supplementary Fig. 31). These insights highlight the importance of thorough behavioural analysis in both understanding PD and evaluating therapies, not only revealing the overall therapeutic

benefits but also delineating key treated and untreated areas, as well as the emergence of divergent, treatment-induced behaviours.

Whilst optogenetic interventions have shown promise, particularly in less severe cases, further exploration is warranted to optimise treatment variables for enhanced efficacy. Our findings indicate that different schedules of light stimulation to activate optoRET yield different therapeutic outcomes. The significant beneficial effects of optoRET were only noticeable under daily stimulation given periodically (biweekly and alternate days). This finding might emphasise that discontinuous but repetitive activation of RET signalling is a more effective therapeutic approach, as implicated by previous research[22,23]. Similarly, our results with L-DOPA and BT44 underscore the importance of tailoring medication protocols to the mechanism of action and therapeutic strategy[1,50,57]. Collectively, these observations support the notion that refining stimulation protocols – whether pharmacological or optogenetic – is essential to maximise therapeutic outcomes. Such optimisation may offer critical insights for guiding delivery schedules in ongoing and future clinical trials targeting the RET signalling.

Further investigation into the underlying beneficial mechanisms of optoRET intervention will also be crucial. Although optoRET has neuroprotective effects, such as preserving nigral DA neurons and striatal fibres, the specific mechanisms are yet to be understood. Our prior in vitro work suggested that optoRET may enhance axonal regeneration, offering a potential mechanism by which it could treat PD[21]. Future in vivo studies could employ whole-brain 3D imaging techniques to assess how optoRET alters DA neuron projections and their striatal terminals in PD. We could also use 2-P imaging and SynapShot[58], a reversible synapse visualisation tool in vivo, to track treatment-induced changes in striatal microcircuits, offering deeper insights into the mechanisms and behavioural effects of optoRET. The basal effect of optoRET may also warrant further investigation, particularly in relation to the protein overload lesion (POL), as the proportion of severe PD mice in the A5OD group was reduced to approximately half that observed in the A5 group. Collectively, our approaches with AI-driven diagnostics and optogenetic intervention hold the potential to advance our understanding of PD and RET signalling.

Although direct translation from rodents to humans presents challenges, the similarities in limb kinematics, interlimb coordination, rhythmicity and tremors between PD mice and patients lay a foundation for exploring potential targeted therapies[59]. Our therapeutic evaluation approach holds promise for developing personalised treatments tailored to specific PD symptoms. However, it is important to acknowledge our reliance on a single PD mouse model warrants caution. Notably, the historically limited success in identifying robust PD animal models[2] may partially stem from the low sensitivity and resolution of human-observer-based assessments, an issue potentially overcome by our AI-based high-resolution feature analysis. Future research should include various PD mouse models that recapitulate wider aspects of human PD. A universally applicable AI model that captures the common traits of PD could greatly improve diagnostic and treatment assessments. To this end, we integrated the AVATAR system into our analytical pipeline to enable robust 3D behavioural tracking and interpretable phenotyping.

We combined the AVATAR system with binary classification models and explainable AI to identify behavioural features associated with PD. The AVATAR system employs five calibrated cameras for multi-view triangulation and a pre-trained YOLO-Darknet backbone to enable out-of-the-box 3D keypoint tracking with high spatial and temporal resolution. This approach takes a different form from widely used 2D pose estimation pipelines such as DeepLabCut, LEAP, and SLEAP, which typically require additional workflows for 3D inference, and from keypoint-free, depth-based frameworks such as MoSeq, which analyse whole-body posture in 3D voxel space[60–62]. While these tools are widely adopted and highly versatile, AVATAR provides behavioural phenotyping without the need for manual 3D reconstruction or retraining. Rather than replacing existing tools, our method is intended to complement them by integrating 3D pose estimation with downstream classification and interpretability analysis. Although this study focuses on PD, the modular structure of the framework may be applicable to other disease models – such as autism, epilepsy, or Huntington's disease – where subtle behavioural changes are of interest. Direct benchmarking against public datasets was not performed, as AVATAR operates within a specialised multi-camera environment that differs structurally from standard 2D or depth-based video datasets. To facilitate future comparisons and community use, we have deposited an example AVATAR dataset – including a multi-view video and its corresponding 3D reconstructed keypoint data – in a public repository (see Data Availability). This dataset may serve as a reference for future benchmarking efforts and cross-platform integration.

In conclusion, this study represents a substantial leap forward in PD research, illustrating the potential of AI to revolutionise our approach to diagnosing and managing neurodegenerative diseases. By integrating AI with optogenetic interventions, we not only deepen our understanding of PD but also pave the way for more effective treatment strategies. The ongoing refinement of our AI diagnostic model, coupled with its integration into therapeutic evaluations, holds great promise for transforming PD management and ultimately improving the quality of life of those affected by this debilitating disease.

## Methods

### Ethics statement
Animal experiments and treatments were conducted in accordance with the guidelines of the Institutional Animal Care and Use Committees (IACUC) at KAIST under approved protocols KA2020-112 and KA2024-007.

### Cloning and AAV production
Human alpha-synuclein (*SNCA* gene) with a point mutation (A53T) was amplified by polymerase chain reaction (PCR) from the EGFP-alphasynuclein-A53T (Addgene #40823) with primers, 5'- TCT AGA CCG CCA CCa tgg atg tat tca tga aag gac ttt caa agg (forward) and 5' AAG CTT ttc tta ggc ttc agg ttc gta gtc ttg ata c (reverse), and inserted into the pAAV2-ITR-hSyn1 transgene vector using the *XbaI/HindIII* restriction enzyme sites. The pAAV2-ITR-hSyn1-hA53T, pAAV2-ITR-hSyn1-DIO-mScarlet, pAAV2-ITR-hSyn1-mScarlet and pAAV2-ITR-hSyn1-DIO-optoRET were used for the AAV production, using a triple transfection system with a packaging vector (pRC-DJ/8) and a helper vector[21].

### Animals, surgery and treatments
**Mouse lines and genders.** In this study, three distinct B6J mouse lines were used: a C57BL/6 J wild-type (WT) and two transgenic lines maintained as B6.SJL × C57BL/6J hybrids. The transgenic lines include DAT-CRE and SOD1-G93A mice, both obtained from The Jackson Laboratory (strain #006302 and #002726, respectively).

Sex was considered in the study design. Male mice were used in all groups except for the motor neuron disease (MND) cohorts, which included female wild-type (NALS group) and female SOD1-G93A (ALS group) mice. No sex-based statistical comparisons were performed, as sex was not a variable of primary interest in this study. Detailed information regarding mouse line, sex, and group assignment is provided in Supplementary Table 1.

**Housing and maintenance of mice.** All mice were group-housed under a 12-h light/12-h dark cycle at room temperature with free access to food and water. All mice underwent handling and baseline behavioural tests at 20 wk of age, followed by stereotaxic surgery as

appropriate. For the MND cohorts, experiments commenced at 10 wk of age. If housing conditions or experimental timelines differed for specific cohorts, those details are provided where relevant. After the surgery, each home cage housed two mice, separated by a transparent divider. Mice non-subjected to surgery were also housed in the same way at the same time point. Additionally, to prevent mortality in hA53T-injected mice, a disposable 50-mm dish containing supplementary dietary gel (ClearH2O, Cat# 72-06-5022) and moistened food pellets was magnetically secured to the cage floor and replenished every other day.

**Anaesthesia and euthanasia conditions.** For all surgical procedures, mice were anaesthetised via intraperitoneal injection of Avertin (200 mg/kg; 2,2,2-tribromoethanol in PBS; Sigma, Cat# T48402), and body temperature was maintained using a heating pad set to 37 °C throughout the procedure. For immunohistochemistry (IHC), mice were deeply anaesthetised with a double dose of Avertin and euthanised via transcardial perfusion with PBS followed by 4% paraformaldehyde (PFA). For terminal procedures not requiring tissue preparation, mice were euthanised using carbon dioxide inhalation in accordance with institutional animal care guidelines.

**Stereotaxic surgery for virus injection.** Using the stereotaxic device, the SNc was targeted with the following coordinates (relative to bregma): anterior-posterior (AP) = −3.4 mm; medial-lateral (ML) = +1.3 mm for unilateral lesions or ± 1.3 mm for bilateral lesions; and dorsal-ventral (DV) = +4.3 mm. For each injection, a total volume of 1 µL of AAV diluted in DPBS was injected into the SNc. For the EV group, WT mice received AAV-DJ/8-hSyn1-DIO-RFP at a titre of 1E + 11 gp/mL. For the R1 and R5 groups, control viruses were used at titres of 1E + 12 gp/mL and 5E + 12 gp/mL, respectively – WT mice were injected with AAV-DJ/8-hSyn1-RFP, whilst DAT-CRE mice received AAV-DJ/8-hSyn1-DIO-RFP. For the A1 and A5 mice, AAV-DJ/8-hSyn1-hA53T was injected at titres of 1E + 12 gp/mL and 5E + 12 gp/mL, respectively, along with a control virus (1E + 11 gp/mL of RFP for WT mice or DIO-RFP for DAT-CRE mice). For the optogenetic groups, the control virus (RFP-DIO) was replaced by AAV-DJ/8-hSyn1-DIO-optoRET at 1E + 11 gp/mL. Detailed information is summarised in Supplementary Table 1.

**Direct brain infusion of a RET agonist at a constant flow rate.** For the RET agonist cohort (A1BT), mice underwent a second surgery 1 wk after the initial stereotaxic virus injection at 20 wk of age. During this procedure, a pair of cannulas (Alzet Brain infusion kit no.2, Durect, USA) was implanted bilaterally into the striatum using the following coordinates: AP = 0.7, ML = 1.8, DV = −3 mm. Two osmotic pumps (Alzet model 2006, Durect, USA; Lot# 10426-21; mean pumping rate 0.18 µL/hr), connected to the cannulas via catheters, were then implanted into a subcutaneous pocket in the midscapular region. For immediate chemical infusion into the brain, the pre-assembled cannula-catheter-osmotic pump system was filled with BT44 (120 µM, ProbeChem, Cat# PC-72010) dissolved in propylene glycol (Sigma, Cat# P4347) and incubated in saline at 37 °C overnight prior to surgery. The cannulas were custom-cut to a reduced base diameter to facilitate bilateral mounting on the mouse brain. Due to the limited capacity of the osmotic pumps, the experimental endpoint for this group was set at 6 wk post-virus injection. Because the cannula insertion and osmotic pump implantation surgery is unique to the RET agonist (A1BT) cohort, we consistently refer to the initial stereotaxic virus injection as 'post-surgery' throughout the manuscript.

**Transcranial photoactivation with LED home cage lids.** For the photoactivation of the optogenetic light groups (A5O1, A5O2, A1O1, A1O2, A1O3), we utilised a custom-made cage lid fitted with an array of blue LEDs (470 nm). Seven LED control devices were connected with a Bluetooth Low Energy (BLE) circuit board and operated by a nearby

minicomputer (e.g., Raspberry Pi). This minicomputer receives commands from the internet-based WNBN[45] server according to a pre-programmed schedule and subsequently transmits these commands to the LED control devices either simultaneously or selectively at the designated times automatically via BLE communication.

We programmed a daily activation cycle lasting 5 h, from 1 PM to 6 PM. The long-term photoactivation began 1 wk post-surgery and continued until the experimental endpoint at 10 wk, with the daily-based LED toggle signals, programmed at three distinct schedules: daily (S#1), biweekly (S#2), and alternate days (S#3). When the system was activated, the six 60-mm fans on the outer surface of the lid were controlled simultaneously to maintain the inner cage temperature at room temperature, despite an additional noise increase of 18 dB. When the LEDs were on, the light density was measured at 40 µW/cm². Noise, temperature, and light density were all measured at the inner centre base of the cage with the lid closed.

**Intraperitoneal administration of L-DOPA.** For the L-DOPA cohorts (A1L1 and A1L3), L-DOPA (3,4-Dihydroxy-L-phenylalanine, Sigma, Cat# D9628) and Benserazide (BZ, Sigma, Cat#BZ283) were administered via intraperitoneal injection. The compounds were dissolved in normal saline (Quality Biological, Cat# 114-055-101) at working concentrations of 3.3 mg/mL for L-DOPA and 1 mg/mL for BZ, with 1.5 mM L-ascorbic acid (Sigma, Cat# A7631) as an antioxidant. The solutions were stored at −20 °C, protected from light, and used within 1 wk of preparation. Each mouse was weighed and injected with L-DOPA (50 mg/kg)[46] and BZ (15 mg/kg) in an injection volume of approximately 400–600 µL, using an insulin syringe. These doses were adapted from a previous study[46]. The A1L1 group received daily injections, whilst the A1L3 group was injected on alternate days from 1 to 10 weeks post-surgery.

## IHC analysis

Mice were anaesthetised with Avertin, followed by transcardial perfusion with PBS and 4% PFA. For photoactivated mice, brains were fixed in PFA after light schedule completion. Fixed brains were coronally sectioned at 50 µm thickness using a Leica vibratome. Sections were then blocked in PBS containing 5% normal donkey serum (Abcam, Cat# ab7475) and 0.3% Triton X-100 for 1.5 h at room temperature (RT).

For immunostaining, each primary antibody's specificity was confirmed by the supplier's datasheet. Brain sections were incubated overnight at 4 °C with primary antibodies (1:1000), sheep anti-TH antibody (Abcam, Cat# ab113), rabbit anti-aSyn (Abcam, Cat# ab52168), or rabbit anti-phospho(S129)-aSyn (Abcam, Cat# ab51253). Following three PBS washes, sections were incubated for 1 h at RT with fluorescent protein conjugated secondary antibodies. Finally, sections were mounted with a DAPI-containing solution and imaged using a microscope and analysed using ImageJ software.

For quantification of striatal TH density, an average of four regions per mouse was analysed from two sections (approximately at bregma +0.62 and +0.02 mm). For each striatum, the mean intensity from 10 subregions was normalised to the mean intensity of the three densest TH-stained subregions in the section. For analysis of nigral TH cell counts, an average of four sections from the SNc (approximately bregma −3.16 to −3.40 mm) per mouse was analysed. In both cases, the values were normalised to the respective group mean values obtained from the EV group.

## Behaviour tests

Mice were acclimated by handling for 5 min per day for three consecutive days within the week before being subjected to any behavioural test training or tests.

**Elevated beam walking test (BWT).** Mice were trained for three consecutive days, a week prior to the stereotaxic surgery. On training days 1–2, a beam (100 cm length and 2.5 cm width) was horizontally

placed on the supporting bars (42 cm height), located at each end. Mice were placed pointing toward the goal, a semi-closed black box. Each mouse had to balance on the beam and was required to reach the goal box three times on each day. On training day 3, the goal box position was elevated to incline the beam at 20 degrees, and mice were required to reach the goal box three times. On the testing days (biweekly), mice performed three trials with at least 10-min breaks between trials, whilst behaviour was recorded using dual FHD cameras located laterally and above the beam. Traverse latency was defined as the time required for a mouse to cross from the start to the end line (80 cm) on the beam. The mean of the two best-performed trials (i.e., those with the shortest traverse latency) was used for analysis. The BWT score (BWS) was calculated as follows:

$$BWS(\%baseline) = \frac{\text{Mean latency of two best trials at test week}\,(x\,\text{wk})}{\text{Mean latency of two best trials at baseline}\,(0\,\text{wk})} \times 100$$

(1)

The exclusion criteria explicitly exclude data from: (1) mice unable to reach the endpoint of the beam in three trials (failure type I); (2) mice with BWS above 600% (failure type II); and (3) mice unable to perform due to mortality. Trials classified as failures (types I and II) were excluded from further analysis.

**Open-field test (OFT).** On the testing days (biweekly), one mouse at a time was placed in the AVATAR studio[13], a transparent chamber ($20 \times 20 \times 30$ cm) equipped with five high-speed cameras (FLIR, U3-23S3M/CC, $1200 \times 1200$ pixels, sampling rate of 30 Hz) for the OFT. The freely moving behaviour in the AVATAR studio was recorded for 20 min.

**Tail suspension test (TST).** At the endpoint (10 wk), one mouse at a time was placed in the AVATAR studio and behaviour was recorded as in the OFT for 5 min. By reviewing the TST recordings, the incidence of each hindlimb clasping[6,63,64] was evaluated by human observers according to a 4-point scale:

Class 0: Extended hindlimbs (no clasping).
Class 1 ($c_1$): One hindlimb is held by the forelimbs.
Class 2 ($c_2$): Both hindlimbs exhibit clasping.
Class 3 ($c_3$): All limbs exhibit clasping.

For each mouse, if fewer than 2 incidences (each lasting over 2 s) were observed for a given class, that class was scored as 0; if 2 or more incidences were observed, the class was scored as 1. Based on these scores, the TST score (TSS) was calculated as follows:

$$TSS = (c_1 \cdot 1) + (c_2 \cdot 2) + (c_3 \cdot 3)$$

(2)

**Rotarod test (RRT).** Mice were trained to walk on the five-lane rotarod (B.S Technolab Inc., Korea) for two consecutive days, a week prior to the stereotaxic surgery. On training day 1, five animals at a time were placed on each lane (separated by panels) and the rod was rotated at a constant speed of 8 rpm for 5 min for three trials with at least 25-min breaks between trials. On training day 2, the rod was rotated at a constant speed of 12 rpm for 5 min, and on the testing day, the rod was operated in accelerating mode from 5 to 35 rpm for 5 min. The mean of the two best-performed trials (longest latency to fall) was used. The RRT score (RRS) was calculated as follows:

$$RRS(\%baseline) = \frac{\text{Mean latency of two best trials at test week}\,(x\,\text{wk})}{\text{Mean latency of two best trials at baseline}\,(0\,\text{wk})} \times 100$$

(3)

**Generation of source data with the AVATAR system**
Source datasets were generated through 3D pose estimation using the AVATAR system[13-16] (commercially available at actnova.io). The

trajectories of a freely moving mouse in the multivision recordings of the OFT in the AVATAR studio were reconstructed into time-series 3D datasets using the AVATARnet algorithm (demo version)[13]. This process detects the 9 key nodes of a mouse, including nose, neck, chest, right-hand (R-hand), left-hand (L-hand), right-foot (R-foot), left-foot (L-foot), anus and tip in 3D Euclidean space in each movie frame. Subsequently, it also provides the 3D trajectories of 8 action skeletons of a mouse, including head, forebody (F-body), hindbody (H-body), tail, right-arm (R-arm), left-arm (L-arm), right-leg (R-leg), left-leg (L-leg). The shape of a raw AVATAR data is 27 columns and ~36000 frames.

As previously reported[13], the accuracy of AVATARnet for keypoint prediction was evaluated using Mean Average Precision (mAP), Intersection over Union (IoU), and Mean Squared Error (MSE). The system achieved IoU scores exceeding 75% and an average mAP of 90% for detecting the 9 key nodes, with each class exhibiting an average MSE distance of 7–15 pixels (1.4–4.5 mm). Average Precision (AP) was defined as:

$$AP = \int_0^1 p(r)\mathrm{d}r$$

(4)

where true positives (TP, IoU > 50%), false positives (FP, IoU <50%), false negatives (FN, not detected), true negatives (TN, no object) were used to compute Precision ($p$) = TP/(TP + FP) and Recall ($r$) = TP/(TP + FN).

AVATARnet is built upon the YOLOv4[65] convolution neural network (CNN) architecture and has been modified to consist of 53 CNN layers, with an input network size ($608 \times 608$ pixels) and 56.9 M parameters, to enhance the detection of small body parts. To obtain 3D coordinates from the 2D detections, triangulation and bundle-adjustment algorithms were implemented using MATLAB's Triangulate function and Computer Vision Toolbox, following the methods of Hartley and Zisserman[66]. The triangulation process utilised 2D images from a minimum of 2 cameras, incorporating both intrinsic camera parameters (focal length, principal point, skew coefficient) and extrinsic parameters (camera location and orientation) as inputs.

**Motion processing**
From the raw AVATAR dataset, the chest 3D velocity was computed, filtering for moving motion sequences. If the instantaneous velocity exceeds a threshold (1.5 cm/s), the rolling sum for every $n$ data was calculated (window = 100 frames). When the sum is larger than $n \times 0.8$, the rolling windows were considered as Moving. For each of this we defined as a moving Episode and each of them are constituted of $n$ number of clips, which has a fixed length of frames. In our study, we generated clips with two different lengths: a short clip (60 frames, 2 s) and a long clip (120 frames, 4 s).

**Development of the XGB model**
Utilising the motion dataset comprised of short clips, we calculated the velocities for the key nodes in 1D ($z$), 2D ($xy$), and 3D ($xyz$) spaces, as well as 7 angles by employing the AVATARpy (v0.1.9)[67,68] library, in Python (v3.8). As a part of the feature engineering process, we calculated the distances between 14 pairs of nodes and the angular differences between 8 pairs of angles. For each clip, we then derived 4 statistical metrics for each feature: minimum, maximum, mean, and standard deviation. In addition, we computed a series of temporal domain parameters for each episode, resulting in a comprehensive set of 340 features per clip (Supplementary Fig. 5b).

We prepared a dataset, composed of featured motion clips from the CT and A5 group (10 and 11 mice, respectively) and balanced the clip numbers between the groups (each ~11,000). This balanced dataset was then subjected to an automated ML process using PyCaret (v2.3.10)[39]. The input dataset was split into training and validation datasets at a ratio of 7:3 (15277 and 6893 clips, respectively). Initially,

the framework was configured to select features by assessing importance weights, removing those with low variance and high multicollinearity, in preparation for training various classifiers. The performance of the classifiers was compared through evaluating metrics, including accuracy, precision, recall, F1, Kappa, Matthews Correlation Coefficient (MCC) by cross validation. Utilising TreeExplainer, we identified and assessed the 30 most impactful features across 4 tree-based models – XGB, ET, Light Gradient Boosting Machine (LightGBM) and CatBoost – to determine the 30 optimal features. We re-set the PyCaret framework to use the optimal features to train and compare the classifiers. As the XGB model was the best-performing model with all features or the optimal features with ~90% accuracy, we finalised our PD diagnostic model as the XGB model trained with the optimal features. The XGB model was used to classify each clip as either non-PD or PD, and the proportion of the identified PD clips over the total clips for each mouse was then calculated, resulting in a PD score expressed as a percentage. We further validated the model by testing it on a dataset, comprised of clips from the leftover unseen mice in the CT and A5 group (4 and 6 mice, respectively).

### Distribution and feature value analysis

**Longitudinal intra-group distribution analysis using Kullback-Leibler (KL) divergence.** For each of the CT, A5, NALS, ALS groups, longitudinal variations of the feature distributions were analysed using KL divergence. Each group had six sampled datasets of 150 clips randomly selected per week. Specifically, for each (sampled dataset, feature) combination, we first applied kernel density estimation (KDE) to the baseline (0 wk) and subsequent time points (2, 4, 6, 8 and 10 wk). KDE was performed using Seaborn's kdeplot function in Python, providing a smooth approximation of the underlying distribution for each time point. We then computed the KL divergence $D_{KL}(p||q)$ between the KDE curves at the baseline and subsequent weeks. The KDE estimates for each week were interpolated onto a common grid derived from the baseline (0 wk) KDE and the integration was performed numerically using the grid spacing. Divergence was computed only when both density estimates were available, providing a quantitative measure of how much the distribution at a given time point diverges from its own baseline. The KL divergence was defined as:

$$D_{KL}(p||q) = \int p(x) \ln \frac{p(x)}{q(x)} \, dx \tag{5}$$

where $p(x)$ and $q(x)$ are the estimated densities at the two different time points, numerically approximated by evaluating the integral at the same set of $x$-coordinates obtained from the KDE results.

**Longitudinal inter-group comparisons between control and disease model.** For comparisons between the CT and A5 groups, and between the NALS and ALS groups, for each feature at each week, the computed intra-group longitudinal divergence trajectories were compared. Additionally, for each feature, we tracked mean feature values over time, normalising them to the group's baseline to account for inter-individual and inter-group variability. Finally, significance testing was performed to assess whether the two groups differed in their distributional (KL divergence) or mean-value shifts for each feature at each week (see Supplementary Tables 3–6).

**Endpoint inter-group comparisons between PD and MND using normalised feature values.** Due to experimental limitations (e.g., differences in gender and age), direct comparisons of KL divergence between PD and MND were not feasible. Instead, a supplementary analysis was performed at the experimental endpoint (PD: 10 wk post-surgery; MND: 20 wk of age) by comparing the normalised feature values. For each feature, the values were normalised relative to the

baseline mean for each group (PD: 0 wk post-surgery; MND: 10 wk of age).

The analysis compared the control and experimental groups within each dataset (PD: CT vs A5; MND: NALS vs ALS) using data previously computed in Supplementary Tables 4, 6 (which provide the number of clips, normalised mean, SEM and Bonferroni-corrected $p$-values). For each feature, the mean difference was computed as:

$$\text{Mean difference} = \text{Norm.mean of exp} - \text{Norm.mean of ndc} \tag{6}$$

where exp denotes the experimental group and ndc denotes the non-disease control group. The SEM difference was calculated by error propagation as:

$$\text{SEM difference} = \sqrt{(\text{SEM}_{exp})^2 + (\text{SEM}_{ndc})^2} \tag{7}$$

The resulting statistics for each feature are summarised in Supplementary Table 7.

**Analysis of feature distributions between CT, PD and A1O3 groups.** For the comparative distribution analysis, KL divergence was computed for each feature by comparing the variation of PD and A1O3 groups against that of the CT group at 10 wk. Each group comprised 6 sampling sets, with each set containing 150 randomly selected clips. A one-way ANOVA was performed across the groups, followed by pairwise comparisons using t-tests against the reference groups (CT or A1). The resulting $p$-values from pairwise comparisons were corrected for multiple comparisons using the Holm−Sidak method. The resulting statistics for each feature are summarised in Supplementary Table 9.

**Comparative analysis of feature values for treatment conditions at endpoint.** For each treatment group (optogenetic: A1O2, A1O3; L-DOPA: A1L1, A1L3; RET agonist: A1BT), the CT and PD groups were compared independently against the respective treatment groups. For each feature measured at the study endpoints (10 wk for all groups, except 6 wk for A1BT), we computed the group mean, standard error of the mean (SEM) and $p$-values. $P$-values were adjusted for multiple comparisons using the Holm−Sidak method (Supplementary Tables 10–14).

The computed group means and corrected significance marks were aggregated, and min-max normalisation was applied to the raw mean values for each feature to facilitate comparisons across features. Specifically, for a given feature, if $x$ denotes a raw mean and the $\min(x)$ and $\max(x)$ denote the minimum and maximum values observed across groups, the normalised value (norm_value) was calculated as:

$$\text{norm\_value} = \frac{x - \min(x)}{\max(x) - \min(x)} \tag{8}$$

This procedure scales all values to the $[0, 1]$ interval. For feature such as motion interval and duration−which exhibited a wide range of raw values−a logarithmic transformation was applied to stabilise variance prior to normalisation. A small constant (1e-6) was added to avoid the logarithm of zero. The log-transformed values were then normalised using the same min-max approach, yielding norm_log:

$$\text{norm\_log} = \frac{\ln(x + 1e-6) - \min(\ln(x + 1e-6))}{\max(\ln(x + 1e-6)) - \min(x + 1e-6)} \tag{9}$$

These normalised metrics are summarised in Supplementary Table 15 and visualised as a heatmap in Supplementary Fig. 31a, facilitating direct comparison of feature differences across treatment groups.

## Axial bending angle analysis

A study by Andreoli et al.[50] introduced an automated measurement of axial bending angle, which showed a strong inverse correlation with trunk/neck dystonia scores ($r = -0.956$)[50]. Building on this approach, we computed axial bending angles from 2D pose coordinates (nose, anus and hindlimbs) using the pose dataset. For each frame, vectors were constructed from the hindlimb midpoint (**midHL**) to the nose ($N$) and to the anus ($A$) and the angle ($\theta$) between them was calculated as:

$$\theta = \cos^{-1}\left(\frac{(N - \mathbf{midHL}) \cdot (A - \mathbf{midHL})}{\|N - \mathbf{midHL}\| \; \|A - \mathbf{midHL}\|}\right) \cdot \left(\frac{180}{\pi}\right) \quad (10)$$

A binary flag was assigned to each frame (1 if $\theta < 75°$, else 0). This signal was smoothed using a 60-frame rolling window. Bending episodes were defined as periods where $\geq 70\%$ of frames in the window met the threshold. Each episode was extended 60 frames backward to capture transitional dynamics and segmented into fixed-length clips (60 frames each), in which the mean axial bending angle was computed.

## Evaluation of PD-affected features and treatment response evaluation (TRE)

**(1) PD-affected feature analysis.** This phase involved comparing PD groups (A1 and A5) against the CT group at a 10-wk mark. Welch's t-test was used to assess statistical significance, with $p$-values corrected for multiple comparisons using the Holm–Sidak method. Features that did not significantly differ from the CT group were labelled as Unaffected, whereas those that did were categorised as Affected. This delineation was crucial in identifying features that are potentially implicated in PD.

**(2) TRE analysis.** In this subsequent phase, our focus shifted to evaluating the impact of treatment. Treatment outcomes were compared against both the A1 and CT groups at the endpoint mark using Welch's t-test (with Holm–Sidak correction) to assess the significance of feature changes across groups.

**Unaffected features.** Features classified under this category remained consistent in the A1 group and showed no significant alterations in the treatment group, suggesting that neither PD nor the treatment influenced these specific features.

**Completely treated features.** Features falling into this category were affected in the A1 group but exhibited no significant differences in the treatment group, indicating an alignment to normalcy and highlighting successful treatment outcomes.

**Partially treated features.** This category included features where treatment led to significant improvements, albeit not fully aligning to the levels observed in the CT group. It suggests a measurable positive shift from the PD state, yet not a complete resolution.

**Treat-associated features.** Features classified here demonstrated significant changes due to treatment but do not align directly with the trajectory toward the CT group's level. This category encompasses both unexpected divergent response that do not match CT or untreated PD conditions.

**Untreated features.** Features that remained significantly altered in the treatment group, similar to their state in the A1 group, were deemed untreated. This signifies lack of observable treatment effect to mitigate or alter the PD-affected state of these features towards an improved or normalised condition.

## Development of TSFEL model and spectro-temporal analysis

For our spectro-temporal analysis, we utilised the motion dataset comprised of long clips. To generate the featured datasets, we selected core features (3D) identified by the XGB model and expanded them to a total of 4485 features using the TSFEL (v0.1.3)[53] feature extraction package in Python (Supplementary Fig. 32). Applying a similar ML pipeline to that used in the development of the XGB model, we prepared and balanced the featured dataset from the CT and A5 groups. Utilising PyCaret[39] framework, total 332 of the features were selected and the balanced dataset was split into the training and validation at a ratio of 7:3 (7429 and 3352 clips, respectively). The XGB model emerged as the best performer with the evaluation metrics by cross validation. For clarity, we refer to this model as the TSFEL model in our manuscript. We further validated this model by testing it on the left-over dataset, as done above.

Whilst our spectro-temporal analysis primarily focuses on the top 20 features highlighted by the TSFEL model, our investigation extends beyond these to include relevant supporting features. Once finding the key aspects of highly ranked features, we examined relevant features that could support former features. The list of TSFEL extracted features can be found in the Supplementary Fig. 32 and the detailed descriptions and their computational methods can be found in the TSFEL GitHub repository[53]. Here, we elaborate on the TSFEL features highlighted in the main results.

**Wavelet entropy of chest 3D acceleration signal.** Wavelet entropy of a signal quantifies the degree of disorder, complexity, or uncertainty of the signal. Higher values indicate more complex or less predictable movement patterns, whereas lower values signify movements that are more regular, repetitive, or predictable. The Shannon entropy formula[54] is employed, operating on the probability distribution of wavelet energies obtained from the continuous wavelet transform (CWT) coefficients. The initial step in the process involves convolving the signal $x(t)$ with the Ricker wavelet function, also known as the Mexican hat wavelet, across discrete scales defined by widths from 1 to 9. This convolution yields a matrix of wavelet coefficients $W(a_i, t)$, where $a_i$ corresponds to the scale factor of the $i$-th width and $t$ denotes the time component. These coefficients are essential for the ensuing wavelet entropy, w.e, computation:

$$\text{w.e} = -\sum_{i=1}^{M} d_i \cdot \log(d_i), \quad (11)$$

such that each $d_i$ is defined by the normalisation condition:

$$d_i = \frac{|W(a_i, t)|}{\sum_{j=1}^{M}|W(a_j, t)|}, \quad (12)$$

where each $d_i$ represents the normalised wavelet energy at a respective scale $a_i$, $|W(a_i, t)|$ is the absolute energy at that scale and $M$ is the total number of scales. The formed probability distribution characterises the relative energy contribution from each scale to the total energy of the signal.

**Fundamental frequency of chest 3D acceleration signal.** The fundamental frequency of a signal provides insight into the principal harmonic component underlying the signal's periodic movements. Initially, the signal, $x$, was detrended, followed by applying a Fast Fourier Transform (FFT) to the normalised signal to transition into the frequency domain:

$$(f, f\text{mag}) = \text{FFT}(x - \mu_x), \quad (13)$$

where $f$ represents the array of frequencies, $f$mag represents the corresponding magnitudes and $\mu_x$ is the mean value of the signal.

Peaks in the frequency spectrum were identified as frequencies where the magnitude exceeded 30% of the maximum magnitude:

$$\text{peaks} = \{f_i, |, f\,\text{mag}_i > 0.3 \cdot \max(f\,\text{mag}), f_i \in f\} \quad (14)$$

The fundamental frequency, $f_0$, as ascertained as the lowest frequency in the set of significant peaks, explicitly excluding the zero frequency to avoid the influence of any residual offset:

$$f_0 = \min(\{f_i \in \text{peaks}, |, f_i > 0\}), \quad (15)$$

**Wavelet standard deviation of neck 3D acceleration at scale 5.** Wavelet standard deviation of a signal quantifies the variability or dispersion of the CWT coefficients at each scale. In this context, the wavelet std at scale 5, corresponding to width 5 ($i = 5$) was computed as following:

$$\text{Wavelet std}(a_i) = \sqrt{\frac{1}{T-1}\sum_{t=1}^{T}\left(\text{W}(a_i, t) - \mu_{a_i}\right)^2}, \quad (16)$$

where $\text{W}(a_i, t)$ are the wavelet coefficients at scale $a_i$ corresponds to the scale factor of the $i$-th width and $t$ denotes the time component, $\mu_{a_i}$ is the mean of the wavelet coefficients at scale $a_i$ and $T$ is the length of the signal.

**Power spectrum density bandwidth of neck 3D acceleration signal.** The power spectrum density (PSD) bandwidth of a signal quantifies the frequency range containing 95% of the signal's power. This range represents the effective range of predominant frequencies in the signal's power spectrum. To calculate the PSD bandwidth, the signal, $x$, is first processed using Welch's method[55]. This approach involves segmenting the signal, applying a window function and then using the FFT to compute the power spectrum. If the standard deviation of the signal is not zero, the signal is normalised by its standard deviation before applying the Welch method:

$$\text{PSD}(f) = \text{Welch}\left(\frac{x}{\text{std}(x)}, f_s, \text{nperseg}\right), \quad (17)$$

where $\text{std}(x)$ is the standard deviation of the signal, $f_s$ is the sampling frequency and nperseg is the number of samples per segment, here set to the length of the signal to include the entire signal in the analysis. Subsequently, the cumulative sum of the power spectrum, $P_{\text{cum}}$, is then computed:

$$P_{\text{cum}} = \sum \text{PSD}(f), \quad (18)$$

The lower and upper frequency limits, $f_{\text{lower}}$ and $f_{\text{upper}}$, are ascertained by finding the frequencies at which the cumulative power first reaches and last exceeds 95% of the total power:

$$f_{\text{lower}} = \text{freq}\left(\min\{i, |, P_{\text{cum}}[i] \geq 0.95 \cdot P_{\text{cum}}[\text{end}]\}\right) \quad (19)$$

$$f_{\text{upper}} = \text{freq}\left(\max\{i, |, P_{\text{cum}}[\text{end} - i] \geq 0.95 \cdot P_{\text{cum}}[\text{end}]\}\right) \quad (20)$$

The PSD band width is the absolute difference between the frequency limits:

$$\text{PSD bandwidth} = |f_{\text{upper}} - f_{\text{lower}}| \quad (21)$$

**Autocorrelation of tail angle signal.** Autocorrelation of a signal quantifies the similarity between a signal and itself at zero lag. A high autocorrelation value indicates more consistency or uniformity in the signal, implying steadier or less varied movement. For the given signal,

autocorrelation is computed as follows:

$$\text{Autocorrelation} = \sum_{t=0}^{T-1} x_t \cdot x_t \quad (22)$$

where $x_t$ is the value of the signal at time index $t$ and $T$ is the length of signal.

**Zero-crossing rate of tip 3D velocity signal.** The zero-crossing rate (ZCR) of a signal quantifies the total number of times a signal changes direction across the zero-amplitude level. Despite its name, the ZCR is a misnomer in this context as it is not normalised; it is better thought of as a 'count' of the zero-crossing events. This metric captures the oscillatory nature of the signal with respect to the time axis. A higher ZCR value indicates a signal with more frequent directional changes. The ZCR of a signal, $x$, is determined by the formula:

$$\text{ZCR} = \sum_{t=1}^{T-1} 1_{\left\{|\text{sgn}(x_t) - \text{sgn}(x_{t-1})| = 2\right\}} \quad (23)$$

where $1_{\{\}}$ is the indicator function that outputs 1 if the condition is true and 0 otherwise, $\text{sgn}(\cdot)$ is the signum function that evaluates the sign of the signal at a given time index, $x_t$ and $x_{t-1}$ represent the value of the signal at consecutive time indices and $T$ is the length of signal. The total number of times the signal crosses zero reflects the instances where the signal transitions from positive to negative or negative to positive, indicative of a reversal in direction.

**Turning behaviour analysis**
For our turning behaviour analysis, we utilised the motion dataset comprised of long clips and employed a computational approach to annotate turning events by quantifying the H-body turning dynamics based on positional data of the chest and anus. Starting from a lag of two timestamps, we computed directional vectors between these positions at each timestamp, using the arctangent function to calculate a directional angle normalised within a range of −180 to 180 degrees. This normalisation quantifies the orientation change of the H-body over time, with angles representing abrupt changes (≥90 degrees) adjusted to zero for noise reduction. Subsequently, we applied a rolling mean with a window of five timestamps to smooth fluctuations and highlight significant turning behaviours. Turning events were detected and categorised into no-turning (0), left-turning (1) and right-turning (-1), based on this smoothed data. For valid turning events (left or right turning), we calculated turning features including the maximum angle reached (turning angle), the duration to this peak (turning duration), turning angle velocity and turning direction.

**Rearing behaviour analysis**
In our rearing behaviour analysis, we utilised the motion dataset with long clips and manually annotated rearing events with an emphasis on capturing significant actions that characterise rearing, including the initial hand lift-off, followed by chest's vertical ascending movements and the eventual return of the hand to the ground following the chest's vertical descending movements. We evaluated the chest's vertical displacement during the rearing by calculating the differences in the chest's vertical position at the beginning, peak and end of the event. This provided insight into the extent of upward and downward movements (transition phases) during rearing. To assess the speed and variability of these movements, we calculated the chest's vertical velocity, determining the mean, maximum and standard deviation for both the upward and downward phases of rearing. Additionally, we estimated the duration of each rearing event, providing a temporal measure of the behaviour.

### Walking behaviour analysis and development of Gait model

In our walking behaviour analysis, we utilised the motion dataset comprised of long clips and manually annotated the swing phases of each foot. Each swing phase covered a period between the toe-off and foot-strike. A stride in our study is determined by identifying consecutive foot strikes of the same foot during movement. This involves detecting transitions in the swing status of a foot, where a shift from swing to stance phase indicates a foot strike. In the analysis of gaits, the rearing behaviour is filleted out and then for each stride, we computed a series of gait features, described in Supplementary Table 21.

For the development of the Gait model, we prepared the gait featured dataset, composed of the CT and A5 group. Using PyCaret[39] framework, total 20 of the gait features were selected and the input dataset at a ratio of 6:4 was used for the ML training and validation (335 and 237 steps, respectively). The best performed model was ET model, which we referred as the Gait model in our manuscript.

### Statistics and reproducibility

No statistical method was used to pre-determine sample size. Instead, sample sizes were selected based on the number of animals necessary to enable robust statistical analysis and detect biologically meaningful effects.

Data cleaning was conducted using two pre-defined outlier detection methods to ensure analytical integrity: (1) quantile-based trimming (lower quantile = 0.01, upper quantile = 0.99) in Python, and (2) the ROUT method (Q = 5%) in GraphPad Prism (v10.5). These were incorporated as part of the analysis pipeline to minimise the influence of extreme data points.

All experiments were conducted with an average of two independent biological replicates. Some supplementary groups were not independently replicated (see Supplementary Table 1). Each experiment included appropriate control and treatment groups.

Mice were allocated to groups based on injection conditions, with an effort to balance experimental variables across groups. Mice used for model training and those retained for validation were randomly selected.

Investigators were not blinded to group allocation due to procedural constraints, including the need for appropriate handling and post-experimental care that differed between groups. Efforts were made to minimise potential bias through standardised protocols.

All statistical tests were two-sided unless otherwise specified. Statistical significance was determined using the following thresholds: $p < 0.05 = *$, $<0.01 = **$, $<0.001 = ***$ and $<0.0001 = ****$.

For the drawing of fitted line graphs, we performed raw data fitting with two different functions, using Prism software (GraphPad, v10.5): the logistic growth function and the one phase decay function.

### Schematic illustrations

Schematic illustrations in Figs. (1a, c, e, i, 2a, 3a, 4a, c, g, j, 5a) and Supplementary Figs. (1b, 6c, g, 26a, b) were created in BioRender. Heo, W. (2025) https://BioRender.com/cykyauz.

### Reporting summary

Further information on research design is available in the Nature Portfolio Reporting Summary linked to this article.

## Data availability

The datasets generated and analysed during this study have been deposited in Figshare and are publicly available at: [https://doi.org/10.6084/m9.figshare.29840969][69]. The archive includes the processed, feature-engineered datasets used for machine learning and analysis. Due to data size constraints, the full raw video recordings and complete 3D pose datasets are not included. However, a representative subset of the 3D reconstructed pose data, along with a sample multi-view video, has been made available for demonstration purposes and to support reproducibility. Raw data are available upon request from the last corresponding author (WDH). Data supporting the findings of this study are included in the main manuscript and Supplementary Information. A high-resolution version of the Supplementary Figs. file is available in the same archive. Source data are provided with this paper as a Source Data file, including the full version of the Supplementary Tables. Source data are provided with this paper.

## Code availability

Custom code used for data preprocessing, feature extraction, machine learning, data analysis, and figure generation is available in the same Figshare archive as the datasets: [https://doi.org/10.6084/m9.figshare.29840969][69]. The code can be downloaded as a ZIP file, and all code files are in the 'Codes' folder after unzipping.

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

## Acknowledgements

This work was supported by KAIST-funded Global Singularity Research Programme for 2023 and by the National Research Foundation of Korea (NRF) grant funded by the Korea government (MSIT) (no. 2020R1A2C 301474213) [W.D.H.]. This research was supported by the ASTRA Project through the National Research Foundation (NRF) funded by the Ministry of Science and ICT (No. RS-2024-00439379) [W.D.H.] and by the Bio& Medical Technology Development Programme of the NRF funded by the MSIT (No. RS-2023-00263628) [W.D.H.]. This work was supported by the National Research Foundation (NRF) funded by the Korean government (MSIT) (RS-2023-00266872, RS-2025-00521226) [D.K.] and Institute for Basic Science (IBS), Center for Cognition and Sociality (IBS-R001-D2 to C.J.L.). In addition, this research was supported by a grant of the Korea Health Technology R&D Project through the Korea Health Industry Development Institute (KHIDI), funded by the Ministry of Health & Welfare, Republic of Korea (grant number: RS-2024-00440398) [B.H.].

## Author contributions

B.H. and W.D.H. conceptualised and designed the study and wrote the manuscript. All experimental work and analysis were performed by B.H., with support from J.H.S. in AI analysis and J.H.L. in experimental work. W.K., J.K. and H.Y.L. provided critical insights for result interpretations, with J.K. also contributing programmes for analysing raw AVATAR data. D.G.K. established AVATAR system and contributed to initial data analysis. C.Y.K. and J.W.J. integrated the WNBN system with the LED cage lid devices and produced schematic illustrations. S.C. provided minor support in animal work. All the authors revised the manuscript, with additional contributions from C.J.L., K.S.K. and D.K. and W.D.H.

## Competing interests

The authors declare no competing interests.
