## [Transparent Peer Review file · Nature Communications]

Integrating artificial intelligence and optogenetics for Parkinson's disease diagnosis and therapeutics in male mice

Corresponding Author: Professor Won Do Heo

Version 0:

Reviewer comments:

Reviewer #1

(Remarks to the Author)

Hyeon, Shin, and colleagues present an impressive body of work describing the behavioral phenotype in a mouse model of Parkinson's disease (PD). Their work addresses the complexities of PD, a progressive neurodegenerative disorder with challenging motor symptoms and limited treatment options. By integrating artificial intelligence (AI) with optogenetic intervention, the authors aim to overcome the limitations of traditional task-specific behavioral assessments, which often rely on a narrow set of metrics. The study employs a sophisticated combination of freely available packages, such as AVATAR, PyCaret, SHAP, TreeExplainer, and TSFEL, to detect and describe the PD phenotype in freely moving mice. Using 3D pose estimation techniques together with explainable AI, the authors identify a wide range of behavioral markers, including gait and spectro-temporal features. While the experiments and analysis are both elegant and cutting-edge, addressing critical questions in the field, there are some drawbacks. The optogenetic experiments, in particular, require further investigation, and there are concerns regarding the statistical analyses.

Critique

Major

1. Since the time series of 3D keypoints form the basis of the PD scoring pipeline, noisy keypoints could significantly affect the extraction of relevant behavioral features (e.g., discussed in DOI:10.1101/2023.03.16.532307). Thus, it is essential to validate and report the keypoint prediction accuracy of AVATARnet on unseen test data acquired in the AVATAR studio. Additionally, testing AVATARnet on an independent dataset (e.g., https://github.com/jessedmarshall/CAPTURE_demo) is highly encouraged to ensure robustness and generalizability.
2. The split into training and validation datasets (10 normal and 11 A5X mice) should be done on a per mouse basis, not on a per clip basis, to minimize the possibility that classifiers identify subject-specific behaviors in the validation dataset. Although the authors showed that the XGB classifier trained on held-out clips can generalize to unseen mice (4 normal, 10 A1X, and 6 A5X mice), it is still possible that another model among the 18 evaluated could generalize better to unseen data (i.e., new mice). This also applies to the selection of the most relevant features (top 20 features). I suggest repeating the evaluation of the 18 different classifiers and the selection of the most relevant features using held-out mice instead of held-out clips to confirm that the XGB model performs best and the top 20 features remain the same.
3. Related to the last point (Major #2), to test whether the top 20 features remain the most relevant across datasets (i.e., test the robustness of the approach), the top 20 behavioral features should be compared between the train-val dataset (10 normal and 11 A5X mice) and the held-out dataset (4 normal and 6 A5X mice).
4. Many behavioral features relevant for PD classification (e.g., "Body 2D length") could be affected by body size differences between the groups. I suggest estimating the body size by, for example, calculating the mean size of the connected keypoints (i.e., individual parts of the skeleton) across the entire session (not just clips) for each mouse. Then, compare these sizes between the control and PD groups to exclude confounds related to group-specific body size differences.
5. Adding statistics for the changes described in Suppl. Fig. 4-6 is recommended, given they are used to identify PD-specific movement patterns such as "postural imbalances" and "a shift from restricted to more erratic and diverse movements" as PD progresses.
6. Statistics (e.g., Supplementary Table 2) lack multiple comparisons correction.
7. The optoRET treatment results are difficult to interpret (Fig. 3). For example, for A1X, there is an effect in some of the light groups and no effect in the dark group. In contrast, there is no effect in A5X groups with optoRET stimulated with light, but a difference in those mice without light, compared to the A5X group. Please comment (especially on the difference in the results between light vs. dark and A1X vs. A5X).

8. Given the results and that the light was delivered transcranially through an array of LEDs in the homecage lid, I suggest repeating some of the optoRET experiments using implanted optic fibers.
9. How different light stimulation paradigms affect the PD features should be clarified. For example, how does the "Treatment response evaluation" and "PD symptomatic evaluation" (Fig. 3d) look for A1X+S#2? Do the affected features and categories overlap with those of A1X+S#3?
10. In the abstract, results, and discussion, the authors highlight that the AI approach (XGB model) outperforms traditional task-dependent assessments. However, there seems to be no direct (statistical) comparison between the two approaches. Supplementary Figure 3c only shows the scores (model and EBWT) for two data points.
11. In the discussion section, the findings of the study should be more thoroughly integrated and compared with previously published work.

Minor

1. Unit (weeks) missing in Figure 1a and Extended Data Figure 1a.
2. Add in the legend of Fig. 1c and the methods section the underlying architecture of AVATARnet and triangulation (e.g., adapted YoloV4, triangulation, and bundle-adjustment algorithms), including their references.
3. In the paragraph discussing the t-SNE approach, the sentence "However, interpreting behavioral data has proven challenging, prompting a shift to using feature-engineered datasets instead of raw or t-SNE data." is unclear and should be rephrased. For example: "However, interpreting behavioral data in an unsupervised manner has proven challenging, prompting us to create feature-engineered datasets to train classifiers in a supervised fashion."
4. It is unclear what parameters were used for the 18 different classifiers. A table should be added or the details should be described in the method section what parameters were used for the different classifiers.
5. In Fig. 1d, it is unclear in the right plot if the skeletons are consecutive frames or frames with a temporal offset. In addition, the term "Action Skeletons" in the figure and in the text is confusing. I suggest revising it to "connected keypoints" or something similar.
6. It is unclear which dataset (train-val: 10 normal and 11 A5X mice; held-out: 4 normal, 10 A1X, and 6 A5X mice; or both) is used to extract the top 20 behavioral features. Please indicate this information in the legend of Figure 2 and where else applicable.
7. On page 9 ("Notably...(Supplementary Fig. 5).") indicate which features show a "progressive narrowing".
8. I suggest replacing words like "vividly".
9. In the legend of Extended Data Figure 7, it should be Supplementary Table 2 and not 3.
10. I suggest replacing the group name "Normal" (and related expressions such as "normalcy") with the group name "Control".

Reviewer #2

(Remarks to the Author)

Overall, I find the study very interesting and revealing. There is no doubt that AI does provide a range of new endpoints as well as insights when applied in PD disease modelling. To help the reader I feel it is important that the methods and data details are presented clearly since the start, and opened perhaps in a more transparent manner so that the advance is better and easily visible. Also, the choices authors made should be discussed and presented in a sufficient detail which is essential for PD and other disease model specialists to whom the paper is mainly targeted. In the PD field one of the main bottlenecks indeed is the lack of well reproducible genetic PD animal models. Thus, it is important to help the reader to follow the line of logic authors implemented since the start. After some hesitation I decided to keep my original comments which I provided while reading into the manuscript. I have to admit I transformed from quite critical (that AI is better in quantifying image/movie data is already quite well known, that RET signaling might be helpful in PD is well known) to an admirer of this work. It is what the field needs -rather than new openings with little follow through, we need better methods to follow through the great leads we already have. This work if anything, illustrates this very well and suggests a route to future further advances.

As mentioned earlier, I keep my original comments which I wrote during the first time read, they I m quite sure, guide the authors well in terms of what to add and how to improve initial contextualization/providing details to better bridge to the part which is truly great -the AI PD evaluation and demonstration of RET timing effects on specific PD features.

Major comments

To score the PD animal model well, with AI or otherwise, one first needs to establish a model which reproduces PD-like features from an experiment to another with a set parametric for both genders. To set stage for this, authors should, throughout the manuscript, provide n numbers of animals and gender information at every Figure, S Figure and Extended Figures. (Comment after reading the ms until the end – I recommend early mention that only males were used, and still always make sure n number is clearly stated at each illustration and table, which however, in most cases is the case). There is great length of effort and literature which focuses solely on which AAV serotypes to use in PD models, thus information on serotype and virus titer should in my opinion be visible at the Figures not the rather arbitrary 1X and 5X. There also should be a stated rationale why A53T mutated alpha-synuclein was over-expressed in mice to induce PD, as in rodents unlike in humans A53T is the wt Snca sequence. Also, a clear injection site should be well described and visualized, so that it is clear that the substantia nigra (SN) is or is not mechanically damaged by the injection? Since SN lies deep intra SN delivery by definition increases inter animal variation due to mechanical lesion. This is important and also should be well controlled for (see below). It is confusing why DAT-Cre mice were used? Reference for the used Dat-Cre line is currently missing (line 88). There should be a Figure explaining rationale for Dat-Cre, and description of used constructs (FLX-STOP-FLX-aSyn A53T?). (After

reading the ms it still remains unclear why Dat-Cre mice were used? There seems to be no DA cell specificity implemented by e-g- FLX-STOP-FLX construct design?).

Then, in terms of behavioral assessment it is important that first authors set the stage for AI method by showing that there is a clearly quantified behavioral phenotype. At this point the authors use terminology in S Figure 1 which reads: increased on decreased locomotion, rotation, tremor etc with description: often, some, varied, rare, frequent, very rare...etc – which are not quantitative features. It is important that the authors align with scientific quantitative measurements such as parameter/event X or Y per time unit eg per second or minute etc to establish the existence of or the lack of the PD-like phenotype at S Fig 1. Authors should also discuss why they expect to see (or do observe) increased locomotion in rodent PD model where the central feature in humans is inability to move. (after reading the ms the explanation came later in the Results section, I'd recommend lifting it earlier)

In some cases authors could contextualize their story and results better by referring to and discussing appropriate data and literature in the field. More detailed comments/per line below.

Detailed questions/comments

Introduction

Line 57 it reads: "Addressing this, recent advancements in artificial intelligence (AI), particularly in pose-estimation techniques such as the AI Vision Analysis for Three dimensional Action in Real-time (AVATAR) system, have opened a new era in behavioural analysis 8-13."

I'd like to note that AVATAR, set forth by the authors, is not a peer-reviewed method as authors refer to their bioRxiv essays. I suggest to split the sentence referring to previous use of ML in behavioral studies with references while properly contextualizing AVATAR as their preliminary work.

Line 63 reads: "As the functional receptor of GDNF family ligands, c-Ret is central to the neuroprotective effects of the ligands^{7, 14}." I suggest to cite in vivo work which demonstrates requirement for RET to mediate GDNF effect in DA neurons in wt and PD mice, specifically PMID: 10331988 and PMID: 27607574.

Line 64 reads: "The complex relationship between c-Ret and its ligands complicates the targeted investigation of their role in PD." I suggest rephrasing and better opening of the idea as now this is somewhat confounding. As far as I am aware, there is no PD patient with mutations in GDNF family ligands or their receptors, or in that matter, with mutations in any receptor tyrosine kinase. Thus, it remains unclear what authors means by "role of GDNF family ligands in PD". The proper way to set the stage would in my view be to emphasize and open the pre-clinical and clinical work with GDNF family ligands which has yielded mixed but as yet promising results due to strong dopamine function and DA neuron survival promoting effect. There have been six clinical trials with GDNF and at least two currently ongoing with results being inconclusive but – in some cases, potentially promising.

Line 65 it reads: "Compounding this challenge, aSyn accumulation leads to the downregulation of c-Ret, as evidenced by a noticeable decrease in c-Ret immunoreactivity within the SNc DA neurons of PD patients postmortem^{15, 16}." First, I'd like to point out that protein is denoted by capital letters, thus c-RET. More importantly, there are several studies on this very topic, reviewed for example in PMID: 34916419. In summary, the issue is clearly not as simple as the authors present – the downregulation of RET cannot be said to be the reason why RET ligands fail in some but not in all aSyn-based models of PD, pls see the above reference and references therein.

Stemming for this the statement at lines 67 to 69: "Recently, a genetically encoded optogenetic intervention tool termed optoRET was shown to precisely modulate c-Ret signalling in the SNc of living mice¹⁷, giving it the potential to bypass the aSyn-associated obstacles and the ligand-related complications." is not quite true. I suggest that the authors contextualize their results in the light of published data on this topic and not state that optoRET solves a problem which, based on current evidence –does most likely not exist. Authors may well present optoRET as proof of concept (POC) tool to model putative new PD drugs with AI-based behavioral evaluation. Moreover, authors seemingly miss the point that if RET indeed is down in PD and that's the reason RET ligands do not work then RET ligands have no future in treating PD since by far most PD patients and about 30% of normally aging individuals, have Lewy Bodies.

Line 70: "Furthermore, the temporally controllable modulation of c-Ret signalling provides an additional therapeutic benefit, considering that prolonged signalling activation diminishes neuroprotective effects¹⁴" provides single reference to an in vitro study. Authors would benefit from familiarizing themselves with other published data on this, including: PMID: 37766790, which addresses the GDNF dose and exposure effects in vivo.

Results

As already mentioned, there should be i) reference for the used DAT-Cre line (Line 88), ii) there should be clear rationale for using A53T human aSyn, since mice already have A53T sequence in their aSyn as wt sequence and iii) there should be clear scheme depicting viral delivery constructs to provide rationale for DAT-Cre allele, together with iv) AAV serotype information and v) titer along with vi) clear sufficiently large illustrative figure of SN injection site, to make sure the reader at once, by looking at Figure 1, understands what has been done and why to damage the SN - is the injection supranigral or into the SN? Given the current major problem in the field– unreproducibility of published genetic PD models, I find this information very important for the informed reader. (after reading the ms I realize that AAV2 was delivered into the SN, however, as noted above key details should better come early with clear rationale).

Line 90: "The Normal group consisted of three subgroups, including WT mice without injection (WT) and DAT-CRE mice with AAV-RFP injection at two different doses (1X or 5X)."

Considering the above, and the well known effect of mechanical SN lesion on its own on motor function as well as that of at least EGFP delivery which results in clear lesion in the SN again on its own, it is important to compare first WT mice to AAV-

vehicle and to AAV-RFP 1X and 5X mice in AVATAR scheme to set the stage. A good experiment with good animal numbers (e.g. n=10 per group) for WT, AAV-vehicle, AAV-RFP and AAV-aSynA53T would set the stage well. This would also at once reveal the effect of mechanical SN lesion from that of RFP and A53T aSyn in the AVATAR setting. Are AVATAR detected features from aSyn, SN mechanical lesion, RFP, or a combination? It seems AVATAR has what it takes to answer that question. Considering the agony on the field over the lack of reproducibility of genetic PD models it is exactly where AVATAR could excel.

Second, the number of mice used along with information on gender should be clearly available on each Figure and Table, including Extended Data Figures and Supplementary Figures/Tables. Given the problem noted above, this is important for convincing specialists in the field. In some critical cases, e.g. Extended Figure 1g -statistical evaluation is missing? Are the groups different with the given number of mice?

Also, as noted above authors should open and provide rationale why they scored increased locomotion as a PD feature in AAV delivered mice (see S Figure 1)?

For setting up the base against what to validate the new AI based method authors use manual evaluation (S Figure 1)

Line 95: "Further behavioural assessments revealed that our PD mice exhibited various PD phenotypes, such as altered locomotion, akinesia, and tremor (Supplementary Fig. 1, Supplementary Movies 1-5)6, 20 "

...but do not provide quantitative parametrics, e.g. n of mice with this or that feature, failure of this or that in that time frame, etc. Authors list subjective descriptions, such as "Increased or decreased locomotion" and provide a measure "varied", which does not mean much in terms of quantitatively evaluating PD. Same holds for the other "scored" features which were noted as "varied, frequent, often and very rare", which are descriptive non-scientific terms. Event number/time etc should be used to provide quantitative phenotype assessment at S Fig 1.

This lack of explaining what was done and if and how data was quantified seems to emerge as frequent issue prior the AI implementation in the manuscript. For example Extended Figure 1h the ms describes: "we introduced a straightforward latency-based scoring system to quantify motor impairments across the entire cohorts (Extended Data Fig. 1h). Unlike other methods²² that incorporate parameters such as feet slip counts for detailed analysis, our method includes data from mice that were unable to complete the task, which would have been excluded by those methods. With the novel EBWT scoring approach, the A1X and A5X PD groups scored 2.2 } 0.57 and 3.55 } 0.45, respectively, at 10 wk (Fig. 1b)."

The problem here is that I do not see the necessary description on what exactly was measured and how exactly this method is innovative and discovers something others have not considered? It should be clearly stated what the numbers depicted on Extended Data Figure 1h measure, and it is very important to clearly spell out what (behavioral features-steps, falls, slips, etc etc) and how was measured on Figure 1b and on Extended Data Figure 1g and h with n numbers, gender information and employed statistical tests.

For the reader to see the advance, these details are critical, and they set the stage for the next steps in the ms with AI.

In my view authors should also, throughout the ms make clear what is peer reviewed published data and what is still their own unpublished research. Line 109 serves as a good example, stating: "To overcome the limitations of traditional task-dependent behavioural tests, which often

miss the subtleties of PD's complex symptomatology, we employed the AVATAR^{9, 10, 13, 23} system. This AI-based 3D pose estimation technique enables comprehensive and multidimensional behavioural assessments, revealing detailed behavioural phenotypes that are crucial for an accurate and early diagnosis of PD." The issue here is that what authors refer to as AVATAR is their own unpublished method where they cite Github repository for code and bioRxiv for manuscripts. I think for scientific rigor authors should make clear what is peer reviewed reference and what is their own unpublished work, not bundle them together.

However, this being said, I do appreciate the idea of using AI in scoring living animals, as depicted on Figure 1c. AI based image scoring has provided base for numerous startups during the recent years to improve diagnosis/analysis of image-based data, for example in cancer tissue histological evaluation. I do appreciate the idea spelled out on Line 114: "This system detects key nodes of a mouse's body in each movie frame and returns the 3D coordinates of the nodes (Fig. 1c)." I do agree that such "deep analysis", AI based or other, may improve the detection of behavioral anomalies in disease models including in PD. I agree that data shown on Figure 1d and S Figure 2c and d, when reproducible, shows an ability for better and deeper scoring than other available methods.

Line 136: "We further evaluated our model performance by testing an unseen-mouse dataset, composed of the Normal, A1X, and A5X groups. The PD scores at 10 wk were significantly higher for the A1X (50.92% \pm 8.75) and A5X (85.77% \pm 2.09) groups than for the Normal group (11.75 \pm 1.75; 139 Fig. 1g)." I consider this data very important, it seems that the results repeat, although the number of mice for some groups is really too low, n=4 for Normal mice (which is unknown mixture of WT and AAV-RFP SN delivered animals) and n=6 for A5X mice. Without providing data with proper n numbers and controls it is not possible for the reader to be sure what is reproducible and which feature comes from the SN mechanical lesion, RFP delivery, A53T aSyn delivery or combination?. Thus, I strongly recommend authors to increase the n number to clearly dissect their PD model and AVATAR potential. (After finishing the ms I now know that only males were used, but since many papers use mixed gender I think it will improve the ms to also state this clearly early in the text and in the Figure legends).

Line 141: "The PD and EBWT scores were strongly correlated, with a Pearson correlation coefficient of 0.82 (Supplementary Fig. 3)." Without knowing the detail about how and what was measured in EBWT it is hard to evaluate this data. As stated above, pls provide detail on EBWT -what units in what time per animal and how were measured?

Line 148: "The 20 most impactful features were identified, and their effects were visualised in a SHAP summary plot, offering a concise yet comprehensive view of the model's analytical depth (Fig. 2a)." I think this is very nice data, potentially leading the way into new scoring systems in PD models.

I do agree that for human eye unseen features may have been hidden in many PD models, and thus the AI based approach may open new doors in that direction. It will be interesting to see if authors linked PD features to striatal DA, DA fibers and DA neuron numbers in the SN later in the ms?

(after reading the ms the answer is yes).

I also appreciate that the difference between Normal (again, needs to be analyzed properly as suggested above) and "PD" animals for the 20 ML/AI defined features (S Figures 4-6). However, such clear difference in quantitative data makes me ask why was statistics not applied? Seems like very significant difference for most endpoints?

Line 162: "Another advantage of our model is its ability to account for feature interactions. SHAP dependence plots for each feature illustrate the influence of other features as the degree of vertical dispersion of the data points (Extended Data Fig. 4)."

I wonder in what way the authors mean the reader should gain more insight into the above advantage by looking at Extended Data Figure 4? I think this should be explained/visualized better with concrete correlating features. Thus I'd improve resolution of the statement at Line 165-166 "We employed colour coding to represent the values of the most interacting feature to the plotted feature, elucidating the relationship between them, which otherwise would remain undetermined." Perhaps again highlighting the potential of AVATAR in comparing WT to AAV-vehicle, AAV-RFP and AAV-A53T mice? This experiment takes WT animals +injections and 10 weeks followup plus analysis, thus I'd believe is quite in the range of a revision.

Line 168: "The XGB model pinpointed the top 20 features as crucial for PD diagnosis, particularly emphasising limb movements by ranking half of these features related to hands and feet (Fig. 2a)." I recommend colour coding the hand-feet rows to make Fig 2 a reader friendly.

Lines 170-178 – I consider this data very important, suggesting that "PD" animals do display clear hand and feet coordination defects. I agree that those features are hard or almost impossible to see, let alone properly score by human eye and by regular human eye-based tests. However, statistical evaluation of this likely numerical data would be important for scientific rigor? In case not possible to implement then explain why?

Lines 180-187 describe increased movement in PD model and the likely explanation - I feel this should come earlier in the ms, where increased/decreased movement in PD models is first described.

Lines 188-213 over body posture and stooped gait – also this data is very important and really promising. Overall, I feel that the authors might make even stronger case if they would repeat the study using another PD model, for example in striatal 6-OHDA or supranigral lactacystin-based PD models. Are the reported features, as expected, universal "Mouse lack of nigrostriatal DA features?" If so, we would be looking at new PD AI-based standard set of disease features in mice, turning page for the field?

(Now after reading the whole ms until the end I tend to think that this is not required in this ms. Right now it is more important to publish and let the field absorb and repeat in other PD models).

However, I'd again would like to point out that proper comparison/repeat analysis of WT to AAV-vehicle to AAV-RFP to AAV-A53T mice is important. It may help to train/tune the ML/AI system better, hopefully leading to clearer PD definition in mice and perhaps approaching then 100% accuracy as opposed to Figure 1e and f current results which, according to authors are close to about 85% PD prediction for 5X and about 51% for 1X.

Lines 214 – 282. I absolutely love optoRET system with blue LED activation from cage lids!

It shows that Daily RET activation regimen does not restore the DA system, which can be fundamental information to the field. This has clear implications to ongoing clinical trials with AAV-GDNF where GDNF cannot be turned off – which may lead to another failure based on REF 14 and on PMID: 37766790. Authors nicely demonstrate the critical role of timing and duration of RET activation in alleviating SN lesion driven defects. Essential and elegant.

I would perhaps, suggest reducing technical detail somewhat as exact Hz of chest tremor and possible link to human tremor in PD are possibly better topics for Discussion section, or for Extended Discussion section rather than for Results as we do not know how well those features if at all repeat in other supranigral lesion PD models and thus if they emerge as universal mouse PD correlates.

Lines 283 – 307, great data, may indicate optoRET specific features as well as reveal differential treatment potential, might set stage in the future for human AI based optoRET dosing regimen optimization, possibly guide way to precision medicine (PM) depending on individual treatment response? (Now after reading the ms I see that the authors indeed point this out and discuss this)

Lines 308 – turning and velocity, gait. Again, great data and analysis revealing new endpoints not achievable with any other sofar described tool! Possibly setting stage for PM at individual level first in animals, then hopefully in PD.

Discussion

The discussion, in large, covers the manuscript well. It also addresses several points and questions which rose while reading the ms. I agree that perhaps in this manuscript, it is not the right place to repeat the studies with another PD model as indeed, there are many and none is really good or at "gold standard" level.

Overall, I recommend publication with revision of some technical points and with some improved clarifications during the first part of the manuscript. Great work with implementing AI, loads of new endpoints found and also very revealing data on the timing of RET, this could be pointed out even better as it may inform ongoing and planned clinical trials.

Also, it may make sense to discuss that the lack of good PD models may in fact at least in part relate to low resolution of human eye, now potentially overcome by AI based array of mouse PD features?

Besides the improved presentation the only experiment I strongly recommend is analysis of WT v AAV-vehicle v AAV-RFP v AAV-A53T mice at 1X and 5X to understand what feature comes from what, and to solidly reproduce the data. This may also help to further improve the PD prediction. I realize that this is few months of quite hard work but I think it would both show the power of AVATAR and better set the stage against the current PD field where genetic PD models most often than not do not reproduce well between the studies. AVATR could change that.

Again, great work!

Reviewer #3

(Remarks to the Author)

Hyeon et al. introduced a novel approach to investigate the behavioral impairments in an animal model of Parkinson's disease (PD). The authors utilized a behavior monitoring system equipped with multiple cameras to detect subtle movements of individual animals and record various characteristics relating to their movement and posture. The results included a significant amount of data sets and they effectively organized them to provide well-defined behavioral phenotypes associated to Parkinson's disease. This comprehensive approach to behavioral phenotyping is innovative and would provide valuable insights for future research on movement disorders. I have just a few minor concerns.

1. The process of generating the PD score is not well explained. It is unclear whether the PD score reflects only the 20 major features or if other features are also taken into account. How individual features are weighted for this calculation should be explained with more clarity.

2. The description on A1X and A5X in the main text needs clarification. In line 92, the authors stated that they injected AAV-RFP into DAT-Cre mice at two distinct doses, specifically A1X or A5X. It is likely that AAV-RFP refers to an adeno-associated virus carrying the red fluorescent protein gene. If so, the injection of AAV-RFP shouldn't induce PD-like behaviors. This misstatement should be corrected. Overall, Multiple viral vectors were used in this investigation. Authors should include the full name and specific amounts (e.g. concentration and volume) of the viruses in the main text.

3. While the optogenetic activation of RET signaling in dopaminergic neurons is novel and interesting, it would be challenging to offer practical applications. Other medications that specifically target RET signaling, such as pre-clinically validated RET agonists can be utilized for treatment and their effects can be compared to the effects of OptoRET.

Reviewer #4

(Remarks to the Author)

In the manuscript "Integrating artificial intelligence and optogenetics for advanced Parkinson's disease diagnosis and therapeutics", the authors show a novel AI analysis of sensorimotor function and movement in an AAV-A53T mouse model of PD that can discriminate control mice from mice with different severity of nigrostriatal lesions. The AI and kinematic analysis system was able to detect multiple motor differences such as alterations in limb coordination, more variable movements, shorter paw movement amplitudes, and wider gait stance. They then demonstrate that an optogenetic intervention targeting c-RET signaling can improve multiple aspects of motor dysfunction. The AI motor analysis system is impressive and has the potential to be highly useful for ontogenetic studies. However, there are some concerns regarding the specificity to PD and whether it would show a similar profile of motor dysfunction in other PD models.

1. The authors show that the AI-kinematic analysis system is more sensitive at detecting changes in motor function than an elevated balance beam test but it is not clear how the system compares to other behavior tests that are commonly used including rotarod and gait treadmill equipment (ex. Catwalk).

2. Only one PD model was assessed, the AAV-A53T SNc injected mouse. For the AI system approach to be impactful it would have to show sensitivity across several commonly used models including the alpha-synuclein preformed fibril model and potentially a toxin-based model (MPTP or 6-OHDA).

3. It is not clear how specific the findings are to PD, a spinal cord injury model or motor neuron disease model would be important to assess and compare with PD models.

4. While the findings that the optogenetic intervention targeting c-RET is exciting, it would be important to show the effect of L-DOPA in the AAV-A53T model on the AI analysis.

5. It is not clear how much nigrostriatal cell loss there is in the AAV-A53T mice, from the picture in Figure 3e and the graph in Figure 3f it looks pretty severe in the A1X mouse, which is supposed to be less affected line. If it is severe loss then the AI

system is not necessarily that sensitive. Was stereology performed?

6. The word "transverse" is used several times when describing the beam test and I think they mean to use "traverse" instead.

Version 1:

Reviewer comments:

Reviewer #1

(Remarks to the Author)

The authors have implemented the majority of the requested revisions. Although the proposed optogenetic validation (using implanted fibers) was not performed, I do not consider it critical for the current scope of the study.

A remaining weakness, however, is the limited contextualization of AVATAR (and AVATARnet) relative to existing keypoint-tracking and behavioral-analysis frameworks that are widely used in neuroscience. At a minimum, the revised Discussion should devote a brief paragraph (e.g. ~3-5 sentences) that:

- Positions AVATAR against leading open-source keypoint-tracking pipelines such as DeepLabCut (DOI: 10.1038/s41593-018-0209-y), LEAP (DOI: 10.1038/s41592-018-0234-5), SLEAP (DOI: 10.1038/s41592-018-0234-5), DANNCE (DOI: 10.1038/s41592-018-0234-5) or other approaches, as well as behavioral-analysis pipelines such as MoSeq (DOI: 10.1038/s41592-018-0234-5; DOI: 10.1038/s41592-018-0234-5), A-SOID/B-SOID (DOI: 10.1038/s41592-024-02200-1; DOI: 10.1038/s41467-021-25420-x).

- Summarizes the key technical differences of the approach used in the manuscript (e.g. multi-camera 3-D triangulation, computational throughput, etc.) and indicate whether the phenotyping described here extends, like other pipelines, to other diseases models such as models of autism (MoSeq: DOI: 10.1038/s41593-020-00706-3), epilepsy (MoSeq: DOI: 10.1016/j.neuron.2023.02.003) or Huntington's disease (B-SOID: DOI: 10.1186/s12915-024-01919-9).

- Notes the absence of benchmarking on an independent public dataset, explains whether domain-specific constraints (e.g. unique enclosure geometry) preclude direct comparison, or outlines future plans to release data or to participate in common benchmarks (e.g. CalMS21, B-SOID datasets).

Incorporating this short comparative paragraph would satisfy community expectations for situating new AI methodologies and will help readers judge when AVATAR is preferable to existing tools.

Provided the authors add this discussion point, I recommend acceptance of the manuscript in its present form.

Reviewer #2

(Remarks to the Author)

I would like to thank the authors for addressing my concerns. I still suggest one minor but what I feel is an important revision - please include actual data from R1 and R5 studies (AAV-RFP delivery) on the main Figures/Tables throughout the manuscript. R1 and R5 mice show that RFP has toxic effect on DA neurons/DA function on its own, and it is very important and revealing for the PD field to see this throughout the story as part of the main data, including what AI can and currently cannot achieve there. I do not think that it damages the PD model story - aSyn over-expression v RFP over-expression are just different ways to damage the DA system as is mechanical damage (denoted MI or EV by the authors throughout the ms). Apart from that I have no further comments are concerns.

Reviewer #3

(Remarks to the Author)

The authors addressed all my previous concerns. I believe the authors incorporated an extensive amount of data to address the concerns of other reviewers and enhance the quality and conclusions of this work.

Reviewer #4

(Remarks to the Author)

The authors were responsive to the critiques and this reviewer believes the manuscript is improved both in terms of rigor and transparency.

- It is appreciated that a model of ALS was included to show the difference between a PD model and another movement disorder.

- The authors included new results from L-DOPA administration. The dose was high though and well above the therapeutic range (50mg/kg administered and therapeutic dose is ~6mg/kg) and likely explains the lack of effect.

- While another PD model was not included they do address the limits of the findings using a single model.

Overall, this work is important to the field.

Point-by-point responses to the reviewers' comments

We sincerely thank all reviewers for their constructive and insightful comments, which have been instrumental in identifying key issues and improving the overall quality of our manuscript. We have carefully addressed each of the concerns raised and have conducted the requested experiments where feasible. We are grateful for the opportunity to revise our work and believe that the manuscript has significantly improved as a result.

Revised sections in the manuscript are marked in **blue**. In this document, we provide point-by-point responses (in **blue**) to each reviewer comment (in **black**). Additionally, we include three summary tables outlining: (1) key terminology updates and textual revisions in the manuscript; and (2, 3) all changes made to the main Figures, Extended Data Figures, Supplementary Figures, Supplementary Notes, and Supplementary Tables.

In Original Manuscript (if applicable)	Abbreviation	Full name
	MIL	mechanically induced lesion
	POL	protein overload lesion
	AIL	alpha-Synuclein-induced lesion
	OFT	open-field test
	TST	tail suspension test
	RRT	rotarod test
	BWT	Elevated beam walking test
	TSS	tail suspension test score
	RRS	rotarod test score
	BWS	Elevated beam walking test score
	APS	AI-predicted PD score
	TRE	treatment response evaluation
	PSE	PD symptomatic evaluation
A53T	hA53T	mutant form of the human alpha-synuclein protein
c-Ret	c-RET	REarranged during Transfection
Normal	Control (CT)	
Normalcy or healthy	Non-PD (NP)	
RFP 1X	R1	Group information is provided in Supplementary Table 1
RFP 5X	R5	
RFP 10X	R10	
A1X	A1	
A5X	A5	
A1X+optoRET+Dark	A1OD	
A1X+S#1	A1O1	
A1X+S#2	A1O2	
A1X+S#3	A1O3	
A5X+optoRET+Dark	A5OD	
A5X+S#1	A5O1	
A5X+S#2	A5O2	

Key changes		Title
Main Figures		
Figure 1.	(a) schematic modified, B6J, AAV injection coordinates, enlarged image of injectino site. (b) BWS result was replaced by a table sumerising the group details for genotype and virus construct and titre. Panel, c, d, e, and f, Normal was replaced to Non-PD (NP) or Control (CT). Panel g, Normal group was replaced to EV and violin plot was replaced to dot plots, with increased n numbers for each group. Panel h is newly added.	AI-powered diagnosis in distinct severity cohorts of the hA53T PD mouse model.
Figure 2.	(a) Feature names, associated with limbs are highlighted in grey shading; the blue bar graphs and SHAP summary dot plots were replaced by higher resoputions; Normal was replaced to NP. (b) Normal was replaced to NP.	Exploring PD phenotypes through top 20 behavioural features with insights from XGB model interpretation.
Figure 3.	Pannel e,f and d order changed and panel g was removed. (b) Health status legend changed from Normal, Mild, Severe to Non-PD, Mild PD, Severe PD; Normal group was replaced to EV group. (c) only A1O2 group was removed, violin plot changed to dot plots. (d) olfactory tubercle, ot was replaced to tuberal area. Tu. (e) violin plots changed to dot plots. (f) a minor changes in the values due to the change in the statistical analysis (multiple comparison corrections), the total number of treated features (13) was inaccurately labelled in the original version; corrected to 13. Group names changed.	Evaluation of optoRET in alleviating PD symptoms and neurodegeneration.
Figure 4.	Panel d,e,f,g,h,i were changed to panel e,f,g,i,j,l and panel d,h,k were newly added.	Spectro-temporal insights into optoRET intervention, exploring tremor alleviation and movement complexities in PD.
Figure 5.	In all panels, group names were updated. Panel c-f, violin plots were changed to bar graphs with each n number indicated. Panel g changed to dot plots. (h) heatmap was updated (a minor changes due to the multiple comparisons correction). (i) a minor change in the values. (k) order of images were modified. (l) violin plots were replaced to bar graphs, with each n number.	optoRET enhancement in locomotion behaviour and prevention of foot trailing, the PD gait signature.
Extended Data Figures		
Extended Data Figure 1.	Paenl a,b,d were moved to Supplementary Fig. 1; panel c, e, f were updated to panel b, d, h, respectively; panels g,h were removed.	Development and characterisation of the hA53T PD mouse model.
Extended Data Figure 2.	In the legend, em dash was replaced to en dash.	Description of features engineered to develop AI model for PD diagnosis.
Extended Data Figure 3.	Panel a, group names updated. Panel b,c,d were updated to panel c,d,e, respectively. Panels b, e-j were newly added. Panel j is the updated version of Fig. 3g in the original manuscript.	Development and comparison of AI models for PD diagnosis.
Extended Data Figure 4.	Only legend updated	SHAP dependence plots of the top 20 features (XGB model).
Extended Data Figure 5.	(a) Normal was replaced to NP (non-PD). (b-d) were newly added to exemplify the difference between CT and PD groups.	Summary table of SHAP dependence plots for the top 20 features (XGB model).
Extended Data Figure 6.	Group names were updated. (a) colour bars were added and new data were added. (b) colour scheme of the heatmap changed. The names of categories were updated. The percentage of each cell was visualised. (c) A1O2 dot colour changed. (d) statistical results updated, including additional data and multiple comparisons corrections. Panel e-f were newly added (relevant data were from the Extended Data Fig. 8, in original manuscript).	Behavioural assessments (RRS, BWS, and APS) in PD mice treated with optogenetic stimulation (optoRET).
Extended Data Figure 7.	Entirely new figure.	Behavioural assessments (RRS, BWS, and APS) in A1 PD mice treated with L-DOPA or RET agonist (BT44).
Extended Data Figure 8.	Originally used to be Extended Data Fig. 7. The heatmap was moved to Supplementary Fig. 22a.	Group comparison of the top 20 features (XGB model) in the KDE plots at 10 wk.
Extended Data Figure 9.	Panel a, b, Normal was updated to NP (non-PD); panel c ,group names updated.	Model evaluation and SHAP dependence plots of top 20 features (TSFEL model).
Extended Data Figure 10.	Panel a, b, Normal was updated to NP (non-PD).	Model evaluation and SHAP dependence plots of top 20 features (Gait model).
Supplementary Tables		
Supplementary Table 1.	Newly added.	Summary of details for the key experimental groups.
Supplementary Table 2.	Newly added.	Statistical summary of the KL divergences for the top 20 features identified by the XGB model comparing CT and PD (A5) groups.
Supplementary Table 3.	Newly added.	Statistical summary of the feature values for the top 20 features identified by the XGB model, comparing the CT and PD (A5) groups.
Supplementary Table 4.	Newly added.	Statistical summary of the KL divergences for the top 20 features identified by the XGB model comparing NALS and ALS groups.
Supplementary Table 5.	Newly added.	Statistical summary of the feature values for the top 20 features identified by the XGB model, comparing the NALS and ALS groups.
Supplementary Table 6.	Newly added.	Statistical summary of the differences in the feature values for the top 20 features identified by the XGB model comparing PD (A5) and ALS groups.
Supplementary Table 7.	Newly added.	Statistical summary of the APS (XGB model-predicted PD scores) at 10 wk across groups.
Supplementary Table 8.	Originally used to be Supplementary Table 1. Group names were updated and the multiple comparisons refecnece to CT were included and the multiple comparions corrected p-values were included.	Statistical summary of the KL divergences for the top 20 features identified by the XGB model.
Supplementary Table 9.	Originally used to be Supplementary Table 2. Group names were updated and the multiple comparions corrected p-values were included.	Statistical summary of group comparisons (A1O3) for the top 20 features (identified by the XGB model) at 10 wk.
Supplementary Table 10.	Newly added.	Statistical summary of group comparisons (A1O2) for the top 20 features (identified by the XGB model) at 10 wk.
Supplementary Table 11.	Newly added.	Statistical summary of group comparisons (A1L1) for the top 20 features (identified by the XGB model) at 10 wk.
Supplementary Table 12.	Newly added.	Statistical summary of group comparisons (A1L3) for the top 20 features (identified by the XGB model) at 10 wk.
Supplementary Table 13.	Newly added.	Statistical summary of group comparisons (A1BT) for the top 20 features (identified by the XGB model) at 6 wk.
Supplementary Table 14.	Newly added.	Statistical summary of treatment group comparisons for the top 20 features (identified by the XGB model) at endpoints.
Supplementary Table 15.	Newly added.	Statistical summary of the KL divergences for the top 20 features identified by the TSFEL model comparing CT and PD (A5) groups.
Supplementary Table 16.	Newly added.	Statistical summary of the feature values for the top 20 features identified by the TSFEL model, comparing the CT and PD (A5) groups.
Supplementary Table 17.	Originally used to be Supplementary Table 3. Group names were updated and the multiple comparions corrected p-values were included.	Statistical summary of group comparisons (A1O3) for the top 20 features (TSFEL model) at 10 wk.
Supplementary Table 18.	Originally used to be Supplementary Table 4. Group names were updated and the multiple comparions corrected p-values were included.	Statistical summary of group comparisons (A1O3) for the turning features at 10 wk.
Supplementary Table 19.	Originally used to be Supplementary Table 5. Group names were updated and the multiple comparions corrected p-values were included.	Statistical summary of group comparisons (A1O3) for the rearing features at 10 wk.
Supplementary Table 20.	Originally used to be Supplementary Table 6.	Definitions of the Gait model top 20 features.
Supplementary Table 21.	Originally used to be Supplementary Table 7. Group names were updated and the multiple comparions corrected p-values were included.	Statistical summary of group comparisons (A1O3) for the top 20 features (Gait model) at 10 wk.

Key changes		Title
Supplementary Figures		
Supplementary Figure 1.	Newly added. Panel b-e are updated versions of panels from Exntedned Data Fig. 1 in the original manuscript).	Unilateral hA53T PD model failed to show dose-dependent PD severity.
Supplementary Figure 2.	Originally used to be Supplementary Fig. 1 . For the 'observation', the qualitative terms were replaced with quantitative measures; and the phenotypes which were unable to provide quantitative measures were removed in the revised table. Panels b-g were additionally provided to support the results in the table.	Bilateral hA53T PD mouse model exhibited diverse phenotypes.
Supplementary Figure 3.	Originally used to be Supplementary Fig. 2 . The Group names were updated.	Comparison temporal variabilities with t-SNE datasets.
Supplementary Figure 4.	Newly added.	Table of classifier training parameter by PyCaret default setting.
Supplementary Figure 5.	Newly added.	Comparison of cross-view (CV) and cross-subject (CS) validation methods.
Supplementary Figure 6.	Newly added.	Validation of APS for PD specificity using an amyotrophic lateral sclerosis (ALS) mouse model.
Supplementary Figure 7.	Newly added, including the relevant data, presented in the Supplementary Fig. 3a of the original manuscript.	Detailed plots of correlation analysis with varied dataset filtering at group level.
Supplementary Figure 8.	Newly added.	Comparison train-validation (TV) and unseen datasets for top 20 feature importances.
Supplementary Figure 9.	Newly added.	Comparison of body lengths across non-PD (NP) and PD groups.
Supplementary Figure 10.	Originally used to be Supplementary Fig. 4 . The group name in the title changed from Normal to CT.	CT group's longitudinal analysis of the top 20 features (XGB model) in the KDE plots.
Supplementary Figure 11.	Originally used to be Supplementary Fig. 5 . The group name in the title changed from PD (A5X) to PD (A5).	PD (A5) group's longitudinal analysis of the top 20 features (XGB model) in the KDE plots.
Supplementary Figure 12.	Originally used to be Supplementary Fig. 6 . The title and group names updated. In the original version, the control data was double-normalised and the A5 group included a subset of testing dataset by an error. In revision, these were corrected: each group normalised to each baseline only; and removing the testing dataset in A5 group, which was only included in the KDE plot analysis, and resampled and KDE analysis were repeated. The statistical comparisons were included (summerised in Supplementary Table 2).	Longitudinal comparison of KL divergences (XGB model) between the CT and PD (A5) groups.
Supplementary Figure 13.	Newly added, and the statistical details are summerised in the Supplementary Table 3 .	Longitudinal comparison of feature values (XGB model) between the CT and PD (A5) groups.
Supplementary Figure 14.	Newly added.	NALS group's longitudinal analysis of the top 20 features (XGB model) in the KDE plots.
Supplementary Figure 15.	Newly added.	ALS group's longitudinal analysis of the top 20 features (XGB model) in the KDE plots.
Supplementary Figure 16.	Newly added, and the statistical details are summerised in the Supplementary Table 4 .	Longitudinal comparison of KL divergences (XGB model) between the NALS and ALS groups.
Supplementary Figure 17.	Newly added, and the statistical details are summerised in the Supplementary Table 5 .	Longitudinal comparison of feature values (XGB model) between the NALS and ALS groups.
Supplementary Figure 18.	Newly added, and the statistical details are summerised in the Supplementary Table 6 .	Endpoint comparison of PD and ALS groups for the top 20 features identified by the XGB model.
Supplementary Figure 19.	Originally used to be Supplementary Fig. 7 . The group names were updated.	Group comparison of the decision tree plots (XGB model) at 10 wk.
Supplementary Figure 20.	Originally used to be Supplementary Fig. 8 . Comparisons of KL divercenes were updated (see, Methods).	Comparison of KL divergences of the top 20 features (XGB model) between groups at 10 wk.
Supplementary Figure 21.	Originally used to be Supplementary Fig. 9 . The group names were updated and the statistical details are summerised in the Supplementary Table 9 .	Group comparisons of the feature values (XGB model) at 10 wk.
Supplementary Figure 22.	Newly added. The Extended Data Fig. 7b in the original manuscript was moved here. Statistics are summerised in Supplementary Table 14 .	Comparisons of treatment response and PD symptomatic evaluations in A1 PD mice with different treatments.
Supplementary Figure 23.	Originally used to be Supplementary Fig. 10 .	Summary table of the features engineered and extracted for the spectro-temporal analysis.
Supplementary Figure 24.	Originally used to be Supplementary Fig. 11 . The group name in the title changed from Normal to CT.	CT group's longitudinal analysis of the top 20 features (TSFEL model) in the KDE plots.
Supplementary Figure 25.	Originally used to be Supplementary Fig. 12 . The group name in the title changed from PD (A5X) to PD (A5).	PD (A5) group's longitudinal analysis of the top 20 features (TSFEL model) in the KDE plots.
Supplementary Figure 26.	Originally used to be Supplementary Fig. 13 . The title and group names updated. In the original version, the control data was double-normalised and the A5 group included a subset of testing dataset by an error. In revision, these were corrected: each group normalised to each baseline only; and removing the testing dataset in A5 group, which was only included in the KDE plot analysis, and resampled and KDE analysis were repeated. The statistical comparisons were included (summerised in Supplementary Table 15).	Longitudinal comparison of KL divergences (TSFEL model) between the CT and PD (A5) groups.
Supplementary Figure 27.	Newly added, and the statistical details are summerised in the Supplementary Table16 .	Longitudinal comparison of feature values (TSFEL model) between the CT and PD (A5) groups.
Supplementary Figure 28.	Originally used to be Supplementary Fig. 14 . The group names were updated.	Group comparison of the KDE plots of the top 20 features (TSFEL model) at 10 wk.
Supplementary Figure 29.	Originally used to be Supplementary Fig. 15 . The group names were updated and the statistical details are summerised in the Supplementary Table17 .	Group comparison of the top 20 features (TSFEL model) at 10 wk.
Supplementary Figure 30.	Originally used to be Supplementary Fig. 16 . The group names were updated and the statistical details are summerised in the Supplementary Table18 . Statistical results slightly changed from the multiple comparison corrections.	Group comparison of the turning features at 10 wk.
Supplementary Figure 31.	Originally used to be Supplementary Fig. 17 . The group names were updated and the statistical details are summerised in the Supplementary Table19 . Statistical results slightly changed from the multiple comparison corrections.	Group comparison of the rearing features at 10 wk.
Supplementary Figure 32.	Originally used to be Supplementary Fig. 18 . The group names were updated. A red shading in subplot #3 and #8 were added and panel b were included to highlight the extreme feet distance and stance width observed in PD.	Group comparison of the top 20 features (Gait model) in KDE plots at 10 wk.
Supplementary Figure 33.	Originally used to be Supplementary Fig. 19 . The group names were updated and the statistical details are summerised in the Supplementary Table21 . Statistical results slightly changed from the multiple comparison corrections.	Group comparison of the top 20 features (Gait model) in bar plots at 10 wk.
Supplementary Figure 34.	Newly added.	Snapshot video images of CT and PD mice in walking.
Supplementary Notes		
Supplementary Note 1.	Newly added.	Comparative evaluation of lesion types affecting the DA system
Supplementary Note 2.	Newly added.	Model evaluation methods
Supplementary Note 3.	Newly added.	Comparative evaluation of AI-based and conventional behavioural assessments
Supplementary Note 4.	Newly added.	Stability and generalisability of the feature selection
Supplementary Note 5.	Newly added.	Assessment of body size as a potential confounding factor
Supplementary Note 6.	Newly added.	Basal activity of optoRET in A5 PD mice
Supplementary Note 7.	Newly added.	Interpretation of a RET agonist (BT44) in the hA53T overexpression mouse PD model
Supplementary Note 8.	Newly added.	Interpretation of L-DOPA in the hA53T overexpression mouse PD model

**Reviewer #1 (Remarks to the Author):**

Hyeon, Shin, and colleagues present an impressive body of work describing the behavioral phenotype
in a mouse model of Parkinson's disease (PD). Their work addresses the complexities of PD, a
progressive neurodegenerative disorder with challenging motor symptoms and limited treatment
options. By integrating artificial intelligence (AI) with optogenetic intervention, the authors aim to
overcome the limitations of traditional task-specific behavioral assessments, which often rely on a
narrow set of metrics. The study employs a sophisticated combination of freely available packages,
such as AVATAR, PyCaret, SHAP, TreeExplainer, and TSFEL, to detect and describe the PD
phenotype in freely moving mice. Using 3D pose estimation techniques together with explainable AI,
the authors identify a wide range of behavioral markers, including gait and spectro-temporal features.
While the experiments and analysis are both elegant and cutting-edge, addressing critical questions
in the field, there are some drawbacks. The optogenetic experiments, in particular, require further
investigation, and there are concerns regarding the statistical analyses.

Thank you for your thoughtful and encouraging comments. We appreciate your recognition
of our integrative approach and would be glad to address the concerns raised in the revised
manuscript.

Critique

Major

1. Since the time series of 3D keypoints form the basis of the PD scoring pipeline, noisy keypoints
could significantly affect the extraction of relevant behavioral features (e.g., discussed in
DOI:10.1101/2023.03.16.532307). Thus, it is essential to validate and report the keypoint prediction
accuracy of AVATARnet on unseen test data acquired in the AVATAR studio. Additionally, testing
AVATARnet on an independent dataset (e.g., https://github.com/jessedmarshall/CAPTURE_demo) is
highly encouraged to ensure robustness and generalizability.

**Response #1.1**

We sincerely thank the reviewer for raising this important issue. In response, we have revised our
manuscript to include additional evaluation details of AVATARnet in the Methods section (**lines 665–**
**673**). As previously reported in our bioRxiv preprint¹, and demonstrated in the peer-reviewed studies^{2–}
⁴, the system's accuracy was rigorously assessed using mAP (90%), IoU (>75%), and MSE (1.4–4.5
34 mm). Given these robust validation results – and considering the commercial deployment of the
35 system via Actnova (actnova.io) – we did not further test AVATARnet on an independent dataset (e.g.
CAPTURE_demo). We believe these revisions adequately address the reviewer's concerns regarding
keypoint prediction accuracy and overall system robustness, and we are very grateful for this valuable
suggestion.

2. The split into training and validation datasets (10 normal and 11 A5X mice) should be done on a
41 per mouse basis, not on a per clip basis, to minimize the possibility that classifiers identify subject-
42 specific behaviors in the validation dataset. Although the authors showed that the XGB classifier
trained on held-out clips can generalize to unseen mice (4 normal, 10 A1X, and 6 A5X mice), it is still
possible that another model among the 18 evaluated could generalize better to unseen data (i.e., new
mice). This also applies to the selection of the most relevant features (top 20 features). I suggest
repeating the evaluation of the 18 different classifiers and the selection of the most relevant features
using held-out mice instead of held-out clips to confirm that the XGB model performs best and the top
20 features remain the same.

**Response #1.2**

We sincerely thank the reviewer for the insightful and constructive comments regarding our
evaluation strategy. In our original analysis, we employed a cross-view (CV) evaluation, training
models on CT (n = 10) and A5 (n = 11) mice and validating on unseen subjects from the same groups
(CT, n = 4; A5, n = 6). Acknowledging that cross-subject (CS) evaluation is widely regarded as a more
stringent test of generalisation, yet both methods are highly valid in the machine learning (ML) field⁵,
we repeated the ML processes to develop models with a CS method. Specifically, we included 9
additional mice (CT, n = 4; A5, n = 5) to maintain a subject-level split with approximately 70% training
and 30% validation. The manuscript has been revised (**lines 168–169**), and the results are
summarised in **Supplementary Figure 5 and Supplementary Note 2**.

Notably, the compared model performances were almost the same with both validation methods
(**Supplementary Fig. 5a**). A direct comparison of the top 20 features between the XGB-CV and XGB-
CS models revealed a Spearman correlation of ≈ 0.9789 ($p \approx 7.15e-14$), indicating that the feature
ranking is nearly identical with 100% of the top 20 features overlapping (**Supplementary Fig. 5b**). In
addition, the absolute SHAP values (as measured by Pearson correlation) were nearly identical
between the two protocols ($r \approx 0.9999$, $p \approx 4.81e-114$), further supporting the consistency of feature
importance. Furthermore, one-way ANOVA with Holm–Sidak's multiple comparisons showed no
significant differences in prediction outcomes across all groups between the XGB-CV and XGB-CS
models and highly correlated ($p > 0.05$; Pearson's correlation, $r = 0.99$; **Supplementary Fig. 5c,d**).
Importantly, our original XGB-CV model consistently assessed all unseen A5 mice into the severe PD
category (APS > 75% in rounded values), providing additional rationale for retaining our initial
evaluation method.

Collectively, these findings robustly confirm the generalisability and stability of our original CV-
based XGB model. Given the close agreement in both feature ranking and predictive performance
between the CV and CS evaluations, we remain confident in our model choice. We greatly appreciate
the reviewer's recommendation, which has significantly strengthened our analysis and validated the
robustness of our methodological approach.

3. Related to the last point (Major #2), to test whether the top 20 features remain the most relevant
across datasets (i.e., test the robustness of the approach), the top 20 behavioral features should be
compared between the train-val dataset (10 normal and 11 A5X mice) and the held-out dataset (4
normal and 6 A5X mice).

**Response #1.3**

We thank the reviewer for the valuable suggestion to assess the robustness of our feature
selection. To address the concern, we compared the top 20 behavioural features derived from the
combined train-validation (TV) dataset with those obtained from the unseen dataset (CT, n = 40; A5, n
= 13). The manuscript has been revised (**lines 207–208**), and the results are summarised in
**Supplementary Figure 8 and Supplementary Note 4.**

The analysis revealed a 100% overlap in the top 20 features, with a Spearman correlation of \approx
0.9835 ($p \approx 8.29e-15$). A separate comparison of the absolute SHAP values for these top 20 features
yielded a Pearson correlation of about 0.2013 ($p \approx 0.3948$), indicating some variability in the absolute
magnitudes; however, given the near-perfect consistency in ranking, this variation in magnitudes is
considered of lesser concern. Together, these results provide robust quantitative evidence that the top
20 features remain highly relevant across the TV and unseen datasets, thereby confirming the stability
and generalisability of the feature selection approach. We are grateful for the reviewer's
recommendation, which has significantly strengthened our analysis.

4. Many behavioral features relevant for PD classification (e.g., "Body 2D length") could be affected
by body size differences between the groups. I suggest estimating the body size by, for example,
calculating the mean size of the connected keypoints (i.e., individual parts of the skeleton) across the
entire session (not just clips) for each mouse. Then, compare these sizes between the control and PD
groups to exclude confounds related to group-specific body size differences.

**Response #1.4**

We sincerely thank the reviewer for raising this important issue regarding potential confounds
related to body size differences. To address this concern, we conducted an extensive analysis of body
lengths using both the raw pose dataset (as recommended) and the move dataset (i.e. the filtered
dataset used in our study). Here, the body length was defined as the sum of the forebody (fbody;
neck-chest distance) and hindbody (hbody; chest-anus distance) lengths in 3D, filtering for frames
where the body angle was within ± 10 degree before averaging the body lengths for each mouse at
each week. Please note that this measure differs from the 'body length (neck-anus distance)'
presented in our original manuscript (see, Fig. 2a and Extended Fig. 2b). The manuscript has been
revised (**lines 207–208**), and the results are summarised in **Supplementary Figure 9 and**
**Supplementary Note 5.**

In the raw pose dataset (**Supplementary Fig. 9a, left**), mean body lengths at 0 wk were highly
consistent across groups (CT: 6.58 ± 0.02 , A1: 6.58 ± 0.04 , A5: 6.6 ± 0.04 cm; n = 46, 15, 16,
respectively). However, significant differences emerged at 10 wk (CT: 6.45 ± 0.04 , A1: 6.76 ± 0.07 , A5:
6.99 ± 0.06 cm; n = 48, 15, 18, respectively), likely due to variations in body posture during non-
moving (freeze) states (accounting approximately 60% of the raw pose dataset). Crucially, analysis of
the move dataset (**Supplementary Fig. 9a, right**) showed no significant differences in body lengths
between groups at either 0 wk (CT: 6.95 ± 0.05 , A1: 6.99 ± 0.12 , A5: 7.06 ± 0.05 cm; n = 45, 14, 13) or
10 wk (CT: 7.02 ± 0.06 , A1: 7.28 ± 0.12 , A5: 7.13 ± 0.21 cm; n = 41, 9, 11). The 2-way ANOVA tests
supported these observations (**Supplementary Fig. 9b**): the pose dataset showed significant effects
for time ($p = 0.0006$), group ($p < 0.0001$), and their interactions ($p < 0.0001$), whereas the move
dataset showed no significant effects for group (time: $p = 0.0456$; group: $p = 0.2201$; interaction: $p =$
0.4359).

Additionally, we re-analysed preliminary data to further assess the robustness of our APS (AI-
prediction of PD scores). In this exploratory analysis, we compared data from three individual mice:
R1(50), low dose RFP injection at 50 wk; A1(10), low dose A53T injection at 10 wk; and A1(50), low
dose A53T injection at 50 wk. Although these analyses involved only a single mouse per condition
(thus preliminary), the results were consistent with our expectations: A1(10) exhibited a smaller body
length consistent with younger age, whilst R1(50) and A1(50) had similar lengths to our primary
dataset (**Supplementary Fig. 9c**). Importantly, despite the variations in body lengths, APS
measurements remained robust: R1(50) maintained APS within non-PD status, whilst both A1(10) and
A1(50) exhibited APS consistent with mild PD health status (**Supplementary Fig. 9d**).

Collectively, these findings provide robust evidence that our distance-based features are unlikely
to be confounded by body size (neck-chest-anus) differences, particularly within the move dataset
used for our primary analysis. Although the raw pose dataset exhibited differences (likely related to
postural variations in the non-moving periods), our core analyses on the move dataset did not reflect
significant confounds related to body size. We are extremely grateful to the reviewer for this insightful
recommendation, which has prompted additional validations and significantly strengthened the
robustness and generalisability of our approach.

5. Adding statistics for the changes described in Suppl. Fig. 4-6 is recommended, given they are used
to identify PD-specific movement patterns such as “postural imbalances” and “a shift from restricted to
more erratic and diverse movements” as PD progresses.

**Response #1.5**

We thank the reviewer for the recommendation. In the revised manuscript, we have updated both
the figures and the related text. Firstly, the statistic results are summarised in the **Supplementary**
**Tables 2 and 3**. For the maximum chest 1D velocity (#16), we now include a mini-plot
(**Supplementary Fig. 11**) that shows a significant decrease in the percentage of data above the key

value – from 18% to 1% over 0 to 10 wk. For the standard deviation of the neck 3D velocity (#9), we
have revised the manuscript text (**lines 270–274**) by removing the phrase ‘indicating a shift from
restricted to more erratic and diverse movements as PD advanced,’ as this appeared over-
interpolated. We believe these revisions sufficiently address the reviewer’s comments.

6. Statistics (e.g., Supplementary Table 2) lack multiple comparisons correction.

**Response #1.6**

We thank the reviewer for pointing this out. In the revised manuscript, we have updated all
statistics (e.g., Supplementary Tables) with multiple comparisons corrections where appropriate.
Details of the statistical analyses, including the correction methods, are provided in the legends of the
tables and figures.

7. The optoRET treatment results are difficult to interpret (Fig. 3). For example, for A1X, there is an
effect in some of the light groups and no effect in the dark group. In contrast, there is no effect in A5X
groups with optoRET stimulated with light, but a difference in those mice without light, compared to
the A5X group. Please comment (especially on the difference in the results between light vs. dark and
A1X vs. A5X).

**Response #1.7**

We thank the reviewer for their insightful comment. To address this concern, we have revised the
manuscript (**lines 270–274 and 487–491**) to direct the reader to **Supplementary Table 7** and
**Supplementary Note 6**, which summarise the detailed statistical comparisons between the optoRET
groups and discuss the basal activity of optoRET in A5 PD mice, as detailed below.

Statistically, optoRET activation was ineffective in the A5 mice (see **Supplementary Table 7**),
although a subtle, non-significant trend was observed in the A5OD (A5+optoRET+dark) and A5O2
(A5+optoRET+S#1) groups. Our in vitro studies suggest that the effects of optoRET are non-linear,
with optimal light conditions yielding the most beneficial effects⁶. Although the light-sensitive domain
of optoRET has been engineered to minimise basal activity, some residual effect remains. In severe
PD states (A5), this basal activity may confer a modest benefit compared to daily, full light activation
(A5O1), whereas in milder PD stages the effect appears negligible. We postulate that the basal effect
may have protected A5 mice from the high protein overload lesion (POL) rather than the alpha-
Synuclein-induced lesion (AIL; see **Extended Data Fig. 1a–f**). Further studies are required to
elucidate these state-dependent dynamics; for instance, future experiments could investigate whether
overexpression of R5 with optoRET can prevent motor dysfunction caused by POL.

Given that these trends did not reach statistical significance, our detailed analyses focused on the
condition exhibiting the most robust response (A1O3). **Supplementary Table 7**, detailing the group
comparisons in **Extended Fig. 6d**, now includes an additional comment addressing this issue. We

appreciate the reviewer's insight and agree that further studies are warranted to fully understand the
state-dependent dynamics of optoRET actions.

8. Given the results and that the light was delivered transcranially through an array of LEDs in the
homecage lid, I suggest repeating some of the optoRET experiments using implanted optic fibers.

196 **Response #1.8**

We thank the reviewer for this thoughtful and important suggestion. We fully acknowledge that
fibre-optic light delivery remains the gold standard in many optogenetic studies, particularly for
achieving precise, localised stimulation. However, in the context of our study, we deliberately adopted
a transcranial light delivery strategy using an LED array mounted on the homecage lid, with the
primary aim of enabling a minimally invasive, low-stress, and longitudinal (10 wk) behavioural follow-
up.

Whilst implanted optic fibres with tethered laser sources can indeed provide high spatiotemporal
precision, their use is associated with significant stress and physical constraints – factors that can
substantially influence behavioural outcomes, especially in chronic paradigms. The repeated handling
and tethering required for fibre-based stimulation may introduce confounding variables that are
difficult to control, particularly when investigating subtle phenotypes such as those seen in prodromal
or early-stage conditions.

Wireless optogenetic systems with implantable μ -LEDs represent a promising alternative and may
alleviate some of these concerns; however, current limitations – including device miniaturisation,
power supply constraints, and limited programmability – continue to pose practical challenges for
long-term, untethered experiments, particularly in freely moving animals over extended durations.

Although several innovative devices have been developed^{7, 8}, many of these remain limited in
terms of accessibility, robustness, or widespread adoption, owing to some of unresolved challenges
related to device size, energy management, and flexible programmability.

Broadly, the field of in vivo optogenetics has progressed along two complementary directions: one
focusing on the development of advanced light delivery systems (e.g., wireless or miniaturised
devices), and the other – including our own efforts – on engineering optogenetic tools with enhanced
photosensitivity or red-shifted activation spectra, to enable effective stimulation under less invasive
conditions.

Our work contributes to the latter direction, with efforts focused on improving the photosensitivity
and expression efficiency of optogenetic tools, including the development of red-shifted variants
optimised for non-invasive, transcranial activation. Our experimental results demonstrate that blue-
light delivery through the skull is sufficient to reliably activate optoRET⁶, which contains a modified
light-responsive domain for increased sensitivity, as evidenced by downstream signalling activation,

including increased phosphorylation of S6 and ERK1/2 in the nigral dopaminergic neurons and
midbrain.

In summary, although fibre-optic implants can be valuable in certain experimental contexts, we
believe they are not essential for the aims of this study. Our chosen approach reflects a careful
balance between stimulation efficacy and the need to minimise stress and preserve ecological validity
in a chronic disease model. We hope this explanation clarifies the rationale behind our methodological
decisions. Once again, we sincerely thank the reviewer for their constructive comment, which has
allowed us to more fully articulate and justify our experimental design.

9. How different light stimulation paradigms affect the PD features should be clarified. For example,
how does the “Treatment response evaluation” and “PD symptomatic evaluation” (Fig. 3d) look for
A1X+S#2? Do the affected features and categories overlap with those of A1X+S#3?

**Response #1.9**

We thank the reviewer for the comment. In the revised manuscript, we have included direct
comparisons of A1O2 and A1O3 with regard to both the treatment response evaluation (TRE) and PD
Symptomatic Evaluation (PSE). In addition, two further treatment groups have been incorporated. The
statistical results are now summarised in **Supplementary Tables 9–13**, and the overall data for TRE
and PSE across the different treatment groups are illustrated using a heatmap and pie charts in
**Supplementary Fig. 22**. Specific details regarding the treated features between A1O2 and A1O3 can
be found in **lines 330–333**.

10. In the abstract, results, and discussion, the authors highlight that the AI approach (XGB model)
outperforms traditional task-dependent assessments. However, there seems to be no direct
(statistical) comparison between the two approaches. Supplementary Figure 3c only shows the
scores (model and EBWT) for two data points.

**Response #1.10**

We thank the reviewer for this important comment. In the revised manuscript, we have added
direct statistical comparisons between the AI approach (AI-predicted PD score [APS]) and traditional
task-dependent assessments (rotarod test score [RRS] and beam walking test score [BWS]). These
comparisons include: (1) sensitivity in discriminating between the two PD cohorts at the endpoint, (2)
early detection of PD cohorts, and (3) correlation with histological data. The results of these
comparisons are summarised in **Fig. 3h and Supplementary Note 2**.

(1) **Lines 175–177**: We compared the p-values for differences between the A1 and A5 groups at
the 10 wk endpoint in RRS, BWS, and APS assessments (**Extended Data Fig. 1g,h; Fig. 1g**). The p-
values were 0.0678 (RRS), 0.0423 (BWS), and 0.0002 (APS), indicating that APS provides the most
sensitive discrimination between PD severity cohorts.

(2) **Lines 180–183:** We assessed early-stage (2 wk) detection of the PD cohorts using RRS,
BWS, and APS (**Extended Data Fig. 3g–i**). At 2 wk post-surgery, RRS failed to distinguish any PD
groups from EV, although the A5 group showed a trend toward significance ($p = 0.0521$). BWS
significantly distinguished the A5 group from EV ($p = 0.0018$), but not the A1 group ($p = 0.0965$). In
contrast, APS successfully distinguished both PD groups from EV (A1: $p = 0.0270$; A5: $p < 0.0001$)
and further detected a significant difference between the A1 and A5 groups ($p = 0.0270$). These
results indicate that APS enables earlier and more sensitive detection of PD compared to RRS and
BWS.

(3) **Lines 189–199:** We evaluated the absolute Pearson correlation coefficients ($|r|$) between
behavioural assessments (RRS, BWS, and APS) and histological data (**Extended Data Fig. 3j**). For
striatal TH intensity, $|r|$ values were 0.61 (RRS), 0.74 (BWS), and 0.82 (APS). For nigral DA cell count,
the values were 0.61 (RRS), 0.76 (BWS), and 0.82 (APS). These results indicate that APS exhibits
the strongest correlation with PD pathology among the three methods.

Lastly, we also revised the relevant sections in the abstract and discussion to reflect these
updates. In the abstract, the sentence ‘Without task-specific constraints, these models discriminated
distinct PD severity cohorts with higher precision and accuracy at earlier stages, outperforming
conventional methods.’ has been revised to ‘Without task-specific constraints, these models enabled
earlier and more accurate discrimination of PD severity cohorts compared to conventional methods.’
In the revised first paragraph of the Discussion, we incorporated detailed comparative results to align
with these findings.

We believe these revisions fully address the reviewer’s concern and improve the clarity, rigour,
and impact of the study.

11. In the discussion section, the findings of the study should be more thoroughly integrated and
compared with previously published work.

**Response #1.11**

We thank the reviewer for this helpful comment. In the revised Discussion, we focused on
contextualising our key findings while addressing their potential, limitations, and implications. We have
incorporated relevant comparisons with previously published work where appropriate.

Specifically, we referenced prior studies when discussing our observation that intermittent
optoRET stimulation protocols yielded more beneficial effects (**lines 471–473**), aligning with earlier
findings suggesting that discontinuous activation of RET signalling may be more effective.

We also related our results with L-DOPA and BT44 to existing knowledge, underscoring the
importance of tailoring medication protocols to the mechanism of action and therapeutic strategy
(**lines 474–475**).

Furthermore, when discussing the mechanisms underlying optoRET's therapeutic effects, we
integrated our own previously published in vitro work (**line 479**), which proposed a role for RET
signalling in axonal regeneration. We also highlighted a recently published in vivo imaging tool
developed by our group (**lines 483–487**), suggesting how it could be employed in future studies to
further elucidate optoRET's mechanisms of action.

We believe these revisions more thoroughly situate our findings within the existing literature and
strengthen the Discussion accordingly.

Minor

1. Unit (weeks) missing in Figure 1a and Extended Data Figure 1a.

**Response #1.12**

We thank the reviewer for pointing this out. We have updated the relevant figures to include the
unit (weeks).

2. Add in the legend of Fig. 1c and the methods section the underlying architecture of AVATARnet
and triangulation (e.g., adapted YoloV4, triangulation, and bundle-adjustment algorithms), including
their references.

**Response #1.13**

We thank the reviewer for this valuable suggestion. In the revised manuscript, the **legend of Fig.**
**1c** now includes detailed information about AVATARnet: 'The underlying architecture, AVATARnet, is
adapted from YOLOv4 and modified to include 53 CNN layers. 3D coordinates are computed using
triangulation and bundle-adjustment algorithms.' These details, along with the relevant references,
have also been incorporated into the Methods section (**lines 674–681**).

3. In the paragraph discussing the t-SNE approach, the sentence "However, interpreting behavioral
data has proven challenging, prompting a shift to using feature-engineered datasets instead of raw or
t-SNE data." is unclear and should be rephrased. For example: "However, interpreting behavioral data
in an unsupervised manner has proven challenging, prompting us to create feature-engineered
datasets to train classifiers in a supervised fashion."

**Response #1.14**

Thank you for the suggestion. We have revised our manuscript in **lines 156–157** as suggested.

4. It is unclear what parameters were used for the 18 different classifiers. A table should be added or

the details should be described in the method section what parameters were used for the different
classifiers.

**Response #1.15**

We thank the reviewer for the comment. We used the default settings for training classifiers with
PyCaret (**version 2.3.10**, which was the most up-to-date version at the time of our analysis). Detailed
parameter settings for all compared classifiers have been included in **Supplementary Fig. 4** and in
**Extended Data Fig. 3b** for the XGB model. Additionally, we noted that the reference to ‘18 different
classifiers’ in the originally submitted manuscript was an error; we have revised the number to ‘15
models’ in **line 164**.

5. In Fig. 1d, it is unclear in the right plot if the skeletons are consecutive frames or frames with a
temporal offset. In addition, the term “Action Skeletons” in the figure and in the text is confusing. I
suggest revising it to “connected keypoints” or something similar.

**Response #1.16**

We thank the reviewer for the comment. In the revised manuscript, we have updated the legend of
**Fig. 3d** to clarify that the skeleton images of motion sequences were sampled with temporal offsets –
every 5 frames for rearing and 3 frames for walking. We have retained the term ‘Action Skeletons’ to
maintain consistency with the original pre-print¹ of the AVATAR system and related peer-reviewed
publications²⁻⁴.

6. It is unclear which dataset (train-val: 10 normal and 11 A5X mice; held-out: 4 normal, 10 A1X, and
6 A5X mice; or both) is used to extract the top 20 behavioral features. Please indicate this information
in the legend of Figure 2 and where else applicable.

**Response #1.17**

We thank the reviewer for this comment. In the revised manuscript, we have clarified in the
legend of **Fig. 2a** that the top 20 behavioural features were extracted using the validation dataset. For
clarity, SHAP values were initially computed using the training dataset and then updated with the
validation dataset, in accordance with the default settings of the SHAP library.

7. On page 9 (“Notably...(Supplementary Fig. 5).”) indicate which features show a “progressive
narrowing”.

**Response #1.18**

We thank the reviewer for the comment. In the revised manuscript, we have specified the feature
numbers in the format (# feature rank) where applicable – for example, (#16) is now included after the
feature name that exhibits progressive narrowing (line 246). Additionally, we have updated
**Supplementary Fig. 11** to include a mini-plot, which demonstrates a significant decrease in the

percentage of data above the key value – from 18% to 1% over 0 to 10 weeks – to support our
description.

8. I suggest replacing words like “vividly”.

**Response #1.19**

We thank the reviewer for the comment. We have revised the manuscript by removing adjectives
such as ‘vividly.’

9. In the legend of Extended Data Figure 7, it should be Supplementary Table 2 and not 3.

**Response #1.20**

We thank the reviewer for the comment. Yes, it should have been Supplementary Table 2 instead
of Table 3. In the revised manuscript, we have moved the heatmap for A1O3 to **Supplementary Fig.**
**22** and updated the table accordingly – now referred to as **Supplementary Table 9** (which
corresponds to the originally submitted Supplementary Table 2).

10. I suggest replacing the group name “Normal” (and related expressions such as “normalcy”) with
the group name “Control”.

**Response #1.21**

We thank the reviewer for the comment. We have replaced the group name ‘Normal’ with ‘Control
(CT)’ and revised related expressions, such as ‘normalcy or healthy’ to ‘non-PD (NP)’ where
applicable.

**Reviewer #2 (Remarks to the Author):**

Overall, I find the study very interesting and revealing. There is no doubt that AI does provide a range
of new endpoints as well as insights when applied in PD disease modelling. To help the reader I feel it
is important that the methods and data details are presented clearly since the start, and opened
perhaps in a more transparent manner so that the advance is better and easily visible. Also, the
choices authors made should be discussed and presented in a sufficient detail which is essential for
PD and other disease model specialists to whom the paper is mainly targeted. In the PD field one of
the main bottlenecks indeed is the lack of well reproducible genetic PD animal models. Thus, it is
important to help the reader to follow the line of logic authors implemented since the start. After some
hesitation I decided to keep my original comments which I provided while reading into the manuscript.
I have to admit I transformed from quite critical (that AI is better in quantifying image/movie data is
already quite well known, that RET signaling might be helpful in PD is well known) to an admirer of
this work. It is what the field needs -rather than new openings with little follow through, we need better
methods to follow through the great leads we already have. This work if anything, illustrates this very
well and suggests a route to future further advances.

As mentioned earlier, I keep my original comments which I wrote during the first time read, they I m
quite sure, guide the authors well in terms of what to add and how to improve initial
contextualization/providing details to better bridge to the part which is truly great -the AI PD evaluation
and demonstration of RET timing effects on specific PD features.

Thank you for your thoughtful and constructive feedback; we have revised the manuscript to
improve clarity, provide detailed rationale for our methodological choices, and better contextualise the
significance of our AI-based PD evaluation and RET intervention, guided by your original comments.

Major comments

To score the PD animal model well, with AI or otherwise, one first needs to establish a model which
reproduces PD-like features from an experiment to another with a set parametric for both genders. To
set stage for this, authors should, throughout the manuscript, provide n numbers of animals and
gender information at every Figure, S Figure and Extended Figures. (Comment after reading the ms
until the end – I recommend early mention that only males were used, and still always make sure n
number is clearly stated at each illustration and table, which however, in most cases is the case).

**Response #2.1**

We thank the reviewer for the comment. In the originally submitted manuscript, gender details
were provided in Fig. 1a and in the Methods section but were not explicitly described in the main
text. In the revised manuscript, we have now explicitly stated 'B6J male mice' at the beginning of the
Results section (line 107). Additionally, we have ensured that the n numbers of animals are clearly
indicated within each figure. If any additional details are required, please let us know.

There is great length of effort and literature which focuses solely on which AAV serotypes to use in
PD models, thus information on serotype and virus titer should in my opinion be visible at the Figures
not the rather arbitrary 1X and 5X.

Response #2.2

We thank the reviewer for the comment. In the revised manuscript, we explicitly mention the viral
serotype ('**DJ/8 serotype**') in the main text (**line 106**) and have updated **Fig. 1b** to summarise the
experimental groups clearly, including the details of AAV-DJ/8-hSyn1 viral constructs and their titres.
Additionally, we have included **Supplementary Table 1**, providing comprehensive information such as
subgroup classification, experimental interventions, genetic backgrounds, gender, genotypes,
experimental timelines, AAV serotype with titres, and the numbers of mice used across different
datasets and analyses. The total mouse numbers indicated represent all mice included in this study,
while the specific subset numbers are clearly described in relevant sections of the manuscript. Any
groups not listed in this table or with amended details are explicitly addressed within the
corresponding text.

There also should be a stated rationale why A53T mutated alpha-synuclein was over-expressed in
mice to induce PD, as in rodents unlike in humans A53T is the wt Snca sequence.

Response #2.3

We thank the reviewer for this insightful comment. In the revised manuscript, we have clarified the
rationale for using human A53T alpha-synuclein (hA53T) to induce PD in mice. We now explicitly state
in **lines 98–99**: 'Although A53T represents a wild-type (WT) variant in rodents, the introduction of
hA53T is pathogenic in mice due to the sequence differences in other residues' (**Supplementary Fig.**
**1a**).

We have also included additional detail in the figure legend: Sequence alignment of human
(UniProt ID: P37840) and mouse (UniProt ID: O55042) alpha-synuclein (aSyn) amino acid
sequences, highlighting seven mismatches (A53T, S87N, L100M, N103G, A107Y, D121G, and
N122S). Positions of familial PD-associated substitutions (A30P, E46K, G51D, A53E, and A53T) are
indicated by asterisks (*). Notably, substitutions in the C-terminal region of mouse aSyn (G121 and
S122) effectively dampen membrane-induced aSyn aggregation and vesicle permeabilisation, thereby
protecting rodents from the deleterious effects exerted by the A53T mutation in humans.'*

Additionally, we have revised the abbreviation from 'A53T' to 'hA53T' throughout the manuscript to
clearly indicate that the expressed protein is of human origin. We believe this revision improves clarity
regarding the model rationale and strengthens the biological justification for its use.

Also, a clear injection site should be well described and visualized, so that it is clear that the
substantia nigra (SN) is or is not mechanically damaged by the injection? Since SN lies deep intra SN
delivery by definition increases inter animal variation due to mechanical lesion. This is important and
also should be well controlled for (see below).

Response #2.4

We thank the reviewer for highlighting this important point. In the revised manuscript, we have
clearly illustrated and described the stereotaxic coordinates of viral injection sites 'into' the bilateral
SNc, along with an enlarged schematic brain image (**Fig. 1a**). To address concerns regarding
potential mechanical damage to the SNc, we included a *mechanically induced lesion (MIL)* control
group (empty vector-injected [EV]), which showed no significant impairment in motor function or
reduction in nigral DA cell counts (**lines 119–121; Extended Data Fig. 1a–f**).

It is confusing why DAT-Cre mice were used? Reference for the used Dat-Cre line is currently missing
(line 88). There should be a Figure explaining rationale for Dat-Cre, and description of used
constructs (FLX-STOP-FLX-aSyn A53T?). (After reading the ms it still remains unclear why Dat-Cre
mice were used? There seems to be no DA cell specificity implemented by e-g- FLX-STOP-FLX
construct design?).

Response #2.5

We thank the reviewer for highlighting this important issue. We acknowledge that the original
manuscript did not clearly describe the rationale behind using DAT-CRE mice. In the revised
manuscript, we explicitly clarified the rationale in **lines 112–115**: 'To exclude the possibility that
different genetic backgrounds could affect the PD diagnostic system, we included the DAT-CRE line,
which was used for the expression of DIO-optoRET in later experiments'. Additionally, we provided
detailed tables summarising the genotypes and constructs used across different groups (**Fig. 1b;**
**Supplementary Fig. 1**). The appropriate reference for the DAT-CRE line is also included.

Then, in terms of behavioral assessment it is important that first authors set the stage for AI method
by showing that there is a clearly quantified behavioral phenotype. At this point the authors use
terminology in S Figure 1 which reads: increased on decreased locomotion, rotation, tremor etc with
description: often, some, varied, rare, frequent, very rare...etc – which are not quantitative features. It
is important that the authors align with scientific quantitative measurements such as parameter/event
X or Y per time unit eg per second or minute etc to establish the existence of or the lack of the PD-like
phenotype at S Fig 1.

Response #2.6

We thank the reviewer for this important comment. In the revised manuscript, we have updated
the corresponding figure (now **Supplementary Fig. 2**) to include quantitative measurements rather

than qualitative descriptions. Specifically, we now present the proportion (%) of mice in the A5 group
exhibiting each behavioural phenotype based on clearly defined thresholds. The updated figure
provides numerical details of altered locomotion, motor coordination, tremor, and dystonia observed
during the open-field test (OFT), elevated-beam walking test (BWT), and tail suspension test (TST).
Additionally, metrics for body centre position during OFT and thresholds for hyperactivity, rotation, and
thigmotaxis have been explicitly quantified (panels b–f). Hindlimb clasping has also been
quantitatively assessed by TST scores (panel g; detailed in Methods). We believe these revisions
address the reviewer’s point and align our data presentation with appropriate quantitative standards.

Authors should also discuss why they expect to see (or do observe) increased locomotion in rodent
PD model where the central feature in humans is inability to move. (after reading the ms the
explanation came later in the Results section, I’d recommend lifting it earlier)

**Response #2.7**

We thank the reviewer for this insightful comment. In the original manuscript, we explained the
observation of increased locomotion in lines 183–185: ‘Although bradykinesia is a canonical symptom
of PD, aSyn-overexpressing animal models often exhibit hyperactivity, affected by the upregulation of
the nigral DA receptor D1 and the downregulation of the striatal DA transporter’. Following the
reviewer’s recommendation, we have now moved this explanation earlier in the revised manuscript,
now clearly stating it at the point where the animal model is first introduced (**lines 103–105**).
Additionally, to provide a balanced interpretation, we included supplementary information on
habituation effects within the locomotion results section (**lines 246–252**).

In some cases authors could contextualize their story and results better by referring to and discussing
appropriate data and literature in the field. More detailed comments/per line below.

Detailed questions/comments

Introduction

Line 57 it reads: “Addressing this, recent advancements in artificial intelligence (AI), particularly in
pose-estimation techniques such as the AI Vision Analysis for Three dimensional Action in Real-time
(AVATAR) system, have opened a new era in behavioural analysis 8-13.”

I d like to note that AVATAR, set forth by the authors, is not a peer-reviewed method as authors refer
to their bioRxiv essays. I suggest to split the sentence referring to previous use of ML in behavioral
studies with references while properly contextualizing AVATAR as their preliminary work.

**Response #2.8**

We thank the reviewer for this comment. Following the reviewer’s suggestion, we have revised
the sentence to clearly differentiate between other ML approaches and our preliminary AVATAR work.
In the revised manuscript (**lines 55–59**), we first broadly mention advancements in AI and pose-

estimation techniques, citing appropriate peer-reviewed references. We then explicitly introduce the
AVATAR system, clearly stating it was initially described in a preprint (lines 59–60) and subsequently
validated by several independent peer-reviewed studies (lines 60–61): ‘Building on this foundation,
we initially introduced the AI Vision Analysis for Three-dimensional Action in Real-time (AVATAR)
system in a preprint¹³, which has been validated in several peer-reviewed studies’. We believe this
revision fully addresses the reviewer’s suggestion by properly contextualising AVATAR as preliminary
work, while acknowledging related peer-reviewed research.

Line 63 reads: “As the functional receptor of GDNF family ligands, c-Ret is central to the
neuroprotective effects of the ligands^{7, 14}.” I suggest to cite in vivo work which demonstrates
requirement for RET to mediate GDNF effect in DA neurons in wt and PD mice, specifically PMID:
10331988 and PMID: 27607574.

**Response #2.9**

We thank the reviewer for this comment. In the revised manuscript, we have explicitly addressed
the reviewer’s suggestion by including the recommended references (PMID: 10331988 and PMID:
27607574). This addition strengthens our statement by providing direct in vivo evidence that RET is
required for GDNF-mediated neuroprotection, both in wild-type and PD mouse models.

Line 64 reads: “The complex relationship between c-Ret and its ligands complicates the
targeted investigation of their role in PD.” I suggest rephrasing and better opening of the idea as now
this is somewhat confounding. As far as I am aware, there is no PD patient with mutations in GDNF
family ligands or their receptors, or in that matter, with mutations in any receptor tyrosine kinase.
Thus, it remains unclear what authors means by “role of GDNF family ligands in PD”. The proper way
to set the stage would in my view be to emphasize and open the pre-clinical and clinical work with
GDNF family ligands which has yielded mixed but as yet promising results due to strong dopamine
function and DA neuron survival promoting effect. There have been six clinical trials with GDNF and
at least two currently ongoing with results being inconclusive but – in some cases, potentially
promising.

**Response #2.10**

We thank the reviewer for this insightful and constructive comment. In the revised manuscript, we
have rephrased and expanded the relevant section to more clearly reflect the context of prior
preclinical and clinical investigations into GDNF family ligands and their therapeutic potential in PD.

We now highlight the body of preclinical and clinical work (lines 66–68), which has produced
mixed but potentially promising outcomes, with beneficial effects reported in subsets of patients. To
support the rationale for introducing optoRET, we have added the sentence (lines 68–70): ‘These

inconsistencies highlight the need for alternative tools to investigate the crosstalk between PD-
associated factors and c-RET signalling in PD models.’

We believe that these revisions now more accurately reflect the current landscape of GDNF-
based research and provide a clearer and more appropriate context for introducing our experimental
approach using optoRET.

Line 65 it reads: “Compounding this challenge, aSyn accumulation leads to the downregulation of c-
Ret, as evidenced by a noticeable decrease in c-Ret immunoreactivity within the SNc DA neurons of
PD patients postmortem^{15, 16}.” First, I’d like to point out that protein is denoted by capital letters,
thus c-RET. More importantly, there are several studies on this very topic, reviewed for example in
PMID: 34916419. In summary, the issue is clearly not as simple as the authors present – the
downregulation of RET cannot be said to be the reason why RET ligands fail in some but not in all
aSyn-based models of PD, pls see the above reference and references therein.

Response #2.11

We thank the reviewer for this valuable and well-informed comment. In the revised manuscript,
we have updated the protein notation to ‘c-RET’ to accurately reflect the correct naming convention.

Additionally, we have revised the relevant paragraph, particularly in **lines 75–80**, to reflect the
complexity of the relationship between aSyn accumulation and c-RET expression. As highlighted in
the review by Conway and Kramer (2022, PMID: 34916419), the interplay between aSyn and
GDNF/RET signalling is complex and context-dependent, involving multiple regulatory factors such as
Nurr1 and Nedd4. We acknowledge that c-RET downregulation is not consistently observed and
cannot by itself explain the variability in response to RET ligands in aSyn-based models. We believe
the updated text presents a more balanced interpretation and better frames the rationale for studying
optoRET.

We believe this revised description accurately addresses the reviewer’s concerns; however, we
remain open to further refinement if necessary.

Stemming from this the statement at lines 67 to 69: “Recently, a genetically encoded optogenetic
intervention tool termed optoRET was shown to precisely modulate c-Ret signalling in the SNc of
living mice¹⁷, giving it the potential to bypass the aSyn-associated obstacles and the ligand-related
complications. “ is not quite true. I suggest that the authors contextualize their results in the light of
published data on this topic and not state that optoRET solves a problem which, based on current
evidence –does most likely not exist. Authors may well present optoRET as proof of concept (POC)
tool to model putative new PD drugs with AI-based behavioral evaluation. Moreover, authors
seemingly miss the point that if RET indeed is down in PD and that’s the reason RET ligands do not

work then RET ligands have no future in treating PD since by far most PD patients and about 30% of
normally aging individuals, have Lewy Bodies.

**Response #2.12**

We thank the reviewer for this important clarification. In the revised manuscript, we have clearly
indicated that optoRET is primarily presented as a proof-of-concept (POC) tool for investigating c-RET
modulation (**lines 70–74**): ‘Recently, we introduced a genetically encoded optogenetic tool termed
optoRET as a proof-of-concept platform for precisely modulating c-RET signalling in the SNc of living
mice. Unlike traditional ligand-based approaches, optoRET enables temporally controlled, ligand-
independent activation, offering a versatile experimental framework for exploring c-RET-targeted
therapeutic strategies.’

We also acknowledge the reviewer’s important point regarding the limited therapeutic potential of
RET ligand-based strategies, particularly if endogenous c-RET is consistently downregulated in PD
and ageing populations. Accordingly, we have revised the manuscript, providing more detailed
background (**lines 75–80**) and our rationale (**lines 80–82**): ‘Given these complexities, overexpression
of optoRET could offer an additional advantage by enabling c-RET modulation independently of
endogenous receptor levels’.

Furthermore, we have added new experimental data (A1BT group: A1 mice receiving brain
infusion of a RET agonist, BT44, a compound previously validated in a 6-OHDA rat PD model⁹) to
directly test a RET agonist in our PD model. As described in **Response #3.3**, we observed no clear
therapeutic effect, which we interpret as reflecting the ligand’s dependence on endogenous c-RET,
where the extent of aSyn overexpression likely exceeds that seen in human PD pathology¹⁰. We
additionally provide a more detailed interpretation of this in **Supplementary Note 7**.

We believe these revisions more accurately reflect current evidence, acknowledge the complexity
raised by the reviewer, and clearly position optoRET as an experimental research strategy rather than
a claim of therapeutic resolution. We remain open to further refinement if needed.

Line 70: “Furthermore, the temporally controllable modulation of c-Ret signalling provides an
additional therapeutic benefit, considering that prolonged signalling activation diminishes
neuroprotective effects¹⁴” provides single reference to an in vitro study. Authors would benefit from
familiarizing themselves with other published data on this, including: PMID: 37766790, which
addresses the GDNF dose and exposure effects in vivo.

**Response #2.13**

We thank the reviewer for providing this important reference. We acknowledge that at the time our
study was conceptualised, we had primarily referenced the single in vitro study¹¹ cited in our original
manuscript. The additional in vivo reference (PMID: 37766790) provided by the reviewer significantly
strengthens the rationale for temporally controlled modulation of c-RET signalling. We have now

included this reference in the revised manuscript, further supporting our rationale for temporally
controlled c-RET modulation.

Results

As already mentioned, there should be i) reference for the used DAT-Cre line (Line 88), ii) there
should be clear rationale for using A53T human aSyn, since mice already have A53T sequence in
their aSyn as wt sequence and iii) there should be clear scheme depicting viral delivery constructs to
provide rationale for DAT-Cre allele, together with iv) AAV serotype information and v) titer along with
vi) clear sufficiently large illustrative figure of SN injection site, to make sure the reader at once, by
looking at Figure 1, understands what has been done and why to damage the SN - is the injection
supranigral or into the SN? Given the current major problem in the field– unreproducibility of
published genetic PD models, I find this information very important for the informed reader. (after
reading the ms I realize that AAV2 was delivered into the SN, however, as noted above key details
should better come early with clear rationale).

Response #2.14

We thank the reviewer for this important comment. As mentioned in our previous **Responses**
**#2.1–2.5**, we have addressed all points raised in this comment explicitly in the revised manuscript:

(i) Reference for the DAT-Cre line has been added (**line 114**).

(ii) Rationale for using human A53T aSyn has been clearly provided (**lines 96–99**).

(iii)–(v) Details of the viral delivery constructs, including the AAV-DJ/8 serotype and virus titre,
have been comprehensively described in the revised **Fig. 1b** and **Supplementary Table 1**.

(vi) Clear visualisation and stereotaxic coordinates for SN injection have been illustrated with an
enlarged schematic image (**Fig. 1a**).

Given the complexity of the experimental groups, we have summarised all critical details clearly
and concisely in **Supplementary Table 1**, ensuring ease of understanding for the informed reader.

Line 90: “The Normal group consisted of three subgroups, including WT mice without injection (WT)
and DAT-CRE mice with AAV-RFP injection at two different doses (1X or 5X).”

Considering the above, and the well known effect of mechanical SN lesion on its own on motor
function as well as that of at least EGFP delivery which results in clear lesion in the SN again on its
own, it is important to compare first WT mice to AAV-vehicle and to AAV-RFP 1X and 5X mice in
AVATAR scheme to set the stage. A good experiment with good animal numbers (e.g. n=10 per
group) for WT, AAV-vehicle, AAV-RFP and AAV-aSynA53T would set the stage well. This would also
at once reveal the effect of mechanical SN lesion from that of RFP and A53T aSyn in the AVATAR
setting. Are AVATAR detected features from aSyn, SN mechanical lesion, RFP, or a combination? It

seems AVATAR has what it takes to answer that question. Considering the agony on the field over
the lack of reproducibility of genetic PD models it is exactly where AVATAR could excel.

**Response #2.15**

We thank the reviewer for this insightful comment. In the revised manuscript, we have carefully
addressed the reviewer's suggestion by explicitly including and thoroughly comparing relevant control
groups to distinguish the specific effects attributable to *mechanically induced lesion* (**MIL**, i.e., EV
group), *protein overload lesion* (**POL**, i.e., R1 or R5 groups), and bilateral *alpha-Synuclein-induced*
*lesion* (**AIL**, i.e., A1 or A5 groups), using sufficient animal numbers (approximately n = 10 per group
for beam walking score [BWS] and AI-predicted PD score [APS] analysis), as clearly summarised in
**Supplementary Table 1 and Supplementary Note 1** (Comparative evaluation of lesion types
affecting the DA system).

Comparative analysis of the Control (CT) subgroups suggested that MIL did not significantly
worsen motor function nor decrease nigral DA cell counts (10% decrease, normalised to NI;
**Extended Data Fig. 1a–f**). Contrarily, the POL – particularly the R5 group – showed evidence of
toxicity affecting motor function in rotarod test score (RRS; p = 0.0265) and DA neuron integrity, as
indicated in the revised manuscript **lines 121–125**. Additionally, BWS and APS analyses revealed no
significant differences among the subgroups within control (CT) animals (**Extended Data Fig. 1d**;
**Extended Data Fig. 3f**).

The varied outcomes the behavioural assessments suggest differing sensitivities: RRS may
reflect generalised motor coordination, while BWS likely assesses fine coordination and balance¹².
Importantly, our APS metric – trained for non-PD (NP) class using dataset prepared from NI, R1 and
R5 groups – specifically captures behavioural features associated with AIL rather than POL.

We fully agree with the reviewer that AVATAR is particularly suited for addressing reproducibility
concerns and delineating PD-specific phenotypes with high precision.

Second, the number of mice used along with information on gender should be clearly available on
each Figure and Table, including Extended Data Figures and Supplementary Figures/Tables. Given
the problem noted above, this is important for convincing specialists in the field. In some critical
cases, e.g. Extended Figure 1g -statistical evaluation is missing? Are the groups different with the
given number of mice?

Also, as noted above authors should open and provide rationale why they scored increased
locomotion as a PD feature in AAV delivered mice (see S Figure 1)?

For setting up the base against what to validate the new AI based method authors use manual
evaluation (S Figure 1)

**Response #2.16**

We thank the reviewer for this important comment. In the revised manuscript, we have ensured that
the number of mice and gender (male mice) information is explicitly indicated in all relevant Figures,
Extended Data Figures, Supplementary Figures, and Tables. Regarding **Extended Data Fig. 1g**, this
panel was removed in the revised manuscript as it is no longer relevant following updates to the beam
walking score (BWS) analysis. Additionally, as previously mentioned (**Response #2.7**), we have
provided a clear rationale for scoring increased locomotion as a PD feature in aSyn-overexpressing
AAV mice at an earlier stage in the manuscript (**lines 103–105**), accompanied by quantitative
assessments in **Supplementary Fig. 2**. This comprehensive approach strengthens the reliability and
clarity of our findings and directly addresses the reviewer’s concerns.

Line 95: “Further behavioural assessments revealed that our PD mice exhibited various PD
phenotypes, such as altered locomotion, akinesia, and tremor (Supplementary Fig. 1, Supplementary
Movies 1-5)6, 20 “

...but do not provide quantitative parametrics, e.g. n of mice with this or that feature, failure of this or
that in that time frame, etc. Authors list subjective descriptions, such as “Increased or decreased
locomotion” and provide a measure “varied”, which does not mean much in terms of quantitatively
evaluating PD. Same holds for the other “scored” features which were noted as “varied, frequent,
often and very rare”, which are descriptive non-scientific terms. Event number/time etc should be
used to provide quantitative phenotype assessment at S Fig 1.

**Response #2.17**

We thank the reviewer for this important comment. We acknowledge that our original manuscript
presented behavioural phenotypes with subjective descriptions. In the revised manuscript, we have
replaced these qualitative terms with clearly defined quantitative measures (**Supplementary Fig. 2**).
Specifically, we now provide precise metrics, including the percentage (%) of mice exhibiting each
specific phenotype within defined quantitative thresholds, event frequencies, and clear measurement
parameters (e.g. metrics for locomotion, rotation, and hindlimb claspings), ensuring a robust and
scientifically rigorous behavioural assessment.

This lack of explaining what was done and if and how data was quantified seems to emerge as
frequent issue prior the AI implementation in the manuscript. For example Extended Figure 1h the ms
describes: “we introduced a straightforward latency-based scoring system to quantify motor
impairments across the entire cohorts (Extended Data Fig. 1h). Unlike other methods²² that
incorporate parameters such as feet slip counts for detailed analysis, our method includes data from
mice that were unable to complete the task, which would have been excluded by those methods. With
the novel EBWT scoring approach, the A1X and A5X PD groups scored 2.2 ± 0.57 and 3.55 ± 0.45 ,
respectively, at 10 wk (Fig. 1b).”

The problem here is that I do not see the necessary description on what exactly was measured and
how exactly this method is innovative and discovers something others have not considered? It should

be clearly stated what the numbers depicted on Extended Data Figure 1h measure, and it is very
important to clearly spell out what (behavioral features-steps, falls, slips, etc etc) and how was
measured on Figure 1b and on Extended Data Figure 1g and h with n numbers, gender information
and employed statistical tests.

For the reader to see the advance, these details are critical, and they set the stage for the next steps
in the ms with AI.

**Response #2.18**

We thank the reviewer for highlighting this important issue. We apologise for the insufficient
description of our BWS method in the original manuscript.

**In the original manuscript**, we measured the latency (in seconds) required for each mouse to
traverse the elevated beam, averaging the two fastest trials out of three trials per animal. These
latency values were converted into a percentage relative to baseline performance and categorised
into six score ranges (scores 0 to 5), as summarised in Extended Data Fig. 1h. Importantly, **score 5**
**included data from mice** that failed to complete the task or were unable to perform due to mortality.

**In the revised manuscript**, we have updated and clearly described the beam walking scoring
(BWS) methodology in detail (Methods section, **lines 612–628**). Specifically, we measured latency
(seconds) required for each mouse to traverse the elevated beam, averaging the two fastest attempts
from three trials per animal, and converted these latency values into percentages relative to baseline
performance. **We explicitly defined exclusion criteria as follows, with trials classified as failures**
**(types I and II) were excluded from further analysis:**

(1) mice unable to reach the beam endpoint in three trials (failure type I);

(2) mice with BWS above 600% (failure type II); and

(3) mice unable to perform due to mortality.

Consequently, we have removed the previous mortality data and score-range classifications from
the revised Extended Data Fig. 1, replacing these with a clear schematic illustration (**Extended Data**
**Fig. 1b**) explicitly detailing the measurements (beam traversal latency). These extensive revisions
enhance transparency, scientific rigour, and reproducibility, thereby effectively setting the stage for the
subsequent AI-based analyses presented in our manuscript.

In my view authors should also, throughout the ms make clear what is peer reviewed published data
and what is still their own unpublished research. Line 109 serves as a good example, stating: “To
overcome the limitations of traditional task-dependent behavioural tests, which often
miss the subtleties of PD's complex symptomatology, we employed the AVATAR9, 10, 13, 23 system.
This

AI-based 3D pose estimation technique enables comprehensive and multidimensional behavioural
assessments, revealing detailed behavioural phenotypes that are crucial for an accurate and early
diagnosis of PD.” The issue here is that what authors refer to as AVATAR is their own unpublished
method where they cite Github repository for code and bioRxiv for manuscripts. I think for scientific
rigor authors should make clear what is peer reviewed reference and what is their own unpublished
work, not bundle them together.

Response #2.19

We thank the reviewer for raising this important point. In the revised manuscript, we have clearly
distinguished between peer-reviewed published references and our own preliminary and other peer-
reviewed work using the AVATAR system (**lines 59–61**). Specifically, when referencing the AVATAR
system, we explicitly state that it was first introduced in a preprint (bioRxiv), while separately indicating
studies that have been independently peer-reviewed and published (see **Response #2.8**).

However, this being said, I do appreciate the idea of using AI in scoring living animals, as depicted on
Figure 1c. AI based image scoring has provided base for numerous startups during the recent years
to improve diagnosis/analysis of image-based data, for example in cancer tissue histological
evaluation. I do appreciate the idea spelled out on Line 114: “This
system detects key nodes of a mouse's body in each movie frame and returns the 3D coordinates of
the nodes (Fig. 1c).” I do agree that such “ deep analysis”, AI based or other, may improve the
detection of behavioral anomalies in disease models including in PD. I agree that data shown on
Figure 1d and S Figure 2c and d, when reproducible, shows an ability for better and deeper scoring
than other available methods.

Response #2.20

We sincerely thank the reviewer for this supportive and encouraging comment. We agree with the
reviewer that AI-based deep behavioural analysis, as illustrated in **Fig. 1c** and demonstrated through
the data in **Fig. 1d and Supplementary Fig. 2c–d**, offers significant potential to improve the detection
and quantification of subtle behavioural anomalies in disease models, including PD. We appreciate
the reviewer’s recognition of the innovation and potential impact of our approach.

Line 136: “We further evaluated our model performance by testing an unseen-mouse dataset,
composed of the Normal, A1X, and A5X groups. The PD scores at 10 wk were significantly higher for
the A1X (50.92% ± 8.75) and A5X (85.77% ± 2.09) groups than for the Normal group (11.75 ± 1.75;
139 Fig. 1g).” I consider this data very important, it seems that the results repeat, although the
number of mice for some groups is really too low, n=4 for Normal mice (which is unknown mixture of
WT and AAV-RFP SN delivered animals) and n=6 for A5X mice. Without providing data with proper n
numbers and controls it is not possible for the reader to be sure what is reproducible and which
feature comes from the SN mechanical lesion, RFP delivery, A53T aSyn delivery or combination?.

Thus, I strongly recommend authors to increase the n number to clearly dissect their PD model and
AVATAR potential. (After finishing the ms I now know that only males were used, but since many
papers use mixed gender I think it will improve the ms to also state this clearly early in the text and in
the Figure legends).

Response #2.21

We thank the reviewer for highlighting this critical point. In the revised manuscript, we have
explicitly clarified early in the text (**line 105**) and clearly indicated genders in **Supplementary Table 1**.
Additionally, we included a new subsection titled 'Mouse lines and genders' in the Methods (**lines**
**522–528**), explicitly stating that only male mice were used, except for the motor neuron disease
(MND) cohorts. We acknowledge the reviewer's concern regarding group sizes and controls in the
unseen-mouse dataset. To robustly dissect the behavioural phenotypes and clearly attribute them to
either mechanically induced lesion (MIL), protein overload lesion (POL), or aSyn-induced lesion (AIL),
we substantially increased the number of animals per group (approximately $n = 10$, except for the
RRS and IHC data) and incorporated clearly defined control groups. These revisions enhance clarity
and reproducibility, ensuring that behavioural differences observed via AVATAR specifically reflect the
PD model rather than confounding factors. Lastly, in the revised manuscript, APS at 10 wk for the A1
and A5 groups showed minor changes (**Fig. 1g and h**): the A1 group changed from $50.92\% \pm 8.75$ ($n = 10$)
to $58.87\% \pm 6.53$ ($n = 15$), and the A5 group from $85.77\% \pm 2.09$ ($n = 6$) to $86.62\% \pm 1.79$ ($n =$
12). However, these slight differences reasonably reflect reproducibility and consistency with the
original findings, as the reviewer noted.

Line 141: "The PD and EBWT scores were strongly correlated, with a Pearson
correlation coefficient of 0.82 (Supplementary Fig. 3)." Without knowing the detail about how and what
was measured in EBWT it is hard to evaluate this data. As stated above, pls provide detail on EBWT -
what units in what time per animal and how were measured?

Response #2.22

We thank the reviewer for highlighting this important issue. As mentioned previously, we have
revised the BWS methodology for clarity and scientific rigour (see **Response #2.18**). Additionally,
**Supplementary Fig. 3** from the original manuscript has been removed, and the relevant correlation
analysis has now been updated and relocated to **Supplementary Fig. 6** and **Extended Data Fig. 3j**
in the revised manuscript.

Line 148: "The 20 most impactful features were identified, and their effects were visualised in a SHAP
summary plot, offering a concise yet comprehensive view of the model's analytical depth (Fig. 2a)." I
think this is very nice data, potentially leading the way into new scoring systems in PD models.
I do agree that for human eye unseen features may have been hidden in many PD models, and thus
the AI based approach may open new doors in that direction. It will be interesting to see if authors

linked PD features to striatal DA, DA fibers and DA neuron numbers in the SN later in the ms?
(after reading the ms the answer is yes).

I also appreciate that the difference between Normal (again, needs to be analyzed properly as
suggested above) and “PD” animals for the 20 ML/AI defined features (S Figures 4-6). However,
such clear difference in quantitative data makes me ask why was statistics not applied? Seems like
very significant difference for most endpoints?

Response #2.23

We thank the reviewer for this valuable feedback. As the reviewer noted, the differences in the 20
AI-defined behavioural features between the Control (CT) and PD (A5) groups were clear. Initially, we
did not include statistical comparisons, as our primary intention was to demonstrate data distributions,
given our plan to perform further comparisons with additional groups later in the manuscript (**original**
**manuscript, Supplementary Fig. 9**). However, following the reviewer’s suggestion, in the revised
manuscript, we have now explicitly included statistical analyses for the distributions and feature
values, which are detailed in **Supplementary Figs. 12, 13; Supplementary Tables 2, 3**. This update
provides robust statistical support for the observed differences, enhancing the transparency and
scientific rigour of our findings.

Line 162: “Another advantage of our model is its ability to account for feature interactions. SHAP
dependence plots for each feature illustrate the influence of other features as the degree of vertical
dispersion of the data points (Extended Data Fig. 4).” I wonder in what way the authors mean the
reader should gain more insight into the above advantage by looking at Extended Data Figure 4? I
think this should be explained/visualized better with concrete correlating features. Thus I d improve
resolution of the statement at Line 165-166 “We employed colour coding to represent the values of
the most interacting feature to the plotted feature, elucidating the relationship between them, which
otherwise would remain undetermined.”

Response #2.24

We thank the reviewer for this valuable suggestion. In the revised manuscript, we have provided
explicit explanations with concrete examples in the **legend of Extended Data Fig. 4** to better
illustrate how the SHAP dependence plots highlight feature interactions. Specifically, we clarified that
the interacting feature for each plot is automatically determined by the SHAP library based on the
highest correlation with the plotted feature’s SHAP values. This correlation-based approach helps
visually highlight significant interactions, enhancing the interpretability of the dependence plots. For
instance, subplot #1 shows that smaller values for hand 3D distance (#1) have greater predictive
impact when the interacting feature, body 2D length (min), is longer. Similarly, subplot #8 illustrates
that larger feet distances (max) (#8) more strongly influence the prediction during shorter motion
durations. These explicit examples clarify how SHAP’s automated colour-coding reveals meaningful,

context-dependent interactions between behavioural features, thereby addressing the reviewer's
concern about clarity and interpretability.

Perhaps again highlighting the potential of AVATAR in comparing WT to AAV-vehicle, AAV-RFP and
AAV-A53T mice? This experiment takes WT animals +injections and 10 weeks followup plus analysis,
thus I d believe is quite in the range of a revision.

**Response #2.25**

We thank the reviewer for this insightful suggestion. In revision, we tried applying our existing ML
pipeline (without further adjustments in dataset filtering or feature engineering) and found that our
trained models were unable to discriminate between the subgroups within the Control (CT) animals
(NI, EV, and RFP groups). Nevertheless, we agree with the reviewer that developing a model
specifically designed to discriminate between these CT subgroups could provide valuable insights into
subtle behavioural differences attributable to mechanically induced lesion (MIL) versus protein
overload lesion (POL). However, achieving this would require modifications to our current ML
methodology, which is beyond the scope of this current revision but represents an interesting and
important direction for future research.

Line 168: "The XGB model pinpointed the top 20 features as crucial for PD diagnosis, particularly
emphasising limb movements by ranking half of these features related to hands and feet (Fig. 2a)." I
recommend colour coding the hand-feet rows to make Fig 2 a reader friendly.

**Response #2.26**

We thank the reviewer for this helpful suggestion. In **Fig. 2a**, colour-coded boxes already
indicated the hand- and foot-related features in the second and third columns. To further improve
readability and clearly highlight these features, in the revised manuscript we have additionally shaded
the rows corresponding to the hand- and foot-related feature names. We believe this revision
enhances visual clarity and makes **Fig. 2a** more reader-friendly, directly addressing the reviewer's
recommendation.

Lines 170-178 – I consider this data very important, suggesting that "PD" animals do display clear
hand and feet coordination defects. I agree that those features are hard or almost impossible to see,
let alone properly score by human eye and by regular human eye-based tests. However, statistical
evaluation of this likely numerical data would be important for scientific rigor? In case not possible to
implement then explain why?

**Response #2.27**

We appreciate the reviewer's emphasis on statistical rigour regarding limb coordination features.
As previously mentioned, (**Response #2.23**), our original manuscript did not include statistical
analyses specifically in Fig. 2, as our primary intention was model interpretation to characterise PD
versus non-PD (NP). However, following the reviewer's valuable suggestion, we have
comprehensively addressed this point by including detailed statistical analyses of individual
behavioural features in the revised manuscript. Specifically, we have applied two-way ANOVA and
Welch's t-tests to evaluate longitudinal variability and individual feature differences, as now detailed in
Supplementary Tables 2 and 3, and visualised in Supplementary Figs. 10–13.

Nevertheless, as noted explicitly in **lines 226–218**, 'this single-feature-focused analysis provides
valuable insights but does not capture potential interactions or combined effects of multiple features,
thus warranting cautious interpretation'. As further explained in the **legend of Extended Data Fig. 5**,
our integrated analytical approach in this section (**Fig. 2**) combines multiple complementary analyses
(including SHAP values, raw feature values, and decision-tree splits). These integrated analyses often
reveal complex, non-linear relationships and multiple distinct clusters or inflection points in feature
distributions, which conventional statistical tests at aggregated or single timepoints cannot
meaningfully represent. Additionally, direct statistical analysis solely based on SHAP values is limited,
as SHAP values inherently quantify the model's existing predictive differentiation between groups
rather than testing for significance.

Therefore, while direct statistical analysis of these integrated SHAP outcomes is challenging, we
believe that the statistical analyses at the individual feature level in our revised manuscript effectively
address the reviewer's concern, enhancing transparency, clarity, and overall scientific rigour.

Lines 180-187 describe increased movement in PD model and the likely explanation - I feel this
should come earlier in the ms, where increased/decreased movement in PD models is first described.

**Response #2.28**

We thank the reviewer for this valuable suggestion. As mentioned earlier (**Response #2.7**), we
have moved the explanation regarding increased movement observed in our PD mouse model earlier
in the revised manuscript (**lines 103–105**). This revision ensures that the reader is provided with
appropriate context at the first mention of altered locomotion, enhancing clarity and logical flow of the
manuscript.

Lines 188-213 over body posture and stooped gait – also this data is very important and really
promising. Overall, I feel that the authors might make even stronger case if they would repeat the
study using another PD model, for example in striatal 6-OHDA or supranigral lactacystin-based PD
models. Are the reported features, as expected, universal "Mouse lack of nigrostriatal DA features?" If
so, we would be looking at new PD AI-based standard set of disease features in mice, turning page
for the field?

(Now after reading the whole ms until the end I tend to think that this is not required in this ms. Right
now it is more important to publish and let the field absorb and repeat in other PD models).

**Response #2.29**

We thank the reviewer for this insightful and encouraging comment. We strongly agree that it
would indeed be very valuable and interesting to compare behavioural features across different PD
models such as 6-OHDA etc., to identify commonalities or differences that might reveal universal
signatures of nigrostriatal DA deficits. However, as the reviewer rightly recognises, such a
comparative study would require substantial additional experimental work, which we feel is beyond
the scope of the current manuscript. Related to this, we briefly discussed the limitation of our study
being based on a single PD model, noting that a potential direction for future work would be to
develop a universally applicable AI model capable of capturing 'common traits' across multiple PD
models (**lines 499–501**). Nonetheless, we consider this an important and promising direction for
future studies, and we sincerely appreciate the reviewer's understanding of this practical constraint.

However, I d again would like to point out that proper comparison/repeat analysis of WT to AAV-
vehicle to AAV-RFP to AAV-A53T mice is important. It may help to train/tune the ML/AI system better,
hopefully leading to clearer PD definition in mice and perhaps approaching then 100% accuracy as
opposed to Figure 1e and f current results which, according to authors are close to about 85% PD
prediction for 5X and about 51% for 1X.

**Response #2.30**

We thank the reviewer for raising this important point. In the revised manuscript, we explicitly
tested whether training the AI model with an increased number of mice would enhance prediction
accuracy; however, this did not result in noticeable improvements (**related to Response #1.2;**
**Supplementary Fig. 5**). As our primary objective was to distinguish PD-related (AIL) features from
non-PD (NP) features, we trained the non-PD (NP) class using data from the NI, R1, and R5 groups.
Accordingly, our XGB model and the derived APS metric are specifically tuned to capture behavioural
features associated with AIL, rather than those arising from POL.

In the original manuscript, prediction accuracy was approximately 85% for the A5 and 51% for the
A1 groups. In the revised manuscript, these values slightly improved to approximately 87% and 59%,
respectively. The reason APS does not reach 100% accuracy is mainly due to a small fraction of non-
PD-classified motion clips (false positives) present within the PD (A5) group. This outcome is
expected, given our analysis is based on specific motion clips, and even PD mice may occasionally
exhibit non-PD-like movements. Importantly, however, our model still reliably classifies nearly all A5
mice into the severe PD category (APS > 75%), as demonstrated clearly in **Supplementary Fig. 5**.

Overall, we consider this accuracy sufficient for reliably distinguishing PD severity using the
AVATAR system, although further refinement remains a valuable objective for future studies.

Lines 214 – 282. I absolutely love optoRET system with blue LED activation from cage lids!
It shows that Daily RET activation regimen does not restore the DA system, which can be
fundamental information to the field. This has clear implications to ongoing clinical trials with AAV-
GDNF where GDNF cannot be turned off – which may lead to another failure based on REF 14 and
on PMID: 37766790. Authors nicely demonstrate the critical role of timing and duration of RET
activation in alleviating SN lesion driven defects. Essential and elegant.
I would perhaps, suggest reducing technical detail somewhat as exact Hz of chest tremor and
possible link to human tremor in PD are possibly better topics for Discussion section, or for Extended
Discussion section rather than for Results as we do not know how well those features if at all repeat in
other supranigral lesion PD models and thus if they emerge as universal mouse PD correlates.

**Response #2.31**

We thank the reviewer for the insightful comment and for appreciating the potential implications of
our optoRET system findings. We agree that the spectral analysis results (e.g. exact Hz of chest
movement) represent a technically detailed discussion; however, we included this because it revealed
unexpected, intriguing findings potentially relevant to human PD tremor frequencies¹³. Although we
fully acknowledge the reviewer’s important point that the universality of these features across other
PD models remains untested, we believe that highlighting this potential link in the Results section is
valuable due to its possible translational implications.

Nevertheless, we completely agree with the reviewer that caution is warranted in interpreting
these findings, given the absence of direct supporting accelerometer or electromyography data.
Therefore, in the manuscript, we carefully described these results as ‘potential links’ or having
‘potential moderating effects’ on tremor-like behaviours to clearly communicate these limitations to
readers (lines 365–367).

If the reviewer still feels strongly about moving this discussion elsewhere, we remain open to
reconsideration and further adjustments. However, currently, we believe this nuanced presentation in
the Results section appropriately balances scientific interest with necessary caution.

Lines 283 – 307, great data, may indicate optoRET specific features as well as reveal differential
treatment potential, might set stage in the future for human AI based optoRET dosing regimen
optimization, possibly guide way to precision medicine (PM) depending on individual treatment
response? (Now after reading the ms I see that the authors indeed point this out and discuss this)

**Response #2.32**

We thank the reviewer for this encouraging comment. We completely agree that our data
revealing differential responses to optoRET stimulation regimens indeed have exciting implications for
future research, particularly regarding the development of AI-based strategies to optimise RET-
activation protocols. As the reviewer noted, we explicitly discuss in the manuscript how these findings

could guide personalised treatment regimens and potentially facilitate precision medicine approaches
in PD therapy (lines 494–496). We greatly appreciate the reviewer’s recognition of the significance
and potential impact of this aspect of our study.

Lines 308 – turning and velocity, gait. Again, great data and analysis revealing new endpoints not
achievable with any other sofar described tool! Possibly setting stage for PM at individual level first in
animals, then hopefully in PD.

**Response #2.33**

We sincerely thank the reviewer for this very encouraging comment. We strongly agree that our
detailed analysis of turning behaviours, velocity, and gait, enabled uniquely by our AI-based approach,
reveals behavioural endpoints previously unattainable with conventional methods. As the reviewer
insightfully highlights, these findings indeed have the potential to lay the groundwork for precision
medicine strategies, initially in animal models and ultimately translating into individualised therapeutic
approaches for PD patients. We appreciate the reviewer’s support and recognition of this important
potential direction.

Discussion

The discussion, in large, covers the manuscript well. It also addresses several points and questions
which rose while reading the ms. I agree that perhaps in this manuscript, it is not the right place to
repeat the studies with another PD model as indeed, there are many and none is really good or at
“gold standard” level.

**Response #2.34**

We sincerely thank the reviewer for this supportive and insightful comment. We fully agree that,
given the current context, repeating our analyses in additional PD models would substantially exceed
the scope of the current manuscript, especially considering the inherent limitations and variability of
existing PD models. Nevertheless, we recognise the value of exploring multiple models in future work
to further validate and generalise our AI-based findings. We appreciate the reviewer’s understanding
and acknowledgment of this practical consideration.

Overall, I recommend publication with revision of some technical points and with some improved
clarifications during the first part of the manuscript. Great work with implementing AI, loads of new
endpoints found and also very revealing data on the timing of RET, this could be pointed out even
better as it may inform ongoing and planned clinical trials.

**Response #2.35**

We sincerely thank the reviewer for their supportive and constructive comments. We have
carefully addressed all technical points and improved the clarity of explanations, especially in the first
part of the manuscript, following the reviewer's suggestions.

Additionally, we have explicitly emphasised in the Discussion the potential clinical relevance of
our findings regarding the timing and frequency of optoRET activation. In the revised manuscript, we
have also included supporting results from the L-DOPA and BT44 treatment groups as supplementary
evidence (**Supplementary Notes 7,8**). These findings highlight that optimising stimulation protocols –
both pharmacological and optogenetic – could significantly enhance therapeutic efficacy and inform
optimal delivery schedules in ongoing and future clinical trials targeting the RET signalling pathway
(lines 473–478). We greatly appreciate the reviewer's recognition of our work and its potential impact
on the field.

Also, it may make sense to discuss that the lack of good PD models may in fact at least in part relate
to low resolution of human eye, now potentially overcome by AI based array of mouse PD features?

**Response #2.36**

We thank the reviewer for this insightful suggestion. We fully agree and have included this point
explicitly in our revised manuscript's Discussion (**lines 496–4799**). Specifically, we highlight that
previous challenges in identifying reliable PD animal models may partly arise from the limitations of
traditional human-observer-based behavioural assessments. Such limitations could now be overcome
by using AI-driven, high-resolution feature analyses, potentially revealing previously overlooked,
subtle, yet critical PD phenotypes, thereby significantly enhancing the reproducibility and robustness
of PD animal models.

Besides the improved presentation the only experiment I strongly recommend is analysis of WT v
AAV-vehicle v AAV-RFP v AAV-A53T mice at 1X and 5X to understand what feature comes from
what, and to solidly reproduce the data. This may also help to further improve the PD prediction. I
realize that this is few months of quite hard work but I think it would both show the power of AVATAR
and better set the stage against the current PD field where genetic PD models most often than not do
not reproduce well between the studies. AVATR could change that.

**Response #2.37**

We sincerely thank the reviewer for this thoughtful and valuable recommendation. We fully agree
that systematically comparing WT, AAV-vehicle, AAV-RFP, and AAV-A53T groups at both 1X and 5X
would substantially enhance our understanding of specific behavioural features and their underlying
causes, further demonstrating the power of the AVATAR system. In the revised manuscript, we have
addressed this issue by clearly separating and analysing the control groups (**Extended Data Fig. 1**).

Again, great work!

**Reviewer #3 (Remarks to the Author):**

Hyeon et al. introduced a novel approach to investigate the behavioral impairments in an animal
model of Parkinson's disease (PD). The authors utilized a behavior monitoring system equipped with
multiple cameras to detect subtle movements of individual animals and record various characteristics
relating to their movement and posture. The results included a significant amount of data sets and
they effectively organized them to provide well-defined behavioral phenotypes associated to
Parkinson's disease. This comprehensive approach to behavioral phenotyping is innovative and
would provide valuable insights for future research on movement disorders. I have just a few minor
concerns.

Thank you for your positive assessment of our study; we appreciate your recognition of the
value of our comprehensive behavioural phenotyping approach and will carefully address the
concerns raised.

1. The process of generating the PD score is not well explained. It is unclear whether the PD score
reflects only the 20 major features or if other features are also taken into account. How individual
features are weighted for this calculation should be explained with more clarity.

**Response #3.1**

We thank the reviewer for highlighting this important point. In the manuscript, we clarified how the
AI-predicted PD scores (APS) are generated (**lines 170–172**). In revision, we have updated the
relevant panel (**Extended Data Fig. 3e**) to include a schematic representation of the explainable AI
system, clearly illustrating how different features in the input dataset contribute to the model's
prediction (PD probability). Regarding how individual features are weighted within the XGB model, we
provided a detailed description and comprehensive analysis in **Fig. 2 (lines 200–278)**. These
revisions should significantly enhance clarity and transparency regarding the APS calculation.

2. The description on A1X and A5X in the main text needs clarification. In line 92, the authors stated
that they injected AAV-RFP into DAT-Cre mice at two distinct doses, specifically A1X or A5X. It is
likely that AAV-RFP refers to an adeno-associated virus carrying the red fluorescent protein gene. If
so, the injection of AAV-RFP shouldn't induce PD-like behaviors. This misstatement should be
corrected. Overall, Multiple viral vectors were used in this investigation. Authors should include the full
name and specific amounts (e.g. concentration and volume) of the viruses in the main text.

**Response #3.2**

We thank the reviewer for pointing out this important oversight and apologise for the confusion
caused by our original description. In the revised manuscript, we explicitly clarify the Control (CT) and
PD groups in the main text (**lines 110–112**). We also include a concise summary table clearly

outlining all experimental groups, including full virus names, serotypes, concentrations (titres), and
injection volumes in **Fig. 1b**, supplemented by a more detailed version in **Supplementary Table 1**.
These revisions significantly enhance clarity, accuracy, and reproducibility of our experimental
methodology.

3. While the optogenetic activation of RET signaling in dopaminergic neurons is novel and interesting,
it would be challenging to offer practical applications. Other medications that specifically target RET
signaling, such as pre-clinically validated RET agonists can be utilized for treatment and their effects
can be compared to the effects of OptoRET.

**Response #3.3**

We thank the reviewer for this insightful comment. We fully agree that, although optogenetic
activation of RET signalling using optoRET provides valuable proof-of-concept and mechanistic
insights, its direct clinical translation remains challenging (despite the FDA approval of
channelrhodopsin and the clinical use of deep-brain implanted devices). As the reviewer rightly notes,
alternative RET-targeted therapeutics, such as pharmacologically validated RET agonists, may offer
more immediate translational promise. However, currently available RET agonists lack the capacity for
temporal control and face substantial limitations regarding pharmacokinetics^{14, 15}, including poor
systemic delivery and potential off-target effects – particularly in tissues with high RET expression,
such as the gastrointestinal tract. One of the most extensively validated preclinical RET agonists,
BT44, has thus far only been shown to be effective in a rat 6-OHDA model when delivered directly to
the brain via infusion⁹.

To directly address the reviewer's valuable suggestion, we conducted an additional experiment in
our revised manuscript, introducing a new experimental group, A1BT, in which BT44 (120 μ M) was
infused bilaterally into the brains of A1 mice at a constant flow rate of 0.18 μ L/hr using osmotic pumps
(See Methods, **lines 551–564**). Due to the technical limitations of the pumps' capacity, the experiment
concluded at 6 wk post-virus injection. Despite these constraints, we evaluated the therapeutic
efficacy of BT44 relative to the optoRET group (A1O3). To maintain focus on the central narrative of
the manuscript, we briefly described the results in the main text (**lines 306–309**), with a more detailed
interpretation provided in **Supplementary Note 7**.

Our findings indicated that BT44 was overall ineffective in the A1 PD model (**Extended Data Fig.**
**7; Supplementary Fig. 22**). An initial beneficial trend was observed in the elevated beam walking test
score (BWS) and AI-predicted PD score (APS), albeit smaller in magnitude than in the A1O2 group
(**Extended Data Fig. 6f**). Additionally, the rotarod test score (RRS) showed improvement, although
this did not reach statistical significance compared to the A1 group (**Extended Data Fig. 7d**). These
results suggest that BT44 may moderately improve general motor coordination (RRS) but fails to
significantly enhance fine motor control or balance (BWS) or broader PD-related features (APS).

The early beneficial trend could reflect the agonist's dependency on endogenous RET receptors,
which are known to be downregulated during PD progression – particularly in aSyn overexpression
models^{10, 16, 17}. We propose that the limited efficacy results from a combination of factors: (1) a
reduction in endogenous RET expression in our model, where aSyn overexpression likely exceeds
the levels typically seen in human PD pathology¹⁰; and (2) diminished neuroprotective efficacy
following prolonged ligand exposure^{11, 18}. To distinguish whether these effects arise primarily from
receptor downregulation or prolonged stimulation, further studies are warranted using alternative
aSyn models, such as the aSyn-preformed fibril model, in which GDNF/RET signalling has shown
therapeutic benefit¹⁹.

Finally, treatment response evaluation (TRE) for the A1BT group indicated that only
approximately 10% of behavioural features were marked as treated (**Supplementary Fig. 22**), further
supporting the limited efficacy of BT44 in this context. A comparative TRE analysis across optoRET
(A1O2, A1O3) and RET agonist (A1BT) groups also revealed common off-target effects on
locomotion, potentially reflecting RET signalling activation in brain circuits that influence motor
behaviour.

We believe these additional experiments strengthen our manuscript by clearly illustrating the
comparative efficacy of optoRET over pharmacological RET agonists and offer mechanistic insights
that may help inform future RET-targeted therapeutic strategies for PD.

**Reviewer #4 (Remarks to the Author):**

In the manuscript "Integrating artificial intelligence and optogenetics for advanced Parkinson's disease
diagnosis and therapeutics", the authors show a novel AI analysis of sensorimotor function and
movement in an AAV-A53T mouse model of PD that can discriminates control mice from mice with
different severity of nigrostriatal lesions. The AI and kinematic analysis system was able to detect
multiple motor differences such as alterations in limb coordination, more variable movements, shorter
paw movement amplitudes, and wider gait stance. They then demonstrate that an optogenetic
intervention targeting c-RET signaling can improve multiple aspects of motor dysfunction. The AI
motor analysis system is impressive and has the potential to be highly useful for ontogenetic studies.
However, there are some concerns regarding the specificity to PD and whether it would show a
similar profile of motor dysfunction in other PD models.

Thank you for your encouraging feedback; we acknowledge the concern regarding PD
specificity and have included additional discussion to clarify the applicability and limitations of our
findings across different PD models.

1. The authors show that the AI-kinematic analysis system is more sensitive at detecting changes in
motor function than an elevated balance beam test but it is not clear how the system compares to
other behavior tests that are commonly used including rotarod and gait treadmill equipment (ex.
Catwalk).

**Response #4.1**

We thank the reviewer for this valuable comment. We acknowledge that our original manuscript
specifically compared the AI-based analysis (AVATAR) primarily to the elevated beam walking test
(BWT), without direct comparisons to other commonly used behavioural assessments such as the
rotarod or gait-analysis systems (e.g., CatWalk).

In the revised manuscript, we directly addressed this limitation by including additional behavioural
analyses using the rotarod and tail suspension tests (**Extended Data Fig. 1; Supplementary Fig. 2**).
Regarding gait analysis, we believe additional testing with other gait-assessment systems is
unnecessary, as our detailed gait analysis derived from the pose-based dataset already provides
comprehensive coverage (**Fig. 5, lines 410–437**). These additional experiments help clarify how
AVATAR compares to traditional behavioural tests, further demonstrating the sensitivity and specificity
of our AI-based system in capturing subtle motor deficits associated with PD (**Fig. 1h; Extended Data**
**Fig. 3g–j**). We appreciate the reviewer's insight, which has significantly strengthened our manuscript
by broadening the comparative perspective of our AI-based analysis.

2. Only one PD model was assessed, the AAV-A53T SNc injected mouse. For the AI system

approach to be impactful it would have to show sensitivity across several commonly used models
including the alpha-synuclein preformed fibril model and potentially a toxin-based model (MPTP or 6-
OHDA).

**Response #4.2**

We strongly agree with the reviewer that comparing behavioural features across multiple PD
models – including aSyn preformed fibril and toxin-based models (e.g., MPTP or 6-OHDA) – would be
highly valuable and informative. However, as also noted by another reviewer (**Response #2.29**),
performing such extensive additional experiments currently lies beyond the practical scope of our
study. Nonetheless, we explicitly discuss this limitation in the manuscript (**lines 495–501**) and
highlight the importance of future studies that extend our AI-based approach to a wider range of PD
models, potentially identifying universal behavioural signatures of PD pathology. We appreciate the
reviewer’s understanding and valuable suggestion.

3. It is not clear how specific the findings are to PD, a spinal cord injury model or motor neuron
disease model would be important to assess and compare with PD models.

**Response #4.3**

We appreciate the reviewer’s critical comment regarding the specificity of our findings to PD. The
reviewer suggested comparing our PD model with a spinal cord injury model or a motor neuron
disease (MND) model. Spinal cord injury models typically result in severe or complete paralysis,
making them less suitable for comparison to the progressive neurodegeneration observed in PD.
Therefore, among MND models, we selected an amyotrophic lateral sclerosis (ALS) mouse model, as
it is characterised by progressive neurodegeneration, distinct from PD, providing an informative
comparison (**Supplementary Figs. 6 and 14–18**).

Due to the short lifespan of ALS transgenic mice (SOD1-G93A), the experimental timeline was
adjusted to mice aged 10–20 wk. To address potential confounding from age, we also included an
age-matched young mouse cohort in revision, which achieved a high APS (**Supplementary Fig. 9d**),
supporting that age is unlikely to affect APS significantly. Our evaluation of ALS mice showed clear
motor dysfunction (RRS = $6.11 \pm 1.85\%$), yet they achieved an APS of only $4.24 \pm 1.38\%$ (n = 2;
**Supplementary Fig. 6**). We acknowledge the small sample size as a limitation; however, the data
clearly indicates differential APS outcomes compared to PD. These findings have been incorporated
into the revised manuscript (**lines 184–188**) to clarify the observed differences in APS between ALS
and PD models.

Additionally, we extended our analysis of the ALS model to include longitudinal distribution
analysis (**Supplementary Figs. 14–17**) and endpoint comparisons of the top 20 features against the
PD cohort (**Supplementary Fig. 18**). The ALS group displayed notable variances in feature

distributions, importantly demonstrating distinct patterns of divergence compared to PD mice. Notably,
several features exhibited opposite directions of change relative to their respective controls, further
emphasising the specificity of our findings to PD (revised manuscript **lines 213–216**).

Collectively, these results support our conclusion that the high APS observed in PD groups
specifically reflects PD-related pathology rather than generalised motor dysfunction. We believe these
revisions adequately address the reviewer's concerns regarding the specificity of our findings.

4. While the findings that the optogenetic intervention targeting c-RET is exciting, it would be
important to show the effect of L-DOPA in the AAV-A53T model on the AI analysis.

**Response #4.4**

We thank the reviewer for this insightful comment. Levodopa (L-DOPA; 3,4-Dihydroxy-L-
phenylalanine), a dopamine precursor, remains the gold-standard symptomatic treatment for PD.
However, its clinical utility becomes increasingly limited as PD progresses, primarily due to the
progressive loss of dopaminergic neurons, which narrows the therapeutic window¹³. Over time, the
therapeutic benefits of L-DOPA often diminish ('wearing-off' between medications), whilst side effects
such as L-DOPA-induced dyskinesia (LID) emerge at peak concentrations^{13, 20}. There is general
agreement that LID arises from pronounced, intermittent fluctuations in brain DA levels, leading to
abnormal stimulation of striatal dopamine receptors²⁰.

To directly address this suggestion, we conducted additional experiments by introducing new
treatment groups (A1L1 and A1L3), in which L-DOPA (50 mg/kg) was administered intraperitoneally
either daily or on alternate days, respectively (Methods, **lines 581–590**). To maintain focus on the
central narrative of the manuscript, we briefly described the results in the main text (**lines 306–311**),
with a more detailed interpretation provided in **Supplementary Note 8**.

In our hA53T overexpression PD mouse model, we observed no significant therapeutic
improvements across all assessed metrics, including RRS, BWT, and APS (**Extended Data Fig. 7**).
Notably, the rotarod test score (RRS) showed an early trend towards worsening, although this effect
was not significant at the endpoint. Furthermore, a significant increase in neck/trunk dystonia was
observed, measured by the axial bending angle²⁰ (**Extended Data Fig. 7j–m**). These findings may
reflect the consequences of our high-dose intermittent dosing protocols, potentially exacerbating side
effects through dramatic fluctuations in striatal dopamine levels. Finally, treatment response
evaluation (TRE) further revealed notable off-target effects on overall motor coordination
(**Supplementary Fig. 22**).

Collectively, these results may highlight the importance of selecting appropriate medication
protocols based on therapeutic strategy – continuous delivery may be more suitable for L-DOPA²¹,
whereas intermittent stimulation appears better suited to c-RET activation^{11, 18} via optoRET. We
believe this revision adequately addresses the reviewer's point.

5. It is not clear how much nigrostriatal cell loss there is in the AAV-A53T mice, from the picture in
Figure 3e and the graph in Figure 3f it looks pretty severe in the A1X mouse, which is supposed to be
less affected line. If it is severe loss then the AI system is not necessarily that sensitive. Was
stereology performed?

**Response #4.5**

We thank the reviewer for raising this important point. As noted, although we designated the A1
group as a relatively milder PD model compared to the A5 group, this group already exhibited
significant motor symptoms and substantial neuropathology (**Extended Data Fig. 1g–j**). Therefore,
we acknowledge that our original use of the term ‘milder’ could be misleading, given the notable
severity of nigrostriatal cell loss observed.

Importantly, our study primarily aimed to demonstrate a proof-of-concept, highlighting the
sensitivity and precision of our AI-based system in detecting subtle behavioural changes and enabling
early PD detection beyond the limitations of traditional assessments (**Fig. 1h**). Although the A1 model
is already quite severe, AVATAR successfully discriminated clearly between the two different PD
severity cohorts.

Nevertheless, we fully agree that evaluating a genuinely milder PD model, such as one
representing a silent motor period (10–20% DA neuronal loss) or a prodromal/pre-symptomatic stage
(20–30% DA neuronal loss)²², would better illustrate the true sensitivity and clinical utility of our AI
approach. We consider this an important direction for future studies.

Finally, regarding quantification of DA neuronal loss, we confirm that stereological analysis was
performed and have explicitly revised the Methods section to clearly describe this stereological
quantification (**lines 603–608**).

6. The word "transverse" is used several times when describing the beam test and I think they mean
to use "traverse" instead.

**Response #4.56**

We thank the reviewer for pointing this out. We agree with the correction and have revised the
manuscript to replace all incorrect uses of ‘transverse’ with ‘traverse’ when describing the mouse’s
movement across the beam.

**References (in this point-by-point responses)**

[revised manuscript text omitted]

1453

**Reviewer #1 (Remarks to the Author):**

The authors have implemented the majority of the requested revisions. Although the proposed
optogenetic validation (using implanted fibers) was not performed, I do not consider it critical for the
current scope of the study.

A remaining weakness, however, is the limited contextualization of AVATAR (and AVATARnet)
relative to existing keypoint-tracking and behavioral-analysis frameworks that are widely used in
neuroscience. At a minimum, the revised Discussion should devote a brief paragraph (e.g. ~3-5
sentences) that:

We thank the reviewer for this insightful suggestion. In response, we have revised the Discussion
(lines 503–520) to addresses the reviewer’s points as follows.

- Positions AVATAR against leading open-source keypoint-tracking pipelines such as DeepLabCut
(DOI: 10.1038/s41593-018-0209-y), LEAP (DOI: 10.1038/s41592-018-0234-5), SLEAP (DOI:
10.1038/s41592-018-0234-5), DANNCE (DOI: 10.1038/s41592-018-0234-5) or other approaches, as
well as behavioral-analysis pipelines such as MoSeq (DOI: 10.1038/s41592-018-0234-5; DOI:
10.1038/s41592-018-0234-5), A-SOID/B-SOID (DOI: 10.1038/s41592-024-02200-1; DOI:
10.1038/s41467-021-25420-x).

We now clarify that AVATAR differs from 2D keypoint-based systems such as DeepLabCut, LEAP,
and SLEAP, which typically require additional pipelines for 3D inference, and from keypoint-free
voxel-based systems such as MoSeq. We highlight AVATAR’s ability to provide high-resolution, out-
of-the-box 3D tracking via five-camera triangulation and a pre-trained YOLO-Darknet backbone.

- Summarizes the key technical differences of the approach used in the manuscript (e.g. multi-camera
3-D triangulation, computational throughput, etc.) and indicate whether the phenotyping described
here extends, like other pipelines, to other diseases models such as models of autism (MoSeq: DOI:
10.1038/s41593-020-00706-3), epilepsy (MoSeq: DOI: 10.1016/j.neuron.2023.02.003) or Huntington’s
disease (B-SOID: DOI: 10.1186/s12915-024-01919-9).

We describe AVATAR's key technical attributes, including its multi-camera triangulation and
integrated classification-interpretation pipeline. We also note that although our current study focuses
on Parkinson's disease, the framework's modularity may extend to other disease models such as
autism, epilepsy, and Huntington's disease.

- Notes the absence of benchmarking on an independent public dataset, explains whether domain-
specific constraints (e.g. unique enclosure geometry) preclude direct comparison, or outlines future
plans to release data or to participate in common benchmarks (e.g. CalMS21, B-SOID datasets).

Incorporating this short comparative paragraph would satisfy community expectations for situating
new AI methodologies and will help readers judge when AVATAR is preferable to existing tools.

Provided the authors add this discussion point, I recommend acceptance of the manuscript in its
present form.

We acknowledge that direct benchmarking was not performed, citing AVATAR's domain-specific
constraints due to its specialised multi-camera setup. To support future benchmarking, we have
deposited a representative dataset including multi-view videos and corresponding 3D keypoints in a
public repository (<https://doi.org/10.6084/m9.figshare.29840969>; See Data Availability section), which
may facilitate cross-platform comparisons.

**Reviewer #2 (Remarks to the Author):**

I would like to thank the authors for addressing my concerns. I still suggest one minor but what I feel
is an important revision - please include actual data from R1 and R5 studies (AAV-RFP delivery) on
the main Figures/Tables throughout the manuscript. R1 and R5 mice show that RFP has toxic effect
on DA neurons/DA function on its own, and it is very important and revealing for the PD field to see
this throughout the story as part of the main data, including what AI can and currently cannot achieve
there. I do not think that it damages the PD model story - aSyn over-expression v RFP over-
expression are just different ways to damage the DA system as is mechanical damage (denoted MI or
EV by the authors throughout the ms).

Apart from that I have no further comments or concerns.

We thank the reviewer for emphasising the importance of including the R1 and R5 control subgroup
data (AAV-RFP delivery). In response, we have revised the manuscript to incorporate these data in
the main Figure 1, which now presents DA neuron-related metrics across all relevant subgroups,
including R1 and R5. Additionally, we introduced a new main table titled 'Quantitative comparison of
analytic metrics across experimental groups affecting the DA system', which summarises key
behavioural and pathological metrics across the different lesion models. We agree that the inclusion
of RFP-related data adds valuable insight into how RFP expression independently affects the DA
system and further contextualises the AI analysis.

We sincerely appreciate the reviewer's constructive feedback and continued support throughout the
revision process.

**Reviewer #3 (Remarks to the Author):**

The authors addressed all my previous concerns. I believe the authors incorporated an extensive
amount of data to address the concerns of other reviewers and enhance the quality and conclusions
of this work.

We sincerely appreciate the reviewer's positive assessment and thoughtful comments. We are
grateful for your recognition of our efforts to enhance the quality of the manuscript through the
incorporation of additional data and revisions. Your support is sincerely appreciated.

**Reviewer #4 (Remarks to the Author):**

The authors were responsive to the critiques and this reviewer believes the manuscript is improved
both in terms of rigor and transparency.

-It is appreciated that a model of ALS was included to show the difference between a PD model and
another movement disorder.

-The authors included new results from L-DOPA administration. The dose was high though and well
above the therapeutic range (50mg/kg administered and therapeutic dose is ~6mg/kg) and likely
explains the lack of effect.

-While another PD model was not included they do address the limits of the findings using a single
model.

Overall, this work is important to the field.

We thank the reviewer for the encouraging feedback. We appreciate the acknowledgement of the ALS
model inclusion and the note on the high L-DOPA dose, which will guide future optimisation. We also
value the recognition of the study's importance and our discussion of the single-model limitation.